# Efficient Morphology-Control Co-Design via Stackelberg Proximal Policy Optimization

**Yanning Dai**[1*]    **Yuhui Wang**[1*†]    **Dylan R. Ashley**[1,2,3,4]    **Jürgen Schmidhuber**[1,2,3,4]

[1] Center of Excellence for Generative AI, King Abdullah University of Science and Technology (KAUST), Thuwal, Saudi Arabia.

[2] Dalle Molle Institute for Artificial Intelligence Research (IDSIA), Lugano, Switzerland.

[3] Università della Svizzera italiana (USI), Lugano, Switzerland.

[4] Scuola universitaria professionale della Svizzera italiana (SUPSI), Lugano, Switzerland.

## Abstract

Morphology-control co-design concerns the coupled optimization of an agent's body structure and control policy. This problem exhibits a bi-level structure, where the control dynamically adapts to the morphology to maximize performance. Existing methods typically neglect the control's adaptation dynamics by adopting a single-level formulation that treats the control policy as fixed when optimizing morphology. This can lead to inefficient optimization, as morphology updates may be misaligned with control adaptation. In this paper, we revisit the co-design problem from a game-theoretic perspective, modeling the intrinsic coupling between morphology and control as a novel variant of a Stackelberg game. We propose *Stackelberg Proximal Policy Optimization (Stackelberg PPO)*, which explicitly incorporates the control's adaptation dynamics into morphology optimization. By modeling this intrinsic coupling, our method aligns morphology updates with control adaptation, thereby stabilizing training and improving learning efficiency. Experiments across diverse co-design tasks demonstrate that Stackelberg PPO outperforms standard PPO in both stability and final performance, opening the way for dramatically more efficient robotics designs.

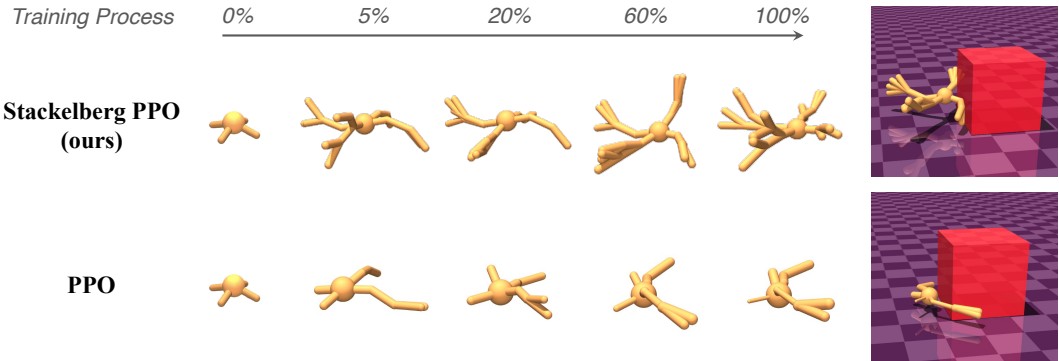

Figure 1: Showcasing how our Stackelberg PPO can autonomously design task-specific robots for the "Pusher" task, starting with a bare-bones structure and ultimately evolving into a sophisticated design with arm-like structures for pushing boxes and leg-like limbs for movement. This highlights the method's ability to create adaptive and complex designs. In comparison, the traditional PPO method generates simpler structures that can't support more complex behaviors. For more examples of evolved designs and animations, as well as open-source code, visit: https://yanningdai.github.io/stackelberg-ppo-co-design.

---

*Equal contribution.

†Corresponding author.

# 1 INTRODUCTION

Morphology-control co-design optimizes both an agent's body structure and its control policy. The morphology defines the agent's design, including topology, geometry, joint layout, and actuation limits, while the control policy dictates how this structure produces behavior to interact with the environment (Paul, 2006; Ha & Schmidhuber, 2018). Both aspects are critical for task performance. For instance, a quadruped robot with rigid legs can't walk without an appropriate gait policy, and a locomotion policy is ineffective if the robot's morphology lacks the necessary joints to support movement. These examples show that morphology and control must be co-designed to ensure complementarity. Agents optimized under this paradigm are typically more versatile, robust, and efficient than those optimized for only one aspect (Sims, 1994; Lipson & Pollack, 2000; Bongard et al., 2006; Kriegman et al., 2020).

A key challenge in morphology-control co-design is the dynamic interplay between morphology and control: the control policy must adapt to the evolving morphology to fully realize its potential; without this, the morphology's true performance may be underestimated (Schaff & Walter, 2022). Existing methods often treat morphology and control as separate optimization processes (Cheney et al., 2018; Lu et al., 2025; Yuan et al., 2022). Specifically, morphology updates are typically made assuming a fixed control policy, neglecting that control must adapt to structural changes. As a result, morphology updates become misaligned with the optimal control response, leading to unstable and inefficient optimization in the morphology space and, ultimately, degraded performance.

In this paper, we revisit the co-design problem from a game-theoretic perspective, formulating it as phase-separated Stackelberg Markov Games (SMGs), where the leader first updates the morphology, and the follower adapts its control policy accordingly. The key insight is that the leader must anticipate how morphology changes will influence the follower's control dynamics, enabling more effective designs. This perspective leads to the development of Stackelberg Implicit Differentiation (SID), a technique that incorporates the follower's adaptive dynamics into morphology optimization. However, applying SID to morphology–control co-design is non-trivial due to the phase-separated nature of the interaction and the non-differentiable interface between the leader and follower.

To address these challenges, we derive Stackelberg policy gradients tailored to phase-separated SMGs with non-differentiable interfaces, building on SID to explicitly account for the follower's adaptation in the morphology optimization process. Since direct differentiation is obstructed by the non-differentiable leader–follower interface, we apply the log-derivative technique (Williams, 1992) to derive a new Stackelberg surrogate formulation that bypasses this issue and provides a tractable gradient estimator. We further provide theoretical guarantees, showing that these surrogates are locally equivalent to the true Stackelberg gradients. To stabilize training under large policy shifts, we adapt PPO's likelihood-ratio clipping to our Stackelberg framework, ensuring robust optimization of the surrogate gradients. Our method, *Stackelberg PPO*, outperforms state-of-the-art baselines by 20.66% on average and by 32.02% on complex 3D tasks.

# 2 RELATED WORK

**Morphology–Control Co-design**   Co-optimizing morphology and control is attracting increasing attention in embodied intelligence (Li et al., 2024; Huang et al., 2024b; Liu et al., 2025). Prior work optimizes only continuous attributes under a fixed topology, without generating new structural topologies (Banarse et al., 2019; Huang et al., 2024a). Early topology-editing work treated co-design as a discrete, non-differentiable search solved using evolutionary strategies (Sims, 1994; Cheney et al., 2018), requiring each morphology to be paired with a separately trained controller and thus incurring high computational cost. Subsequent methods introduced structural priors and parameter sharing to reuse experience across designs (Dong et al., 2023; Zhao et al., 2020; Wang et al., 2019; Xiong et al., 2023). More recent RL-based approaches cast structure generation as sequential edits in an MDP (Gupta et al., 2021; Yuan et al., 2022), with graph and attention architectures improving representation quality (Chen et al., 2024; Lu et al., 2025; Yuan et al., 2022). However, the discrete nature of morphology-editing operations blocks gradient propagation across the morphology-control interface, preventing efficient learning by capturing their coupled dynamics. Our work establishes a gradient-driven pathway that allows controller adaptation to directly affect morphology updates.

**Stackelberg Game** Learning systems with asymmetric components often exhibit directional dependencies, where one module's decisions influence another's adaptation in a non-reciprocal manner (Schmidhuber, 2015). Stackelberg games formalize this asymmetry, with a leader committing to strategies that followers then best-respond to. Classical approaches study static normal-form games (Başar & Olsder, 1998; Conitzer & Sandholm, 2006; Von Stengel & Zamir, 2010) while more recent extensions integrate this structure into sequential decision processes and RL frameworks(Gerstgrasser & Parkes, 2023; Zhong et al., 2023; Bai et al., 2021), often incorporating confidence-aware or optimistic mechanisms to manage follower uncertainty (Ling et al., 2023; Kao et al., 2022; Kar et al., 2015; Mishra et al., 2020). Another line of research applies implicit differentiation to enable direct gradient flow from the follower to the leader in Stackelberg games. A complementary line leverages implicit differentiation to propagate follower gradients to the leader (Zheng et al., 2022; Yang et al., 2023; Vu et al., 2022), typically under DDPG-style settings with explicit action-level coupling and alternating updates. Our problem differs in two key aspects: leader actions (morphology edits) cannot be directly transmitted to the follower, and both agents use non-alternating PPO-style updates. We extend implicit Stackelberg gradient methods to this more general regime, enabling (to our knowledge) the first application of implicit Stackelberg differentiation to morphology–control co-design under PPO algorithm.

## 3 PRELIMINARIES

**Proximal Policy Optimization (PPO)** Our method builds upon PPO due to its empirical stability and simplicity. In reinforcement learning, an agent interacts with the environment by observing a state $s_t$, selecting an action $a_t$ according to its policy $\pi_\theta$, and receiving feedback in the form of rewards. Vanilla policy gradient methods (Sutton, 1984; Williams, 1992; Sutton et al., 1999) optimize $\pi_\theta$ using a surrogate objective that locally approximates the true performance, which has been shown to cause instability when policy updates become too large (Schulman et al., 2015). PPO (Schulman et al., 2017) addresses this by constraining the likelihood ratio between new and old policies through a clipping technique:

$$\mathcal{L}^{\text{PPO}}(\theta) = \mathbb{E}_t\Big[ \min\big(r_t(\theta)\hat{A}_t,\ \text{clip}(r_t(\theta), 1-\epsilon, 1+\epsilon)\hat{A}_t\big)\Big],$$

where $r_t(\theta) = \frac{\pi_\theta(a_t|s_t)}{\pi_{\theta_o}(a_t|s_t)}$ is the likelihood ratio and $\hat{A}_t$ is an estimator of the advantage function. The clipping mechanism prevents likelihood ratio $r_t(\theta)$ from deviating excessively, thereby limiting policy updates and balancing stability with performance improvement.

**Stackelberg Game** A Stackelberg game models a sequential and asymmetric interaction in which a leader first commits to a strategy, and a follower subsequently optimizes its strategy in response. *One key characteristic of a Stackelberg game is that the leader explicitly accounts for the follower's best-response dynamics when making its decision.* Let $\theta^L$ and $\theta^F$ denote the decision variables of the leader and the follower, and let $J^L(\theta^L, \theta^F)$ and $J^F(\theta^L, \theta^F)$ denote their respective objectives. The leader solves the following bi-level optimization problem, referred to as the *Stackelberg objective*:

$$\max_{\theta^L}\ J^L\big(\theta^L, \theta^F_*(\theta^L)\big) \quad \text{s.t.}\ \theta^F_*(\theta^L) = \arg\max_{\theta^F} J^F(\theta^L, \theta^F) \tag{1}$$

where $\theta^F_*(\theta^L)$ denotes follower's best response. The leader's gradient can be written as

$$\nabla_{\theta^L} J^L\big(\theta^L, \theta^F_*(\theta^L)\big) = \underbrace{\nabla_{\theta^L} J^L(\theta^L, \theta^F)}_{\text{direct gradient}} + \underbrace{\big(\nabla_{\theta^L}\theta^F_*(\theta^L)\big)^\top \nabla_{\theta^F} J^L(\theta^L, \theta^F)}_{\text{indirect gradient via influencing follower}} \tag{2}$$

The first term, the *direct gradient*, captures the steepest direction along which the leader can directly improve its objective. The second term, the *indirect gradient via influencing follower*, captures the indirect strategic effect: it characterizes the direction along which the leader can further improve its objective by influencing the follower's response. In particular, the term $\big(\nabla_{\theta^L}\theta^F_*(\theta^L)\big)$ captures how the follower's optimal decision changes in response to variations in the leader's decision. The Jacobian $\nabla_{\theta^L}\theta^F_*(\theta^L)$ follows from the first-order optimality condition of the follower's maximization problem, $\nabla_{\theta^F} J^F(\theta^L, \theta^F_*) = 0$, which indicates that $\theta^F_*(\theta^L)$ is an implicit function of the leader's variable $\theta^L$. This yields

$$\big(\nabla_{\theta^L}\theta^F_*(\theta^L)\big)^\top = -\nabla_{\theta^L\theta^F} J^F(\theta^L, \theta^F)\big(\nabla^2_{\theta^F} J^F(\theta^L, \theta^F)\big)^{-1} \tag{3}$$

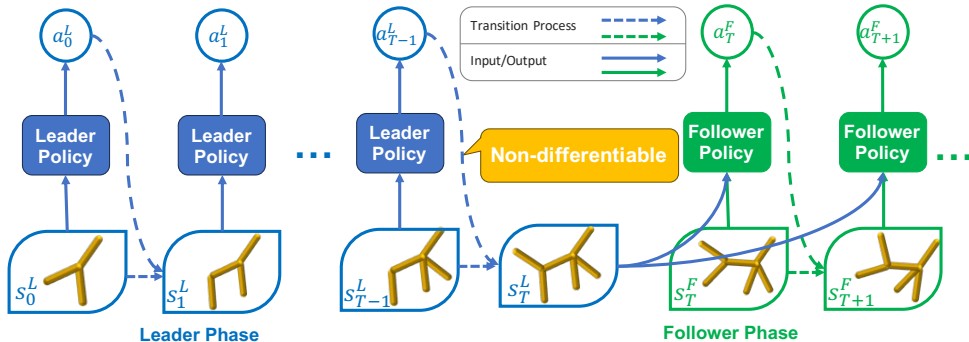

Figure 2: Illustration of the phase-separated Stackelberg Markov Game for morphology–control co-design. In the leader phase (blue part), the agent incrementally edits the morphology via discrete topology-altering actions, producing a terminal morphology $s_T^L$. In the follower phase (green part), the control policy is optimized based on this morphology.

This implicit differentiation framework, often referred to as **Stackelberg implicit differentiation (SID)**, is widely used in Stackelberg reinforcement learning (e.g., Stackelberg DDPG (Yang et al., 2023)), bi-level optimization (Zucchet & Sacramento, 2022), and meta-learning (Pan et al., 2023).

**Morphology–Control Co-Design** Morphology–control co-design aims to jointly optimize a robot's morphology (body structure) and its control policy. The morphology defines the robot's structural and physical properties, such as topology (limb and joint connectivity), geometry (body proportions, limb lengths), and material properties for soft robots. The controller determines the robot's actions based on its body, using motor commands, torque signals, or higher-level behaviors.

The process unfolds in two stages. First, the morphology is constructed incrementally through a sequence of step-by-step editing actions (e.g., adding/removing limbs, adjusting lengths, attaching joints), rather than producing a complete morphology in a single step. This incremental approach is necessary due to the high-dimensional and combinatorial morphology space, which makes one-shot generation intractable. Formally, starting from an initial morphology $s_0^L$, morphology-editing actions $a_t^L$ are applied sequentially until a terminal morphology $s_T^L$ is obtained:

$$a_t^L \sim \pi^L(\cdot \mid s_t^L), \quad s_{t+1}^L \sim \mathcal{P}^L(\cdot \mid s_t^L, a_t^L), \quad t = 0, 1, \cdots, T-1.$$

The transition function $\mathcal{P}^L$ updates the morphology through discrete topology-altering operations, making the morphology dynamics non-differentiable.

Next, the control policy is optimized based on the generated terminal morphology $s_T^L$. The morphology defines the controller's action space (e.g., available joints and actuators), state space (e.g., joint positions, forces, velocities), and underlying dynamics. At each timestep, given a state $s_t^F$, the controller applies a control action $a_t^F$ and transitions to a new state:

$$a_t^F \sim \pi^F(\cdot \mid s_t^F; s_T^L), \quad s_{t+1}^F \sim \mathcal{P}^F(\cdot \mid s_t^F, a_t^F; s_T^L), \quad t = T, T+1, \cdots.$$

The overall process is illustrated in Figure 2.

Existing works (Lu et al., 2025; Yuan et al., 2022) commonly model the co-design problem as a bi-level optimization structure, similar to the formulation in eq. (1), where $\theta^L$ and $\theta^F$ represent the parameters of the morphology policy $\pi_{\theta^L}^L$ and the controller policy $\pi_{\theta^F}^F$, respectively. However, to simplify implementation, these approaches typically adopt a *single-level shared objective*, given by

$$\max_{\theta^L, \theta^F} J^{\text{shared}}(\theta^L, \theta^F) = \mathbb{E}\left[\sum_{t=T}^{\infty} \gamma^{t-T} R^F(s_t^F, a_t^F; s_T^L)\right], \tag{4}$$

where $R^F(s_t^F, a_t^F; s_T^L)$ measures the controller's performance under the specified morphology, such as locomotion speed or task success.

This shared-objective formulation treats the controller parameters $\theta^F$ as jointly optimized with $\theta^L$, ignoring that $\theta_*^F(\theta^L)$ is implicitly determined by the morphology. Consequently, the leader up-

date includes only the direct gradient term and omits the implicit response term, which may steer morphology optimization in a direction misaligned with the controller's best response.

# 4 A STACKELBERG PERSPECTIVE ON CO-DESIGN

In this section, we formalize morphology–control co-design from a Stackelberg game-theoretic perspective. We first introduce asymmetric objectives for the morphology designer and the controller, and then cast their interaction as a leader–follower Stackelberg game.

**Asymmetric Objectives** In contrast to classical approaches that adopt a single shared objective (eq. 4) (Lu et al., 2025), we introduce asymmetric objectives for morphology and control:

$$J^L(\theta^L, \theta^F) = \mathbb{E}\left[\sum_{t=0}^{T-1} \gamma^t R^L(s_t^L, a_t^L) + \sum_{t=T}^{\infty} \gamma^{t-T} R^F(s_t^F, a_t^F; s_T^L)\right] \tag{5}$$

The reward $R^L(s_t^L, a_t^L)$ provides immediate feedback for morphology-editing actions, typically reflecting structural costs such as material usage or design complexity. The morphology objective therefore combines immediate morphology-editing rewards with long-term control performance.

The control objective focuses solely on maximizing its long-term return, conditioned on the terminal morphology induced by $\pi_{\theta^L}^L$:

$$J^F(\theta^L, \theta^F) = \mathbb{E}\left[\sum_{t=T}^{\infty} \gamma^{t-T} R^F(s_t^F, a_t^F; s_T^L)\right] \tag{6}$$

This formulation makes asymmetry explicit: control adapts to a fixed morphology, while morphology optimizes both its structural objectives and downstream control performance.

**Stackelberg Formulation** Although existing work formulates co-design as a bi-level optimization problem, it typically treats the control policy as fixed when optimizing the morphology (see eq. 4) (Lu et al., 2025; Yuan et al., 2022), thereby failing to capture adaptive control dynamics. We revisit this structure from a Stackelberg game-theoretic perspective, where the morphology policy acts as the leader and the control policy acts as the follower. This perspective emphasizes strategic anticipation: the leader commits to a decision while accounting for the follower's rational response.

This perspective naturally gives rise to Stackelberg Implicit Differentiation (SID), leading to different optimization behavior by incorporating the follower's adaptive dynamics into morphology optimization (see eq. 2). However, applying SID to morphology–control co-design is non-trivial due to two intrinsic characteristics: a) a phase-separated interaction structure and b) a non-differentiable leader–follower interface, both of which obstruct gradient propagation across the interface.

*a) Phase-separated interaction.* Unlike standard Stackelberg Markov Games (SMG) in which the leader and follower alternate actions (Li et al., 2020), the interaction in the co-design problem is phase-separated: the leader acts for $T$ steps, after which the follower acts for the remaining horizon. Formally, we define this structure as a *Phase-Separated Stackelberg Markov Game*.

**Definition 1.** *A* Phase-Separated SMG *between a leader policy $\pi_{\theta^L}^L$ and a follower policy $\pi_{\theta^F}^F$, parameterized by $\theta^L$ and $\theta^F$ respectively, is defined as*

$$\mathcal{G} = \left((\mathcal{S}^L, \mathcal{A}^L, \mathcal{P}^L, R^L, \mu^L, T), (\mathcal{S}^F, \mathcal{A}^F, \mathcal{P}^F, R^F, \mu^F), \gamma\right).$$

(i) *The leader's components are given by its state space $\mathcal{S}^L$, action space $\mathcal{A}^L$, transition function $\mathcal{P}^L$, reward function $R^L$, initial state distribution $\mu^L$, and acting horizon $T$.*

(ii) *The leader–follower interaction is phase-separated (i.e., non-alternating): the leader first acts for $T$ steps, producing a terminal state $s_T^L$, after which the follower acts until termination.*

(iii) *The leader and follower interact through the terminal state $s_T^L \in \mathcal{S}^L$, induced by the leader's action sequence under the transition dynamics $\mathcal{P}^L$. The follower acts conditioned on terminal state $s_T^L$, and all its components $(\mathcal{S}^F, \mathcal{A}^F, \mathcal{P}^F, R^F, \mu^F)$ are defined conditionally on $s_T^L$.*

(iv) *The leader aims to solve the Stackelberg objective defined in eq. (1).*

*b) Non-differentiable interfaces.* In morphology–control co-design, the resulting Phase-Separated SMG exhibits an inherently non-differentiable leader–follower interface. Specifically, the leader's transition function $\mathcal{P}^L$ updates the morphology through discrete topology-altering actions, so the terminal state $s_T^L$ linking the leader and follower does not permit direct gradient propagation from the follower to the leader.

**Value Functions**   For subsequent analysis, we introduce value functions induced by the asymmetric objectives. Analogous to standard RL, the leader's Q-function, induced from its objective in eq. (5), is defined as

$$
Q_{\pi^L,\pi^F}^L \left( s_{t'}^L, a_{t'}^L \right) = \mathbb{E} \left[ \sum_{t=t'}^{T-1} \gamma^{t-t'} R^L(s_t^L, a_t^L) + \sum_{t=T}^{\infty} \gamma^{t-t'} R^F(s_t^F, a_t^F; s_T^L); s_t^L = s_{t'}^L, a_t^L = a_{t'}^L, \pi^L, \pi^F \right]
$$

This Q-function captures the leader's long-term return from a given state–action pair, accounting for both its rewards before morphology finalization and the follower's rewards conditioned on the final morphology. From this, the leader's advantage function is defined as $A_{\pi^L,\pi^F}^L \left( s_{t'}^L, a_{t'}^L \right) = Q_{\pi^L,\pi^F}^L \left( s_{t'}^L, a_{t'}^L \right) - \mathbb{E}_{a_{t'}^L \sim \pi^L} \left[ Q_{\pi^L,\pi^F}^L \left( s_{t'}^L, a_{t'}^L \right) \right]$. The advantage function measures how much better (or worse) a specific action $a_{t'}^L$ is compared to the leader's average behavior at state $s_{t'}^L$. A follower's advantage function $A_{\pi^F}^F \left( s_{t'}^F, a_{t'}^F; s_T^L \right)$ can be defined analogously from its objective in eq. (6).

## 5   METHOD

In this section, we present how to realize Stackelberg Implicit Differentiation (SID; see Section 3) within the phase-separated Stackelberg Markov Game (SMG) framework introduced in Section 4. This enables us to account for the follower's adaptive response during morphology optimization, which is typically ignored in prior approaches (Lu et al., 2025; Yuan et al., 2022).

A direct way to implement SID is via backpropagation-through-interface, which propagates gradients from the follower to the leader through the leader–follower interface, as in Stackelberg MADDPG (Yang et al., 2023). However, as highlighted in Section 4, the phase-separated SMG formulation of the co-design problem exhibits a non-differentiable leader–follower interface and a temporally separated interaction structure, both of which complicate gradient propagation across the interface. These challenges require new derivations of the Stackelberg gradients, as presented below.

### 5.1   STACKELBERG POLICY GRADIENT

We now introduce Stackelberg implicit differentiation into our phase-separated Stackelberg Markov Game defined in Definition 1. Since this formulation departs from the classical SMG, we develop new derivations for all gradient components in eqs. (2) and (3). We present each term in turn.

**Cross-Derivative** $\nabla_{\theta^L \theta^F} J^F(\theta^L, \theta^F)$ (see eq. (3)).   This is the most challenging term. Unlike classical SMGs where the follower directly takes the leader's action as input, in our setting the interface is the terminal state $s_T^L$, generated through the transition $\mathcal{P}^L$. Backpropagation-through-interface methods (e.g., Stackelberg MADDPG) are infeasible here, since reaching $\theta^L$ would require differentiating through the non-differentiable transition $\mathcal{P}^L$. Instead, we derive the cross-derivative using the log-derivative technique, analogous to the stochastic policy gradient (Sutton, 1984; Williams, 1992; Sutton et al., 1999), which bypasses the transition's non-differentiability while relying only on sampled trajectories. Let $(\theta_o^L, \theta_o^F)$ denote the parameters of the behavior policies used for collecting data. Formally, we obtain the following theorem.

**Theorem 1.** *We define the surrogate*

$$
\mathcal{L}_{L,F}^F \left( \theta^L, \theta^F; \theta_o^L, \theta_o^F \right) = c \mathbb{E} \left[ \frac{\pi_{\theta^L}^L \left( a^L | s^L \right)}{\pi_{\theta_o^L}^L \left( a^L | s^L \right)} \left[ \gamma^T \mathbb{E} \left[ \frac{\pi_{\theta^F}^F \left( a^F | s^F; s_T^L \right)}{\pi_{\theta_o^F}^F \left( a^F | s^F; s_T^L \right)} A_{\pi_{\theta_o^F}^F}^F \left( s^F, a^F; s_T^L \right) \right] \right] \right] \quad (7)
$$

*Then, we have* $\nabla_{\theta^L \theta^F} J^F(\theta^L, \theta^F)|_{\theta^L = \theta_o^L, \theta^F = \theta_o^F} = \nabla_{\theta^L \theta^F} \mathcal{L}_{L,F}^F \left( \theta^L, \theta^F; \theta_o^L, \theta_o^F \right)|_{\theta^L = \theta_o^L, \theta^F = \theta_o^F}.$

In eq. (7), the outer expectation is taken over $s^L \sim d^L_{\theta^L_o}$, $a^L \sim \pi^L_{\theta^L_o}$, $s^L_T \sim d^{L,T}_{\theta^L_o}$, where $d^{L,t}_{\theta^L_o}(s^L) = P(s^L_t = s^L; \pi^L_{\theta^L_o})$ is the visitation distribution probability of leader policy at step $t$, and $d^L_{\theta^L_o}(s^L) \triangleq 1/T \sum_t d^{L,t}_{\theta^L_o}(s^L)$. The inner expectation is taken over $s^F \sim d^F_{\theta^F_o}(\cdot; s^L_T)$, $a^F \sim \pi^F_{\theta^F_o}(\cdot; s^L_T)$, where $d^F_{\theta^F_o}$ denotes the follower's visitation distribution. The constant $c = T/(1-\gamma)$ normalizes the distribution, and its effect can be absorbed by the learning rate in practice. Proofs of this and subsequent theorems are provided in Appendix B. This theorem shows that the cross-derivative can be expressed as an expectation involving likelihood-ratio (importance-weighted) advantage estimators, thereby extending the classical policy gradient theorem to capture leader-follower coupling in our phase-separated SMG.

**First-Order Derivatives** $\nabla_{\theta^L} J^L(\theta^L, \theta^F)$ **and** $\nabla_{\theta^F} J^L(\theta^L, \theta^F)$ (see eq. 2). These first-order terms are relatively straightforward, as they follow the same structure as the policy gradient theorem (Sutton, 1984; Williams, 1992; Sutton et al., 1999). They quantify how the leader's objective changes with respect to its own parameters (leader's direct gradient) and w.r.t. the follower's parameters. Both can be expressed using advantage functions under importance weighting, as follows.

**Proposition 1.** *We have*

$$\nabla_{\theta^L} J^L(\theta^L, \theta^F)|_{\theta^L=\theta^L_o, \theta^F=\theta^F_o} = \nabla_{\theta_L} \mathcal{L}^L_L(\theta^L, \theta^F; \theta^L_o, \theta^F_o)|_{\theta^L=\theta^L_o, \theta^F=\theta^F_o}$$
$$\nabla_{\theta^F} J^L(\theta^L, \theta^F)|_{\theta^L=\theta^L_o, \theta^F=\theta^F_o} = \nabla_{\theta_F} \mathcal{L}^L_F(\theta^L, \theta^F; \theta^L_o, \theta^F_o)|_{\theta^L=\theta^L_o, \theta^F=\theta^F_o}$$

*where* $\mathcal{L}^L_L(\theta^L, \theta^F; \theta^L_o, \theta^F_o) = \mathbb{E}\left[\frac{\pi^L_{\theta^L}(a^L|s^L)}{\pi^L_{\theta^L_o}(a^L|s^L)} A^L_{\pi^L_{\theta^L_o}, \pi^F_{\theta^F_o}}(s^L, a^L)\right]$

$$\mathcal{L}^L_F(\theta^L, \theta^F; \theta^L_o, \theta^F_o) = \mathbb{E}\left[\gamma^T \frac{\pi^F_{\theta^F}(a^F|s^F; s^L_T)}{\pi^F_{\theta^F_o}(a^F|s^F; s^L_T)} \pi^F_{\theta^F_o}(s^F, a^F; s^L_T)\right] \tag{8}$$

**Inverse of Second-Order Derivative (Hessian)** $\left(\nabla^2_{\theta^F} J^F(\theta^L, \theta^F)\right)^{-1}$ (see eq. 3). This last component involves the inverse Hessian. Although the Hessian can be computed from the derived loss function (see Appendix Proposition 2), the Hessian is typically indefinite due to the advantage term, making its inversion unstable. A standard remedy is to approximate it by the Fisher information matrix, $\mathcal{F}(\theta^F) = \mathbb{E}\left[\nabla_{\theta^F} \log \pi^F_{\theta^F}(a^F \mid s^F; s^L_T) \nabla_{\theta^F} \log \pi^F_{\theta^F}(a^F \mid s^F; s^L_T)^\top\right]$, which is positive semi-definite and can be estimated via the KL divergence between policies:

$$\mathcal{F}(\theta^F) = \nabla^2_{\theta^F} \mathcal{L}^F_{\text{KL}}(\theta^L, \theta^F; \theta^L_o, \theta^F_o) = \nabla^2_{\theta^F} \mathbb{E}\left[\text{KL}\left(\pi^F_{\theta^F_o}(\cdot \mid s^F; s^L_T) \| \pi^F_{\theta^F}(\cdot \mid s^F; s^L_T)\right)\right], \tag{9}$$

This natural-gradient approximation, used in methods such as natural policy gradient and trust region policy optimization (Kakade, 2001; Schulman et al., 2015), avoids indefiniteness and improves stability. Further stability is obtained by regularizing the Hessian with a small multiple of the identity $\left(\nabla^2_{\theta^F} \mathcal{L}^F_{\text{KL}} + \lambda I\right)^{-1}$ with $\lambda > 0$, which has been shown to interpolate between the standard policy gradient (when $\lambda \to \infty$) and the Stackelberg gradient (when $\lambda \to 0$) (Yang et al., 2023).

## 5.2 ALGORITHMS

Based on the surrogate functions in Eqs. (7) to (9), we compute the leader's Stackelberg gradient in Eq. (2). Since these surrogates are locally equivalent to the true Stackelberg gradients, we adopt the likelihood-ratio clipping technique from PPO (Schulman et al., 2017) to constrain policy divergence and ensure stable optimization. Note that this application is not a simple reuse of PPO clipping. Rather, it is grounded in our local-approximation theory on the newly derived Stackelberg surrogate (see Theorem 1). Moreover, the expectation terms are estimated from sampled trajectories. This yields sample-based surrogates with PPO clipping, denoted by $\widehat{\mathcal{L}}$, and the corresponding estimation of the leader's Stackelberg gradient can be expressed as

$$\nabla_{\theta^L} \widehat{J}^L(\theta_L, \theta^*_F(\theta_L)) = \nabla_{\theta_L} \widehat{\mathcal{L}}^L_L - \overbrace{\nabla_{\theta_L \theta_F} \widehat{\mathcal{L}}^F_{L,F} \underbrace{\left(\nabla^2_{\theta_F} \widehat{\mathcal{L}}^F_{\text{KL}} + \lambda I\right)^{-1} \nabla_{\theta_F} \widehat{\mathcal{L}}^L_F}_{\text{step 2}}}^{\text{step 1}} \tag{10}$$

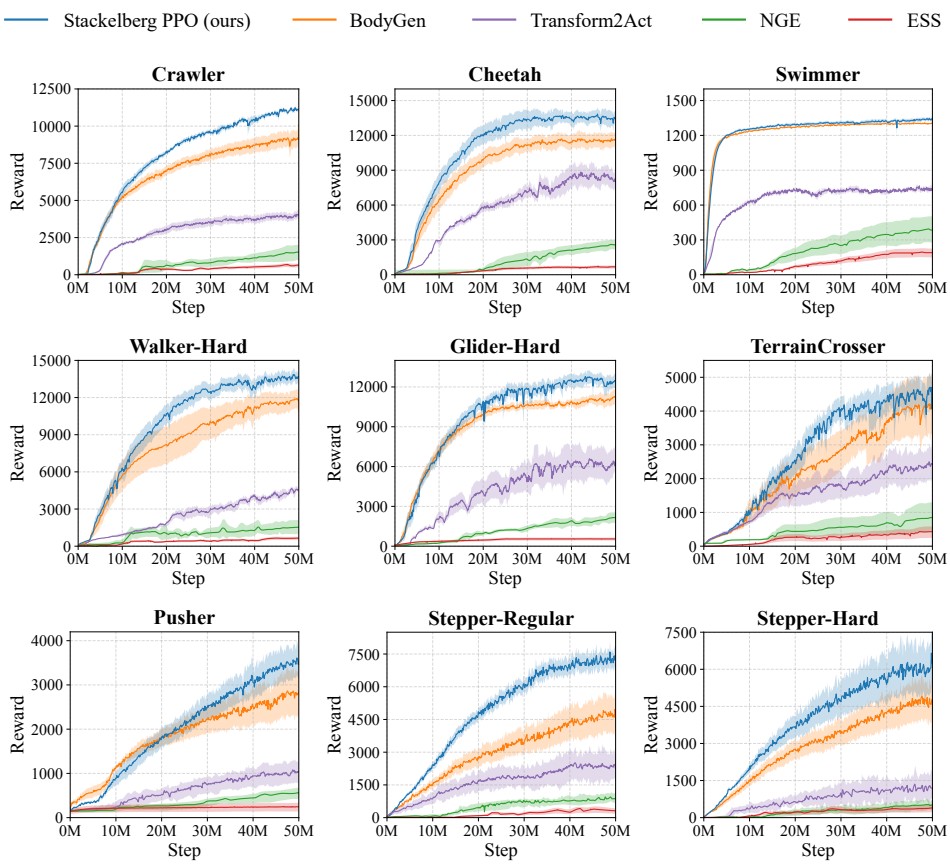

Figure 3: Performance curve with respect to the number of follower steps during training. Shaded regions denote standard error across seven random seeds.

We refer to this overall procedure as *Stackelberg PPO*, which integrates PPO-style clipping into the Stackelberg gradient computation. We first compute *step 1* in the above equation, which can be efficiently implemented using the conjugate gradient method. Conjugate gradient only requires Hessian-vector products, which can be obtained without explicitly constructing the Hessian via Pearlmutter's method (Pearlmutter, 1994; Møller, 1993): $\nabla_\theta^2 \mathcal{L}(\theta) v = \nabla_\theta (\nabla_\theta^\top \mathcal{L}(\theta) v)$. We then compute *step 2*, which in turn only requires Jacobian-vector products. These can likewise be computed efficiently without explicitly forming the full Jacobian by using the Jacobian-vector product operation provided by automatic differentiation frameworks.

## 6 EXPERIMENTS

Our goal is to test whether leveraging Stackelberg implicit differentiation to regularize the leader's gradient can improve sample efficiency and final performance.

All experiments are conducted on MuJoCo-based morphology–control co-design tasks. We adopt benchmarks from prior work, including three flat-terrain tasks (`Crawler`, `Cheetah`, `Swimmer`, `Glider`, `Walker`) and one complex-terrain task (`TerrainCrosser`) (Lu et al., 2025). To further evaluate performance under more challenging conditions, we introduce two new tasks, `Stepper-Regular` and `Stepper-Hard`, where the agent must climb stair-like structures. These tasks require morphologies capable of effective climbing in addition to robust control. To test generality beyond locomotion, we also include a contact-rich 3D manipulation task, `Pusher`, designed to evaluate whether co-design methods can evolve structures aligned with manipulation objectives. Additional results on other tasks are provided in Appendix C due to space constraints. In all environments, morphologies are represented as tree structures with constraints on depth, branch-

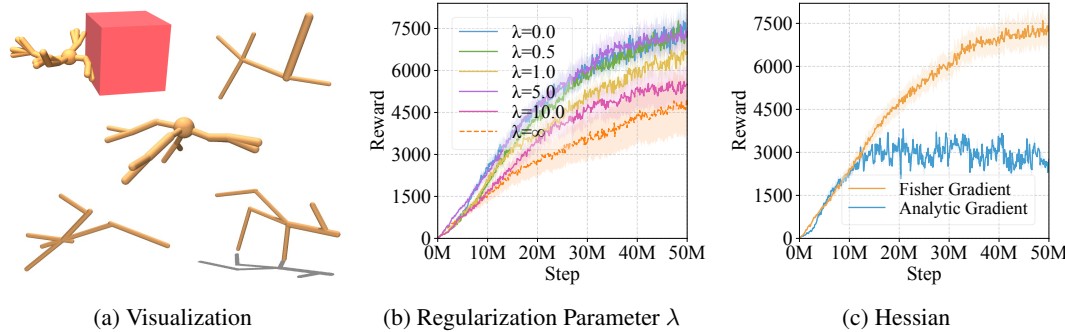

(a) Visualization      (b) Regularization Parameter $\lambda$      (c) Hessian

Figure 4: (a) Evolved morphologies. Ablation studies on (b) $\lambda$ sweep from 0 to $\infty$ and (c) Fisher information matrix on/off, both evaluated on Stepper-Regular task.

ing factor, and joint degrees of freedom. While structural complexity and terrain difficulty vary, the reward function consistently emphasizes forward velocity, ensuring fair comparisons of how different methods balance morphology and control. Each algorithm is evaluated with seven random seeds per task. All reported learning curves show mean values with shaded areas representing standard deviations. Further details and visualization of the environments are provided in Appendix C.

## 6.1 COMPARISON WITH BASELINES

We implement our Stackelberg PPO on top of *BodyGen* (Lu et al., 2025), a PPO-based framework that employs transformer-based co-design with graph-aware positional encodings, optimizing morphology and control independently under shared rewards. BodyGen serves as our primary baseline, with the only modification being the use of Stackelberg policy gradients. Implementation details are provided in Appendix C. In addition to BodyGen, we compare Stackelberg PPO against several advanced co-design methods:

- *Evolutionary Structure Search (ESS)* (Sims, 1994): A canonical evolutionary-algorithm approach to robot design, where candidate morphologies are scored by handcrafted fitness functions. Here we instead use a lightweight RL-based training loop for principled evaluation.

- *Neural Graph Evolution (NGE)* (Wang et al., 2019): Evolutionary search over graph-structured morphologies with GNN controllers, each generation training the inherited parent controller.

- *Transform2Act* (Yuan et al., 2022): Concurrent RL co-design using separate GNNs for morphology and control within unified PPO training, with joint-specific MLP heads for universal control.

Figure 3 presents the learning curves across all environments. Stackelberg PPO consistently achieves the best performance, yielding an average **+20.66%** improvement over the strongest baseline. Compared to evolutionary approaches (ESS, NGE), it attains substantially higher sample efficiency by avoiding the costly rollouts required to evaluate each morphological candidate. Relative to the vanilla gradient method without Stackelberg differentiation (BodyGen), Stackelberg PPO achieves superior results in both sample efficiency and final performance. The advantage is most evident on challenging 3D tasks with large design spaces (`Crawler`, `Stepper-Regular`, `Stepper-Hard`, `Pusher`), where our method delivers an average **+32.02%** improvement. Figure 4(a) showcases examples of the evolved creatures generated by our method. Additional morphology examples and evolution processes are provided in the appendix E.1 and E.5.

## 6.2 ABLATION STUDIES

We conduct ablation studies to validate key components of Stackelberg PPO, including (1) the regularization parameter $\lambda$ that controls gradient interpolation (eq. 10); and (2) the Fisher gradient approximation of the Hessian for stability (eq. 9).

**Regularization Parameter $\lambda$ (eq. 10).** The parameter $\lambda$ interpolates between pure Stackelberg gradients and standard policy gradients. We evaluate Stackelberg PPO on the `Stepper-Regular`

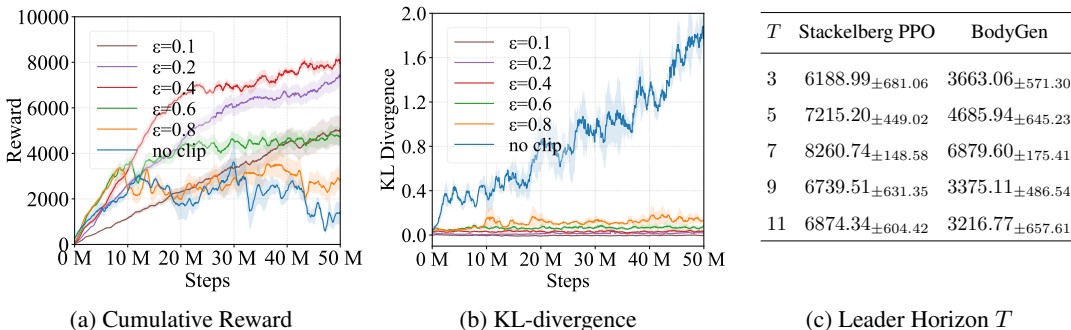

Figure 5: (a) Reward curves and (b) KL-divergence traces for different clipping thresholds $\epsilon$. (c) Performance comparison under varying leader horizons $T$. All evaluated on Stepper-Regular.

environment with $\lambda \in \{0.0, 0.5, 1.0, 5.0, 10.0, \infty\}$, where $\lambda = 0$ corresponds to *no regularization* and $\lambda = \infty$ reduces to the vanilla gradient without Stackelberg differentiation. Figure 4(b) shows robust performance for $\lambda \in [0.5, 10]$, with degradation only at the extremes ($\lambda = 0$ or $\infty$). This highlights both the robustness of the method to $\lambda$ values and the necessity of regularization.

**Hessian Computation (eq. 9).** We compare our Fisher approximation with direct analytic second-order gradients (eq. 11). As shown in Figure 4(c), the Fisher approximation achieves stable learning with nearly twice the performance of the analytic gradient (6000 vs 2500). This improvement arises from the positive semi-definiteness of the Fisher matrix, which avoids the numerical instabilities caused by the indefinite raw Hessian.

**Sensitivity to PPO Clipping Threshold $\epsilon$.** We evaluate the sensitivity of Stackelberg PPO to the clipping parameter $\epsilon$ by sweeping over multiple thresholds and measuring its effect on task performance and KL-divergence stability. Figure 5(a) and (b) shows that moderate clipping (e.g., $\epsilon \leq 0.4$) yields stable learning with low KL divergence, while removing clipping causes rapid KL growth and clear performance degradation. Full quantitative results are reported in Appendix E.3.

**Leader Horizon $T$ (eq. 7).** We examine how the leader horizon $T$ affects structural optimization. As shown in Figure 5(c), larger horizons generally improve performance by enabling richer morphology edits, while overly large values (e.g., $T = 11$) become harder to optimize and cause mild degradation—yet still outperform very small horizons such as $T = 3$. Importantly, increasing $T$ does *not* introduce higher variance in the leader-gradient update of eq. 7 relative to BodyGen baseline, confirming that the Stackelberg update remains stable across a wide range of horizon lengths.

## 7 CONCLUSIONS

We introduced *Stackelberg Proximal Policy Optimization (Stackelberg PPO)*, a reinforcement learning framework grounded in the **Stackelberg game** paradigm, which explicitly captures the leader–follower coupling between high-level design decisions and adaptive control responses. While this formulation is general, we instantiate it in the context of morphology–control co-design, where the leader specifies the body structure and the follower adapts the control policy. Instead of treating design and control as independent, Stackelberg PPO exploits the leader–follower coupling to anticipate how the follower will adapt, enabling the leader to update its policy toward morphologies that are more compatible with downstream control. Experiments demonstrate that this coupling yields superior performance and stability over standard PPO, particularly on complex locomotion tasks where tight coordination between morphology and control is essential.

Despite these promising results, several avenues remain for future work. A key direction is sim-to-real transfer, which remains challenging due to unmodeled hardware constraints and material dynamics. Bridging this gap could enable the real-world deployment of self-evolving robotic systems. We further envision advances in this area leading to truly adaptive artificial life forms capable of self-directed evolution, reshaping our understanding of intelligence, embodiment, and the boundary between designed and evolved systems.

## REPRODUCIBILITY STATEMENT

The demonstrations and open-source code are available at: https://yanningdai.github.io/stackelberg-ppo-co-design. The computational requirements, hyperparameters, and key implementation details are provided in Appendix D.

## ACKNOWLEDGEMENTS

We gratefully acknowledge the insightful comments from the ICLR 2026 reviewers. The research reported in this publication was supported by funding from King Abdullah University of Science and Technology (KAUST) - Center of Excellence for Generative AI, under award number 5940. This work was additionally supported by the European Research Council (ERC, Advanced Grant Number 742870). For computer time, this research used Ibex managed by the Supercomputing Core Laboratory at King Abdullah University of Science and Technology (KAUST) in Saudi Arabia.

## ETHICS STATEMENT

As fundamental AI research and to the best of the authors' knowledge, there are no clear ethical risks associated with this work beyond the risks already posed by prior work.

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

## A  LLM USAGE STATEMENT

We used LLMs for drafting and refining text extensively throughout the paper. LLMs were not used to develop algorithms, provide theoretical results, run experiments, or contribute in any other way to the work beyond the aforementioned writing help.

## B  THEORETICAL ANALYSIS

In this section, we provide the theoretical foundations of our approach. We first present the trajectory factorization in the proposed phase-separated Stackelberg Markov Game, which serves as the basis for proving all the theorems.

Given the trajectory $\tau \triangleq \left( \{s_t^L, a_t^L\}_{t=0}^{T-1}, s_T^L, \{s_t^F, a_t^F\}_{t=T}^{\infty} \right)$, the trajectory distribution naturally factorizes into two phases:

$$
\begin{aligned}
& P\left(\tau ; \pi_{\theta^L}^L, \pi_{\theta^F}^F\right) \\
= & \mu^L\left(s_0^L\right) \prod_{t=0}^{T-1} \pi_{\theta^L}^L\left(a_t^L | s_t^L\right) \mathcal{P}^L\left(s_{t+1}^L | s_t^L, a_t^L\right) \mu^F\left(s_0^F ; s_T^L\right) \prod_{t=T}^{\infty} \pi_{\theta^F}^F\left(a_t^F | s_t^F ; s_T^L\right) \mathcal{P}\left(a_{s_{t+1}}^F | s_t^F, a_t^F ; s_T^L\right)
\end{aligned}
$$

*Proof of Theorem 1.*  Based on policy gradient theorem (Sutton, 1984; Williams, 1992; Sutton et al., 1999), we have

$$
\nabla_{\theta^F} J^F\left(\theta^L, \theta^F\right) = \int P\left(\tau ; \theta^L, \theta^F\right) \gamma^T \sum_{t=T}^{\infty} \nabla_{\theta^F} \log \pi_{\theta^F}^F\left(a_t^F | s_t^F ; s_T^L\right) A_t^F \, d\tau
$$

Differentiating both sides with respect to $\theta^L$ yields the cross derivative:

$$
\begin{aligned}
& \nabla_{\theta^L, \theta^F}^2 J^F\left(\theta^L, \theta^F\right) \\
= & \nabla_{\theta^L}\left(\nabla_{\theta^F} J^F\left(\theta^L, \theta^F\right)\right) \\
= & \nabla_{\theta^L} \int P\left(\tau ; \theta^L, \theta^F\right)\left[\gamma^T \sum_{t=T}^{\infty} \nabla_{\theta^F} \log \pi_{\theta^F}^F\left(a_t^F | s_t^F ; s_T^L\right) A_t^F\right] d\tau \\
= & \int \nabla_{\theta^L} P\left(\tau ; \theta^L, \theta^F\right)\left[\gamma^T \sum_{t=T}^{\infty} \nabla_{\theta^F} \log \pi_{\theta^F}^F\left(a_t^F | s_t^F ; s_T^L\right) A_t^F\right]^{\top} d\tau \\
= & \int P\left(\tau ; \theta^L, \theta^F\right) \nabla_{\theta^L} \log P\left(\tau ; \theta^L, \theta^F\right)\left[\gamma^T \sum_{t=T}^{\infty} \nabla_{\theta^F} \log \pi_{\theta^F}^F\left(a_t^F | s_t^F ; s_T^L\right) A_t^F\right]^{\top} d\tau \\
= & \int P\left(\tau ; \theta^L, \theta^F\right)\left[\sum_{t=0}^{T-1} \nabla_{\theta^L} \log \pi_{\theta^L}^L\left(a_t^L | s_t^L\right)\right]\left[\gamma^T \sum_{t=T}^{\infty} \nabla_{\theta^F} \log \pi_{\theta^F}^F\left(a_t^F | s_t^F ; s_T^L\right) A_t^F\right]^{\top} d\tau \\
= & \nabla_{\theta^L, \theta^F} \int P\left(\tau ; \theta^L, \theta^F\right)\left[\sum_{t=0}^{T-1} \log \pi_{\theta^L}^L\left(a_t^L | s_t^L\right)\right]\left[\gamma^T \sum_{t=T}^{\infty} \log \pi_{\theta^F}^F\left(a_t^F | s_t^F ; s_T^L\right) A_t^F\right]^{\top} d\tau
\end{aligned}
$$

Evaluating this identity at the reference parameters $(\theta^L, \theta^F) = (\theta_o^L, \theta_o^F)$ gives

$$\nabla^2_{\theta^L,\theta^F} J^F\left(\theta^L,\theta^F\right)|_{\theta^L=\theta^L_o,\theta^F=\theta^F_o}$$

$$=\nabla^2_{\theta^L,\theta^F}\int P\left(\tau;\theta^L_o,\theta^F_o\right)\left[\sum_{t=0}^{T-1}\frac{\pi^L_{\theta^L}\left(a^L_t|s^L_t\right)}{\pi^L_{\theta^L_o}\left(a^L_t|s^L_t\right)}\right]\left[\gamma^T\sum_{t=T}^{\infty}\frac{\pi^F_{\theta^F}\left(a^F_t|s^F_t;s^L_T\right)}{\pi^F_{\theta^F_o}\left(a^F_t|s^F_t;s^L_T\right)}A^F_{\pi^F_{\theta^F}}\left(s^F_t,a^F_t;s^L_T\right)\right]d\tau$$

$$=\nabla^2_{\theta^L,\theta^F}\int P\left(\tau^L_{0:T};\theta^L_o,\theta^F_o\right)\left[\sum_{t=0}^{T-1}\frac{\pi^L_{\theta^L}\left(a^L_t|s^L_t\right)}{\pi^L_{\theta^L_o}\left(a^L_t|s^L_t\right)}\right]$$

$$\int P\left(\tau^F_{T:\infty};\theta^L_o,\theta^F_o\right)\left[\gamma^T\sum_{t=T}^{\infty}\frac{\pi^F_{\theta^F}\left(a^F_{t'}|s^F_{t'};s^L_T\right)}{\pi^F_{\theta^F_o}\left(a^F_{t'}|s^F_{t'};s^L_T\right)}A^F_{\pi^F_{\theta^F}}\left(s^F_t,a^F_t;s^L_T\right)\right]d\tau$$

$$=\nabla^2_{\theta^L,\theta^F}\sum_{t=0}^{T-1}\int P\left(\tau^L_{0:T};\theta^L_o,\theta^F_o\right)\left[\frac{\pi^L_{\theta^L}\left(a^L_t|s^L_t\right)}{\pi^L_{\theta^L_o}\left(a^L_t|s^L_t\right)}\right]$$

$$\sum_{t'=T}^{\infty}\int P\left(\tau^F_{T:\infty};\theta^L_o,\theta^F_o,s^L_T\right)\left[\gamma^T\frac{\pi^F_{\theta^F}\left(a^F_{t'}|s^F_{t'};s^L_T\right)}{\pi^F_{\theta^F_o}\left(a^F_{t'}|s^F_{t'};s^L_T\right)}A^F_{\pi^F_{\theta^F}}\left(s^F_{t'},a^F_{t'};s^L_T\right)\right]d\tau$$

$$=\nabla^2_{\theta^L,\theta^F}\int d^L_{\theta^L_o}(s^L)\pi^L_{\theta^L_o}\left(a^L|s^L\right)d^{L,T}_{\theta^L_o}(s^L_T)\left[\frac{\pi^L_{\theta^L}\left(a^L|s^L\right)}{\pi^L_{\theta^L_o}\left(a^L|s^L\right)}\right]$$

$$\int d^F_{\theta^F_o}(s^F;s^L_T)\pi^F_{\theta^F_o}\left(a^F|s^F;s^L_T\right)\left[\gamma^T\frac{\pi^F_{\theta^F}\left(a^F|s^F;s^L_T\right)}{\pi^F_{\theta^F_o}\left(a^F|s^F;s^L_T\right)}A^F_{\pi^F_{\theta^F}}\left(s^F,a^F;s^L_T\right)\right]d\tau$$

$$=\nabla_{\theta^L,\theta^F}\mathcal{L}^F_{L,F}\left(\theta^L,\theta^F;\theta^L_o,\theta^F_o\right)|_{\theta^L=\theta^L_o,\theta^F=\theta^F_o}$$

$\square$

*Proof of Proposition 1.* The result follows directly by applying the likelihood-ratio trick in the same way as the standard proof of the policy gradient theorem (Sutton, 1984; Williams, 1992; Sutton et al., 1999).

$\square$

**Proposition 2.** *We have*

$$\nabla^2_{\theta^F}J^F\left(\theta^L,\theta^F\right)=\nabla^2_{\theta_F}E_{\pi^L_{\theta^L},\pi^F_{\theta^F}}\left[\sum_{t=T}^{\infty}\log\pi^F_{\theta^F}\left(a^F_t|s^F_t;s^L_T\right)A^F_t\right]$$

$$+E_{\pi^L_{\theta^L_o},\pi^F_{\theta^F_o}}\left[\nabla_{\theta^F}\left(\sum_{t=T}^{\infty}\log\pi^F_{\theta^F}\left(a^F_t|s^F_t;s^L_T\right)\right)\nabla_{\theta^F}\left(\sum_{t=T}^{\infty}\log\pi^F_{\theta^F}\left(a^F_t|s^F_t;s^L_T\right)A^F_t\right)^{\top}\right] \quad (11)$$

*Proof of Proposition 2.* Based on policy gradient theorem (Sutton, 1984; Williams, 1992; Sutton et al., 1999), we have

$$\nabla_{\theta^F}J^F\left(\theta^L,\theta^F\right)=\int P\left(\tau;\theta^L,\theta^F\right)\nabla_{\theta_F}\sum_{t=T}^{\infty}\log\pi^F_{\theta^F}\left(a^F_t|s^F_t;s^L_T\right)A^F_t d\tau$$

Differentiating this expression again with respect to $\theta^F$, we obtain

$$\nabla^2_{\theta_F} J_F(\theta_L, \theta_F)$$

$$= \nabla_{\theta_F} \int P\left(\tau; \theta^L, \theta^F\right) \nabla_{\theta_F} \sum_{t=T}^{\infty} \log \pi^F_{\theta^F}\left(a^F_t | s^F_t; s^L_T\right) A^F_t \, d\tau$$

$$= \int P\left(\tau; \theta^L, \theta^F\right) \nabla^2_{\theta_F} \sum_{t=T}^{\infty} \log \pi^F_{\theta^F}\left(a^F_t | s^F_t; s^L_T\right) A^F_t \, d\tau$$

$$+ \int P\left(\tau; \theta^L, \theta^F\right) \nabla_{\theta^F} \log P\left(\tau; \theta^L, \theta^F\right) \left(\nabla_{\theta^F} \sum_{t=T}^{\infty} \log \pi^F_{\theta^F}\left(a^F_t | s^F_t; s^L_T\right)\right)^{\top} d\tau$$

$$= \int P\left(\tau; \theta^L, \theta^F\right) \nabla^2_{\theta_F} \sum_{t=T}^{\infty} \log \pi^F_{\theta^F}\left(a^F_t | s^F_t; s^L_T\right) A^F_t \, d\tau$$

$$+ \int P\left(\tau; \theta^L, \theta^F\right) \left(\nabla_{\theta^F} \sum_{t=T}^{\infty} \log \pi^F_{\theta^F}\left(a^F_t | s^F_t; s^L_T\right)\right) \left(\nabla_{\theta^F} \sum_{t=T}^{\infty} \log \pi^F_{\theta^F}\left(a^F_t | s^F_t; s^L_T\right) A^F_t\right)^{\top} d\tau$$

$$\square$$

## C  ENVIRONMENT DETAILS

In this section, we provide comprehensive details about the nine environments employed in our experimental evaluation. To ensure fair comparison with existing methods, we adopt all 6 environments from the BodyGen framework: **Crawler**, **Cheetah**, **Glider**, **Walker**, **Swimmer**, and **TerrainCrosser**. Additionally, we introduce three novel environments designed to evaluate different aspects of our algorithm: **Stepper-Regular** and **Stepper-Hard** feature complex topographical structures to test robustness in challenging terrain, while **Pusher** evaluates manipulation capabilities. These additional environments are specifically crafted to assess the robustness and adaptability of our algorithm under more challenging conditions, thereby providing a more rigorous assessment of the proposed method's capabilities. Figure 6 provides visualizations of all nine environments.

Each agent undergoes dynamic morphological evolution through topological and attribute modifications during training. The observation includes the root body's spatial position and velocity, all joints' angular positions and velocities, and motor gear parameters. All joints use hinge connections enabling single-axis rotation. Joint attributes encompass bone vectors, sizes, and motor gear values. The action space consists of one-dimensional control signals applied to each joint's motor.

**Crawler** operates in a 3D environment, where agents exhibit quadrupedal crawling locomotion. The initial morphology consists of a central root node with four limb branches extending outward. The body tree is constrained to a maximum depth of 4 levels, with each non-root node supporting at most 2 child limbs. Episodes are terminated when the agent's body height exceeds 2.0 units to prevent unrealistic vertical extensions. The reward function encourages forward movement while penalizing excessive control effort:

$$r_t = \frac{x_{t+1} - x_t}{\tau} - w \cdot \frac{1}{N} \sum_{j \in \mathcal{J}_t} ||\mathbf{u}_j^t||^2 \tag{12}$$

where $x_t$ denotes the agent's forward position at timestep $t$, $\mathbf{u}_j^t$ represents the effective control input applied to joint (i.e., the raw action scaled by the joint's gear ratio) applied to joint $j$ at time $t$, $w = 0.0001$ is the control regularization coefficient, $N$ is the total number of joints, and $\tau = 0.04$.

**Cheetah** features 2D locomotion focused on fast running gaits. The agent begins with an initial design comprising a root body connected to one primary limb segment. The morphological search allows a maximum tree depth of 4 with up to 3 child limbs per node. The root body's angular orientation is constrained within 20 degrees to maintain stable running posture. Episode termination occurs when body height falls below 0.7 or exceeds 2.0 units. The reward follows the velocity-based formulation:

$$r_t = \frac{x_{t+1} - x_t}{\tau} \tag{13}$$

where $\tau = 0.008$.

**Glider and Walker** enable 2D aerial and terrestrial locomotion. Both environments share the same base morphology: agents start from an initial configuration with three limb segments attached to a central root, each limb node can support up to three children, and joints can oscillate within a $60°$ range to accommodate wide-range motion. The reward structure follows eq. 13, emphasizing forward displacement. The two tasks differ only in their allowable morphology depth: Glider restricts the body tree to a maximum depth of 3, while Walker permits up to 4 levels.

**Swimmer** enables undulatory, snake-like locomotion in a 2D aquatic environment. The agent evolves in water with a viscosity coefficient of $vis = 0.1$. The initial morphology consists of a root body connected to a single limb segment, and each limb node may support up to three child segments, enabling flexible articulated structures suited for wave-based propulsion. This task serves as a lightweight validation environment, and thus imposes no early-termination conditions such as height limits or joint-rotation thresholds. The reward structure follows eq. 13, emphasizing forward displacement under hydrodynamic resistance.

**TerrainCrosser** presents a challenging 2D terrain navigation task using the Cheetah agent configuration. The environment features fixed terrain heights with maximum elevation differences of $z_{max} = 0.5$. Agents must traverse gaps generated from single-channel height maps. Height constraints maintain agent stability between 0.7 and 2.0 units, with violations leading to episode termination. The reward function prioritizes forward progress as defined in eq. 13.

**Stepper-Regular and Stepper-Hard** introduce challenging staircase navigation tasks that test agents' morphological adaptation capabilities for vertical terrain traversal. Both environments utilize the Crawler agent configuration in a 3D setting. Stepper-Regular features stairs with step width of 1.0 units and height of 0.4 units; Stepper-Hard increases the difficulty by elevating step height to 0.8 units while maintaining the same width. Unlike the standard Crawler environment, height termination constraints are removed to allow full exploration of vertical climbing capabilities. The reward function follows eq. 12, focusing solely on forward progression, thereby maintaining reward consistency across environments.

**Pusher** is a challenging 3D manipulation task designed to evaluate whether the co-design system can generate morphologies and control strategies that effectively interact with external objects. This environment reuses the Crawler agent configuration in a 3D setting. A rigid cube of side length 1.0 m is placed in front of the agent and constrained to move in the horizontal $(x, y)$ plane. The observation space augments the agent state with the 3D relative position between the agent's root body and the cube. The reward encourages forward displacement of the cube, penalizes lateral motion, and provides an auxiliary shaping term based on the proximity between the agent and the cube. A control-effort penalty identical to eq. 12 is applied. Formally, the reward is

$$r_t = \frac{x_{t+1}^{\text{cube}} - x_t^{\text{cube}}}{\tau} - \kappa \cdot \frac{\left|y_{t+1}^{\text{cube}} - y_t^{\text{cube}}\right|}{\tau} + \frac{1}{1 + \left|\boldsymbol{p}_t^{\text{cube}} - \boldsymbol{p}_t^{\text{root}}\right|} - w \cdot \frac{1}{N} \sum_{j \in \mathcal{J}_t} ||\mathbf{u}_j^t||^2 \qquad (14)$$

where $x_t^{\text{cube}}$ and $y_t^{\text{cube}}$ denote the cube's forward and lateral positions at timestep $t$, $\boldsymbol{p}_t^{\text{cube}}$ and $\boldsymbol{p}_t^{\text{root}}$ are the 3D positions of the cube and the agent's root body, and $\kappa = 0.1$ controls the lateral-motion penalty.

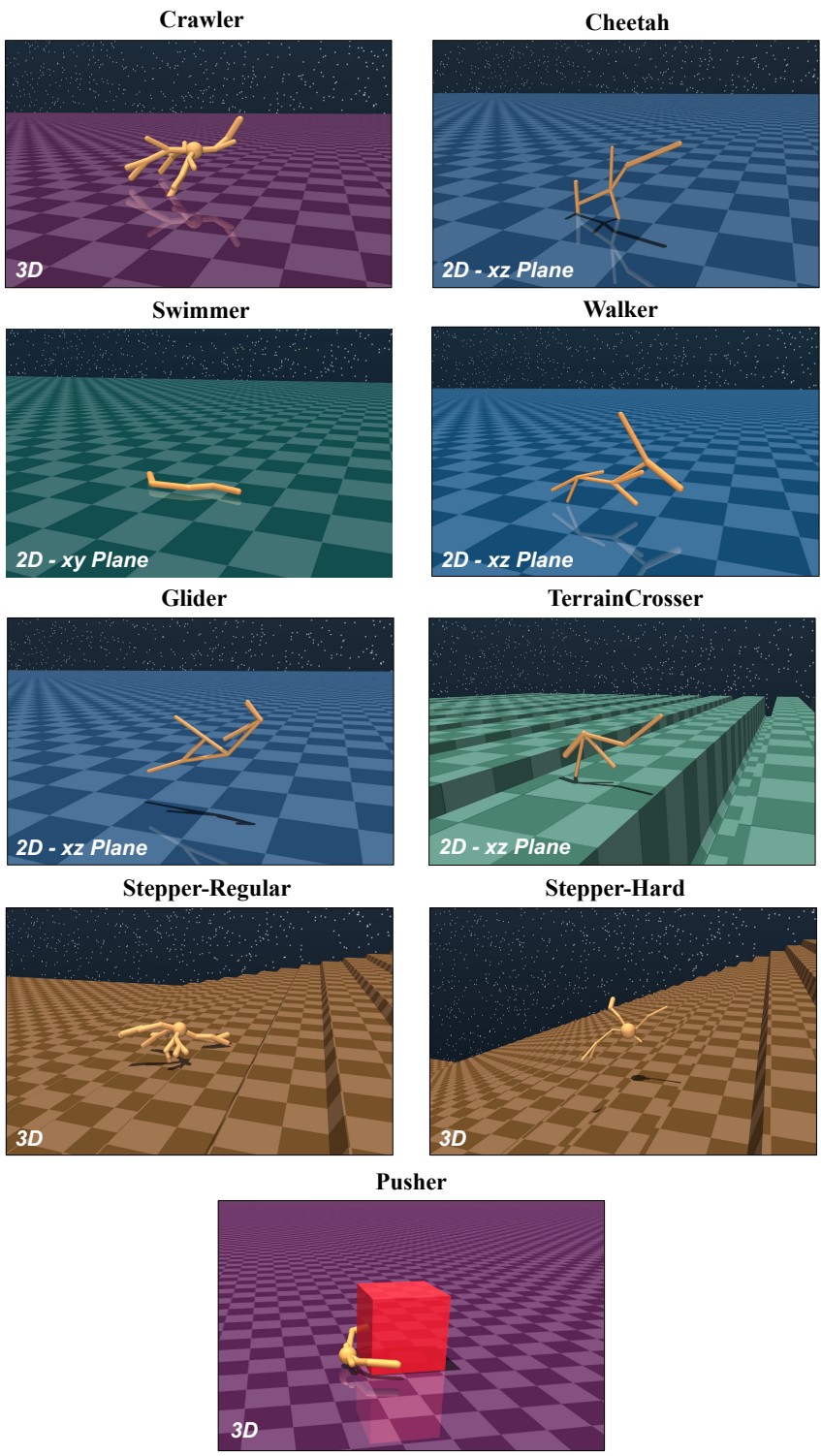

Figure 6: Visualization of the nine benchmark environments used in our experiments. Crawler, Stepper-Regular, Stepper-Hard, and Pusher are 3D tasks; others are 2D (x-z or x-y plane). The environments differ substantially in required morphology depth, symmetry, and limb arrangement, enabling evaluation on the generality of morphology–control co-design across locomotion and manipulation tasks.

# D IMPLEMENTATION DETAILS

## D.1 COMPUTATION COST

Following standard reinforcement learning practices, we utilize distributed trajectory sampling across multiple CPU threads to enhance training efficiency. All models are trained with seven random seeds on a high-performance computing cluster equipped with dual Intel® Xeon® processors (totaling 64 cores) and 24 NVIDIA A100 GPUs. Our implementation uses PyTorch 2.0.1 for all neural network models and MuJoCo 2.1.0 (Todorov et al., 2012) physics engine for the morphology-control simulation environments. The training process is computationally efficient, requiring approximately 30 hours per model when utilizing 10 CPU cores alongside a single NVIDIA A100 GPU across all experimental environments.

## D.2 HYPERPARAMETER CONFIGURATION

**Stackelberg PPO (Ours):** Our method introduces several Stackelberg-specific hyperparameters that require careful tuning. We conduct grid search over key parameters: Fisher information matrix regularization coefficient $\lambda \in \{0.5, 1.0, 5.0, 10.0\}$, maximum conjugate gradient (CG) steps $\in \{10, 20, 30\}$, and follower sampling steps per episode $\in \{6, 15, 30, 60, 100\}$ during leader update. For the underlying network architecture, we maintain the same configuration as BodyGen (Lu et al., 2025) without modification to ensure fair comparison, including their MoSAT transformer blocks and all network-related parameters. The final hyperparameter configuration, along with the underlying BodyGen network architecture we adopt, is detailed in Table 1.

**BodyGen:** We follow their original implementation and released code, adopting the same hyperparameter configuration as reported in their work (Lu et al., 2025). The settings include MoSAT Pre-LN normalization, SiLu activation, hidden dimension 64, policy learning rate 5e-5, value learning rate 3e-4, and other parameters as detailed in Table 1.

**Transform2Act:** Following the original implementation(Yuan et al., 2022), this baseline uses GraphConv layers, policy GNN size (64, 64, 64), policy learning rate 5e-5, value GNN size (64, 64, 64), value learning rate 3e-4, JSMLP activation Tanh, JSMLP size (128, 128, 128) for policy networks, and MLP size (512, 256) for value functions.

**NGE:** Based on the original implementations (Wang et al., 2019), this evolutionary baseline uses 125 generations, population size 20, elimination rate 0.15, with GraphConv layers, Tanh activation, policy GNN size (64, 64, 64), policy MLP size (128, 128), value GNN size (64, 64, 64), value MLP size (512, 256), policy learning rate 5e-5, and value learning rate 3e-4.

## D.3 GRADIENT NORMALIZATION

Recall Eq. 2, where the Stackelberg gradient for the leader decomposes into a *direct* term and a *response-induced* term. To avoid scale imbalance between these components, we scale the response-induced term by a data-dependent factor $\alpha$ computed from the relative norms of the two terms (no extra hyper-parameters). Let $g_{\mathrm{dir}} := \nabla_{\theta_L} J^L(\theta_L, \theta^F)$ and $g_{\mathrm{resp}} := (\nabla_{\theta_L} \theta_*^F(\theta_L))^\top \nabla_{\theta_F} J^L(\theta_L, \theta^F)$. We update the leader using

$$\widehat{\nabla_{\theta_L} J^L} = g_{\mathrm{dir}} - \alpha \, g_{\mathrm{resp}}, \qquad \alpha = \min\left(1, \frac{\|g_{\mathrm{dir}}\|_2}{\|g_{\mathrm{resp}}\|_2 + \varepsilon}\right), \tag{15}$$

where $\varepsilon > 0$ is a small numerical constant for stability. This rule guarantees $\alpha \|g_{\mathrm{resp}}\|_2 \leq \|g_{\mathrm{dir}}\|_2$, ensuring the follower-implicit component never dominates while preserving its direction. We use $\alpha = 1$ across all experiments for simplicity and consistency.

Table 1: Hyperparameters of Stackelberg PPO adopted in all experiments

| Hyperparameter | Value |
|---|---|
| Fisher Regularization Coefficient $\lambda$ | 5.0 |
| Maximum Conjugate Gradient Steps | 20 |
| CG Relative Error Tolerance | $10^{-3}$ |
| Follower Sampling Steps per Episode | 6 |
| Gradient Normalization Ratio $\alpha$ | 1.0 |
| Structure Design Steps $T^{stru}$ | 5 |
| Attribute Design Steps $T^{attr}$ | 1 |
| Transformer Layer Normalization | Pre-LN |
| Transformer Activation Function | SiLu |
| FNN Scaling Ratio $r$ | 4 |
| Transformer Blocks number (Policy Network) | 3 |
| Transformer Blocks number (Value Network) | 3 |
| Transformer Hidden Dimension (Policy Network) | 64 |
| Transformer Hidden Dimension (Value Network) | 64 |
| Optimizer | Adam |
| Policy Learning Rate | 5e-5 |
| Value Learning Rate | 3e-4 |
| Clip Gradient Norm | 40.0 |
| PPO Clip $\epsilon$ | 0.2 |
| PPO Batch Size | 50000 |
| PPO Minibatch Size | 2048 |
| PPO Iterations Per Batch | 10 |
| Training Epochs | 1000 |
| Discount factor $\gamma$ | 0.995 |
| GAE Parameter $\lambda_{\text{GAE}}$ | 0.95 |

# E ADDITIONAL RESULTS

## E.1 VISUALIZATION AND QUALITATIVE RESULTS

Figure 7 presents the diverse morphologies discovered by our Stackelberg PPO framework across different environments. The evolved body designs reveal the sophisticated structural complexity achieved by our approach, confirming that the Stackelberg game formulation enables continuous co-adaptation between morphology and control without premature convergence to suboptimal simple structures. Remarkably, these designs demonstrate emergent functional differentiation, developing specialized appendages for complementary tasks such as maintaining equilibrium versus providing propulsive forces.

As illustrated in the training curves presented in Fig. 3, we provide quantitative performance comparisons across all evaluated environments. Table 2 summarizes the final episode rewards achieved by each method, presenting mean values and standard deviations computed over seven random seeds. All baseline methods are configured using their optimal hyperparameter settings as reported in prior literature, with detailed specifications provided in Appendix D.2.

Table 2: Performance comparison of Stackelberg PPO against baseline methods across morphology-control co-design environments. Results show mean episode rewards and standard deviations over seven random seeds.

| Methods | Crawler | Cheetah | Swimmer |
|---|---|---|---|
| **Stackelberg PPO (Ours)** | $\mathbf{11047.90_{\pm 126.20}}$ | $\mathbf{13514.94_{\pm 653.62}}$ | $\mathbf{1334.98_{\pm 16.06}}$ |
| BodyGen (Lu et al., 2025) | $9098.72_{\pm 558.26}$ | $11575.87_{\pm 640.65}$ | $1302.64_{\pm 3.71}$ |
| Transform2Act (Yuan et al., 2022) | $3950.80_{\pm 268.43}$ | $8297.90_{\pm 825.02}$ | $737.90_{\pm 21.04}$ |
| NGE (Wang et al., 2019) | $1482.45_{\pm 524.97}$ | $2534.76_{\pm 428.68}$ | $384.45_{\pm 112.03}$ |
| ESS (Sims, 1994) | $631.67_{\pm 122.41}$ | $671.67_{\pm 134.65}$ | $190.62_{\pm 37.84}$ |

| Methods | Walker-Hard | Glider-Hard | TerrainCrosser |
|---|---|---|---|
| **Stackelberg PPO (Ours)** | $\mathbf{13612.32_{\pm 501.26}}$ | $\mathbf{12414.50_{\pm 498.53}}$ | $\mathbf{4488.07_{\pm 467.98}}$ |
| BodyGen (Lu et al., 2025) | $11645.89_{\pm 797.77}$ | $11049.95_{\pm 468.44}$ | $4103.25_{\pm 871.90}$ |
| Transform2Act (Yuan et al., 2022) | $4420.63_{\pm 267.48}$ | $6120.62_{\pm 1086.62}$ | $2364.63_{\pm 473.80}$ |
| NGE (Wang et al., 2019) | $1504.55_{\pm 553.15}$ | $2081.25_{\pm 348.17}$ | $827.15_{\pm 427.21}$ |
| ESS (Sims, 1994) | $636.03_{\pm 125.74}$ | $541.55_{\pm 107.56}$ | $426.81_{\pm 168.30}$ |

| Methods | Pusher | Stepper-Regular | Stepper-Hard |
|---|---|---|---|
| **Stackelberg PPO (Ours)** | $\mathbf{3462.77_{\pm 368.09}}$ | $\mathbf{7215.20_{\pm 449.02}}$ | $\mathbf{6003.59_{\pm 1027.77}}$ |
| BodyGen (Lu et al., 2025) | $2779.95_{\pm 509.18}$ | $4685.94_{\pm 845.23}$ | $4685.41_{\pm 800.09}$ |
| Transform2Act (Yuan et al., 2022) | $1015.28_{\pm 247.09}$ | $2325.69_{\pm 664.00}$ | $1192.39_{\pm 544.20}$ |
| NGE (Wang et al., 2019) | $551.57_{\pm 120.65}$ | $870.56_{\pm 215.45}$ | $509.12_{\pm 207.01}$ |
| ESS (Sims, 1994) | $243.14_{\pm 95.93}$ | $351.02_{\pm 136.28}$ | $392.54_{\pm 151.83}$ |

**Crawler**

**Cheetah**

**Swimmer**

**Walker**

**Glider**

**Pusher**

**TerrainCrosser**

**Stepper-Regular**

**Stepper-Hard**

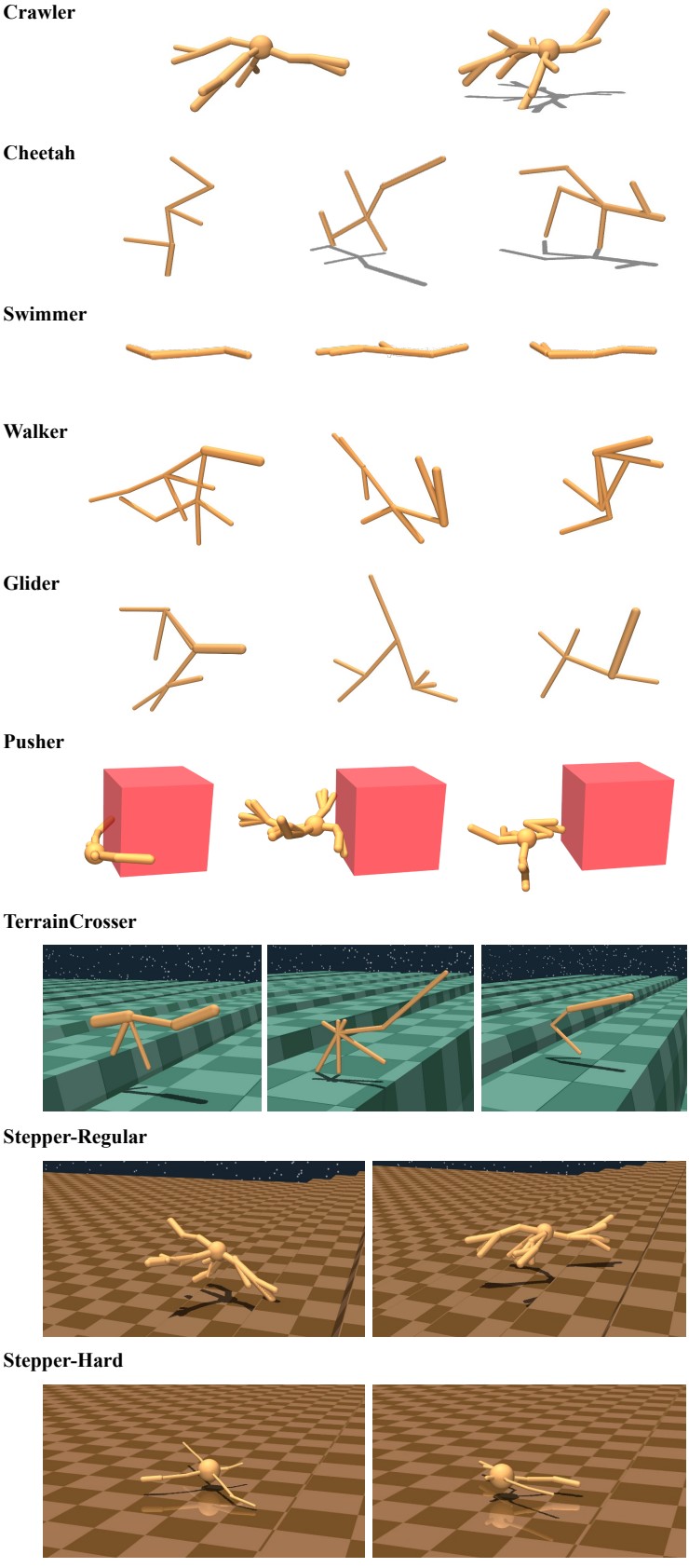

Figure 7: Visualization of co-evolved body designs generated through our Stackelberg PPO.

### E.2 EXTENDED ENVIRONMENT EVALUATION

Beyond the environments reported in the main paper, we also include results for the full sets of Glider and Walker tasks, each provided in three difficulty levels: regular, medium, and hard. In the main text we present only the hard variants, as they offer the largest design spaces and naturally encompass the easier tiers, providing a clearer view of morphology–control co-design under less restrictive structural budgets. Here, we report the complete results mainly for completeness, and to illustrate how our method behaves under different morphology complexity limits. The six environments differ only in their structural allowances: Glider uses a maximum tree depth of 3 and Walker a maximum depth of 4, while the regular/medium/hard variants correspond to maximum child counts of $\{1, 2, 3\}$. All other environment settings are identical.

Figure 8 presents the training curves and the final generated morphologies across all six tasks. Our method outperforms the baseline across all difficulty levels, with the performance gap increasing as the design space becomes larger and more challenging. Interestingly, the three difficulty tiers within each environment achieve similar final performance, suggesting that overall task success is not strictly tied to structural complexity: even simpler configurations can discover diverse, correct, and high-quality locomotion patterns.

### E.3 ADDITIONAL ABLATION STUDIES AND MECHANISM ANALYSIS

**Quantitative Results for PPO Clipping Sensitivity.** To complement the qualitative trends shown in Figure 5(a), we provide the full quantitative statistics for the clipping sweep experiment. The purpose of this analysis is to examine how the surrogate objective behaves under different clipping thresholds and to identify when the underlying assumptions of policy-gradient theory remain valid. From a theoretical perspective, large policy updates can cause the surrogate objective to diverge from the true return, leading to instability. PPO addresses this by bounding the likelihood ratio $\pi_\theta(a \mid s)/\pi_{\theta_0}(a \mid s)$ within $[1-\epsilon, 1+\epsilon]$, which prevents overly aggressive updates and ensures that the surrogate remains a reliable approximation. In this experiment, we vary the clipping parameter $\epsilon$ and measure three quantities that together characterize the stability of the update rule: (1) average performance, (2) likelihood-ratio constraint violations, and (3) KL divergence. Table 3 reports the full numerical results corresponding to the curves shown in the main text.

Table 3: Sensitivity of Stackelberg PPO to the clipping threshold $\epsilon$.

| Clipping Parameter | Performance | Likelihood Ratio Violations (%) | Average KL Divergence |
|---|---|---|---|
| $\epsilon = 0.1$ | $4934.52_{\pm 646.40}$ | $14.82_{\pm 0.55}$ | $0.0030_{\pm 0.0006}$ |
| $\epsilon = 0.2$ | $7215.20_{\pm 449.02}$ | $13.39_{\pm 0.83}$ | $0.0196_{\pm 0.0025}$ |
| $\epsilon = 0.4$ | $\mathbf{7907.01}_{\pm \mathbf{208.02}}$ | $9.92_{\pm 0.94}$ | $0.0343_{\pm 0.0153}$ |
| $\epsilon = 0.6$ | $4778.18_{\pm 407.84}$ | $9.10_{\pm 1.30}$ | $0.0665_{\pm 0.0188}$ |
| $\epsilon = 0.8$ | $2656.92_{\pm 503.93}$ | $7.02_{\pm 0.81}$ | $0.1340_{\pm 0.0388}$ |
| No Clipping | $1233.26_{\pm 443.98}$ | $0$ | $1.7726_{\pm 0.1539}$ |

**Ablation on SID Components and PPO Clipping.** To further disentangle the contributions of our Stackelberg Implicit Differentiation (SID) estimator and PPO clipping, we conduct an additional controlled ablation. Specifically, we evaluate three variants under the same phase-separated, non-differentiable Stackelberg setup:

- **SID+PPO (full)** — our complete method using both SID and PPO clipping,
- **PPO-only** — standard PPO updates without SID,
- **SID-only** — applying our SID estimator without PPO clipping.

This ablation assesses whether (i) our SID estimator meaningfully improves leader optimization and (ii) PPO clipping is required to stabilize the induced surrogate objectives. As shown in Table 4, both components provide clear performance gains, and the full algorithm consistently achieves the highest returns across four environments.

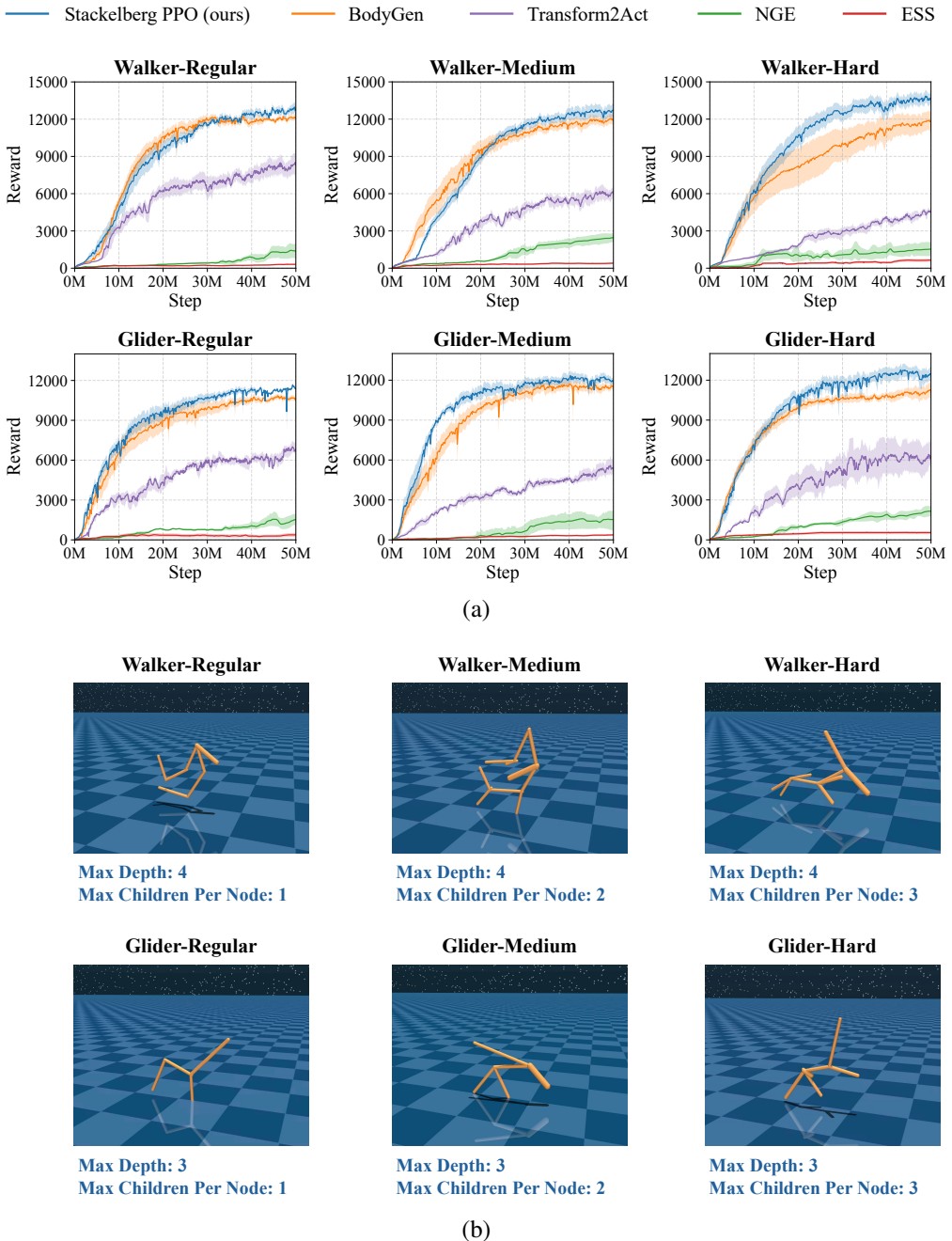

Figure 8: Extended evaluation on Glider and Walker environments under different morphology complexity budgets.(a) Training curves for the regular, medium, and hard variants of each environment. (b) Final generated morphologies under each complexity tier.

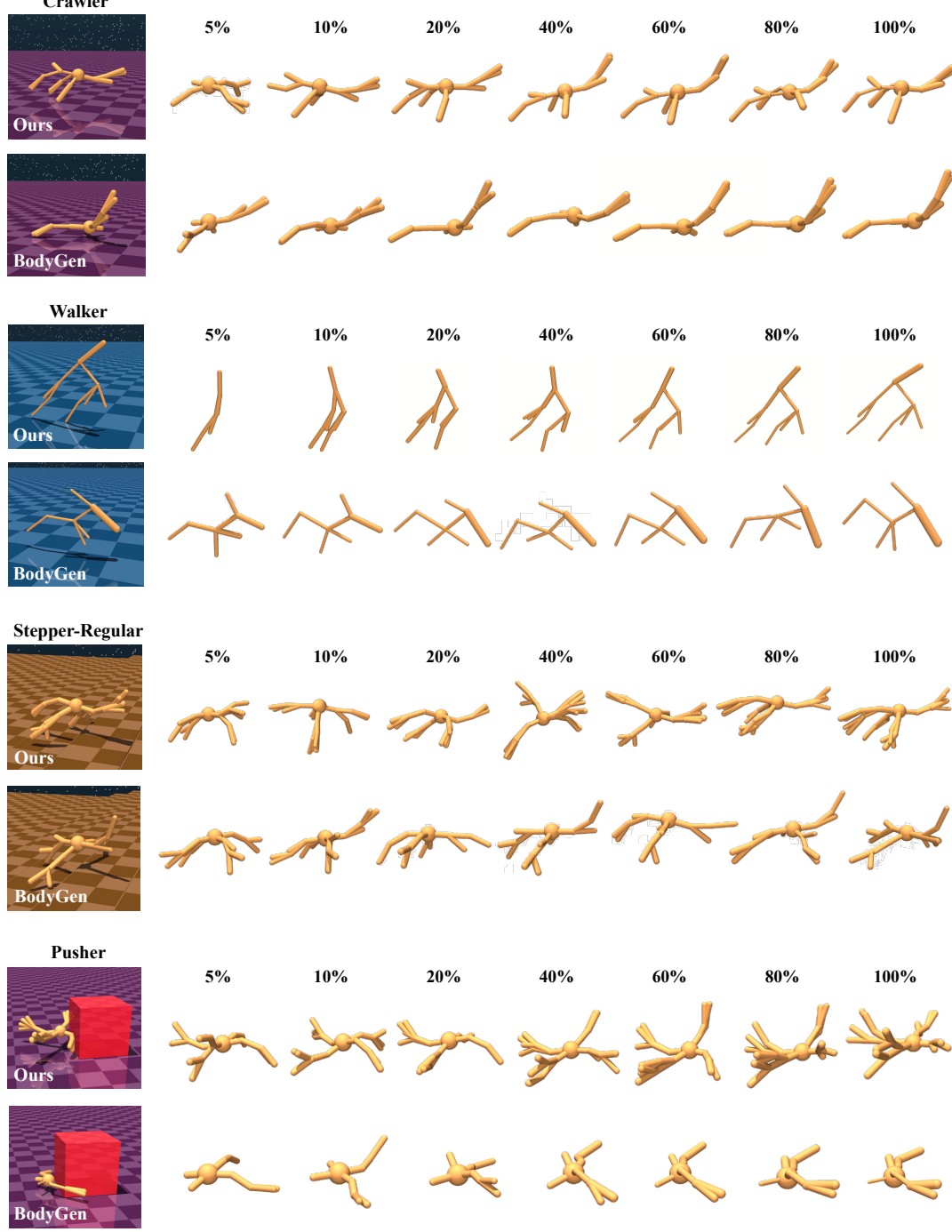

Figure 9: Comparison of morphology evolution between Stackelberg PPO (ours) and Standard PPO (BodyGen) across four environments. BodyGen tends to collapse early into low-complexity designs, while Stackelberg PPO continues exploring structurally richer morphologies, yielding more capable final structures.

Table 4: Ablation studies on the components of our SID estimator and PPO clipping, evaluated under the same phase-separated, non-differentiable Stackelberg setting.

| Environment | SID+PPO (full) | PPO-only (no SID) | SID-only (no clipping) |
|---|---|---|---|
| Stepper-Regular | $\mathbf{7215.20_{\pm 449.02}}$ | $4685.94_{\pm 845.23}$ | $1257.33_{\pm 530.25}$ |
| Crawler | $\mathbf{11047.90_{\pm 126.20}}$ | $9098.72_{\pm 558.26}$ | $35.77_{\pm 12.25}$ |
| Cheetah | $\mathbf{13514.94_{\pm 653.62}}$ | $11575.87_{\pm 640.65}$ | $472.89_{\pm 77.40}$ |
| Glider | $\mathbf{12414.50_{\pm 498.53}}$ | $11049.95_{\pm 468.44}$ | $566.81_{\pm 89.96}$ |

**Effect of Leader Gradients on Controller Adaptation.** To better understand the mechanism behind Stackelberg PPO's performance gains, we analyze how morphology updates interact with controller adaptation. Specifically, we investigate whether the improved performance originates from faster controller adaptation under changing morphologies, or from more informative leader gradients that guide the structure search more effectively. To isolate these effects, we extract *ten intermediate checkpoints* from a BodyGen training run (spanning 10%–100% of training progress). From each checkpoint, we initialize both methods with *identical* morphology, controller parameters, and optimizer state, and then train each method for a *single epoch*. This setup ensures that any difference in performance improvement reflects differences in the leader update rule, rather than controller initialization or long-term training.

As shown in Table 5, Stackelberg PPO consistently achieves a larger one-epoch performance improvement compared to standard PPO (BodyGen). This indicates that Stackelberg PPO does not rely on faster controller adaptation; instead, it provides more informative leader gradients that enable the morphology to improve even when the controller is only partially adapted. These results highlight the role of the Stackelberg update in stabilizing and accelerating the joint morphology–control optimization process.

Table 5: Average performance change after one epoch of training from the same checkpoint model, averaged over 10 checkpoints and 7 seeds, evaluated on Stepper-Regular.

| | Stackelberg PPO (Ours) | BodyGen (PPO) |
|---|---|---|
| Performance Change After 1 Epoch | $\mathbf{+0.392_{\pm 0.075}\%}$ | $+0.224_{\pm 0.043}\%$ |

We further provide a visual comparison of morphology evolution to illustrate this effect (Figure 9). Across multiple environments, BodyGen tends to converge early to low-complexity designs, which restricts later improvements even as the controller becomes stronger. In contrast, Stackelberg PPO continues meaningful structural exploration throughout training, enabling richer and more adaptive morphologies. These qualitative trajectories align with the adaptation results above, reinforcing that the Stackelberg update produces more informative and better-aligned structural gradients.

### E.4 SAMPLE AND TRAINING EFFICIENCY

**Sample Efficiency**. To assess the efficiency of different co-design algorithms, we measure how many environment interaction samples are required to reach a predefined performance threshold. As reported in Table 6, Stackelberg PPO consistently converges with substantially fewer samples across all environments. On average, it reaches the threshold with approximately -39% fewer samples than BodyGen. In contrast, Transform2Act, NGE, and ESS fail to reach any threshold within the available training budget. These results highlight the advantage of explicitly modeling morphology–control coupling via a Stackelberg formulation, enabling faster convergence and more stable co-design dynamics.

Table 6: Sample efficiency comparison: number of samples (in millions) required to reach the performance threshold.

| Environment | Threshold | Stackelberg PPO | BodyGen | Transform2Act | NGE | ESS |
|---|---|---|---|---|---|---|
| Crawler | *9000* | **25.8** | 47.2 | $\infty$ | $\infty$ | $\infty$ |
| Cheetah | *11000* | **19.2** | 42.1 | $\infty$ | $\infty$ | $\infty$ |
| Swimmer | *1200* | **14.8** | 17.0 | $\infty$ | $\infty$ | $\infty$ |
| Walker-Hard | *10000* | **18.1** | 30.3 | $\infty$ | $\infty$ | $\infty$ |
| Glider-Hard | *11000* | **23.6** | 49.7 | $\infty$ | $\infty$ | $\infty$ |
| TerrainCrosser | *3500* | **23.9** | 33.8 | $\infty$ | $\infty$ | $\infty$ |
| Pusher | *2500* | **29.3** | 39.1 | $\infty$ | $\infty$ | $\infty$ |
| Stepper-Regular | *4500* | **18.5** | 40.4 | $\infty$ | $\infty$ | $\infty$ |
| Stepper-Hard | *4500* | **27.2** | 43.1 | $\infty$ | $\infty$ | $\infty$ |

**Training Efficiency.** Despite incorporating a bilevel update, Stackelberg PPO introduces only modest computational overhead. The method avoids explicit Hessian construction or inversion; instead, the conjugate-gradient step relies solely on efficient Hessian–vector products (approximately one backward pass). As a result, its cost scales *linearly* with morphology and controller dimensionality, rather than quadratically. Moreover, rollout collection dominates overall computation in all co-design settings, so the additional optimization cost has limited influence on total training time. Table 7 summarizes the training time under different morphology/control design spaces. Increasing the structural search space does not incur superlinear overhead, confirming the scalability of Stackelberg PPO. The comparison with ES-based approaches in Table 8 further shows that ES reduces wall-clock time only when substantial CPU parallelization is available, while its resulting designs remain far less effective than those produced by PPO-based methods.

Overall, our method achieves strong efficiency–performance trade-offs:

- Compared to BodyGen, Stackelberg PPO achieves substantially better sample efficiency by requiring **-39%** fewer samples to reach the performance threshold while also obtaining **+20.66%** higher final scores. In terms of wall-clock time, the difference between the two methods is modest **(+13%)**, keeping the overall training cost comparable.

- Compared to ES-based baselines, although ESS attains shorter wall-clock time using **6×** more CPU cores (64 cores), its performance is extremely poor, achieving only a **0.16** fraction of our method's performance.

Table 7: Wall-clock training time comparison across environments with different design space sizes (10 CPU cores + A100 GPU).

| Environment | Space Size (mean) | Space Size (max) | Stackelberg PPO | BodyGen (PPO) |
|---|---|---|---|---|
| TerrainCrosser | $4.50_{\pm 0.76}$ | 14 | $33.88_{\pm 0.42}$ | $27.87_{\pm 1.27}$ |
| Swimmer | $5.50_{\pm 0.76}$ | 14 | $32.64_{\pm 0.74}$ | $28.13_{\pm 0.45}$ |
| Cheetah | $6.57_{\pm 0.90}$ | 14 | $32.96_{\pm 0.67}$ | $29.52_{\pm 1.03}$ |
| Glider-Hard | $7.33_{\pm 1.49}$ | 9 | $32.93_{\pm 0.71}$ | $28.93_{\pm 1.50}$ |
| Walker-Hard | $8.43_{\pm 1.50}$ | 27 | $32.54_{\pm 0.62}$ | $30.21_{\pm 1.22}$ |
| Stepper-Hard | $9.57_{\pm 0.90}$ | 29 | $32.70_{\pm 1.01}$ | $30.25_{\pm 2.06}$ |
| Pusher | $14.33_{\pm 4.07}$ | 29 | $33.41_{\pm 0.82}$ | $29.24_{\pm 1.12}$ |
| Stepper-Regular | $16.40_{\pm 4.69}$ | 29 | $32.83_{\pm 0.87}$ | $30.17_{\pm 1.41}$ |
| Crawler | $18.25_{\pm 1.29}$ | 29 | $33.73_{\pm 0.64}$ | $30.54_{\pm 1.33}$ |

Table 8: Wall-clock training time across methods. NGE results are shown under both 10 CPU cores and 64 cores to illustrate parallelization effects.

| | Stackelberg PPO (10 cores) | BodyGen (10 cores) | NGE (10 cores) | NGE (64 cores) |
|---|---|---|---|---|
| Wall-clock Time | $33.07_{\pm 0.49}$ h | $29.43_{\pm 0.97}$ h | $45.16_{\pm 3.72}$ h | $13.52_{\pm 1.52}$ h |

### E.5 Morphology Evolution Process Visualization

Figure 10 showcases the morphological evolution trajectories discovered by our Stackelberg PPO framework across diverse locomotion tasks and environments. Each row represents a distinct embodiment (Crawler, Cheetah, Swimmer, Glider, Stepper-Regular, Stepper-Hard, Terrain Crosser, Walker, and Pusher), and the columns depict the progressive morphological changes from early evolution (5%) through convergence (100%). The evolution demonstrates emergent specialization of appendages for task-specific locomotion requirements.

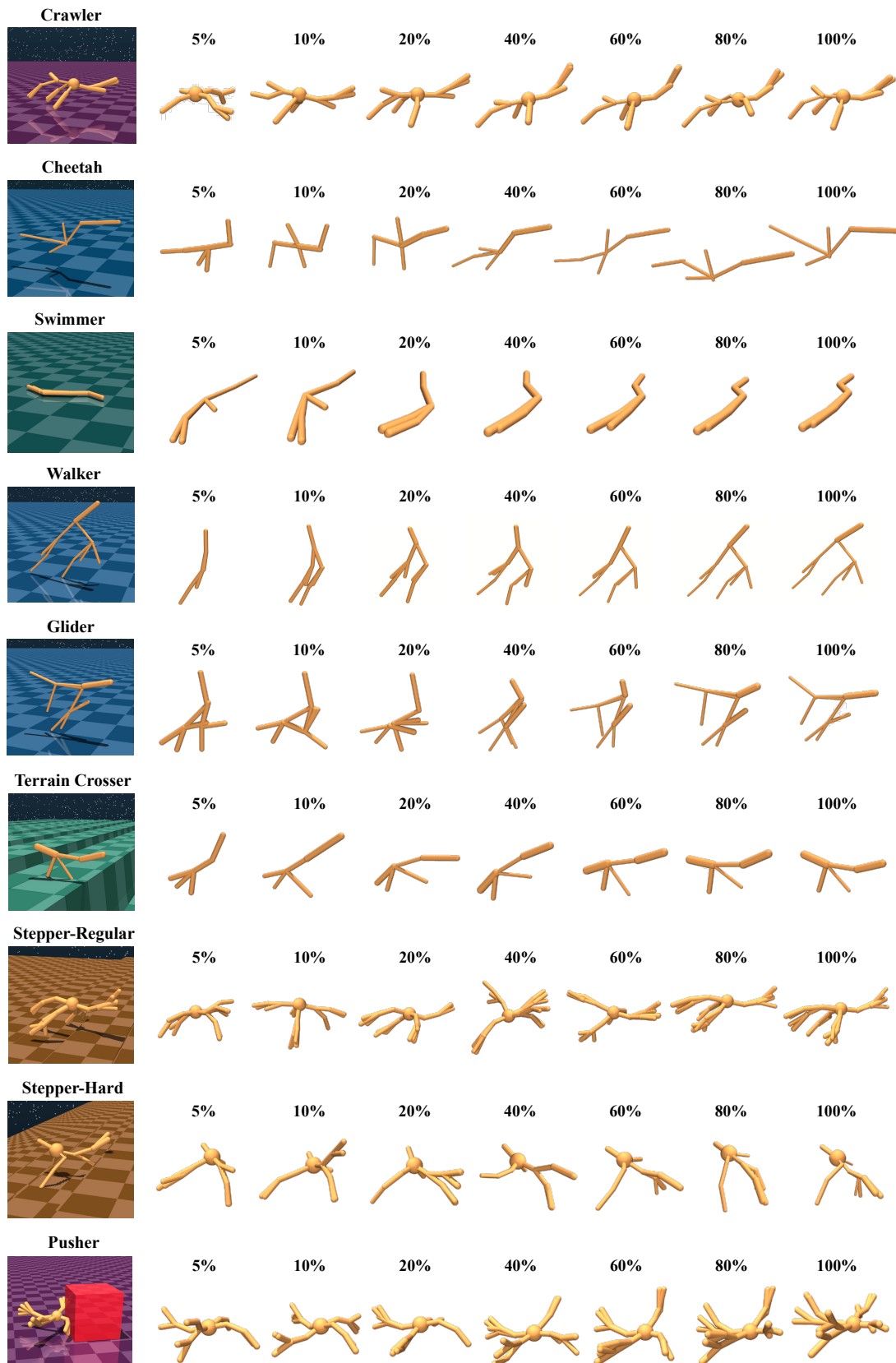

Figure 10: Morphological evolution trajectories across eight environments. Each row represents a distinct robot embodiment, with columns showing progressive stages of morphological adaptation from 5% to 100% training progress.

### E.6 RESULTS UNDER REALISTIC CO-DESIGN CONSTRAINTS

In the main paper, we adopt a unified forward-progress reward to ensure fair comparison across algorithms and to avoid introducing task-specific reward biases. While this setup is standard and suitable for benchmarking algorithmic contributions, real-world robot design is often shaped by additional engineering constraints. To better understand the practical co-design behavior of Stackelberg PPO, we further evaluate five common realistic constraints under an identical crawler task and training budget.

These constraints span both morphology- and control-level considerations, including power usage, manufacturability, torque limits, payload handling, and robustness. Several of these factors are already captured by our experimental setup:

- Power usage: Energy expenditure is discouraged through a small effort penalty included in the reward (Equation eq. (12)).

- Torque limits: Joint torque capacity is implicitly limited by bounding the "allowable torque" attribute during morphology design.

- Manufacturability: Physical realizability is enforced by constraining morphology-editing attributes such as limb length, joint count, and topology depth (see Appendix C).

- Robustness: Robustness naturally emerges from the evaluation protocol: each morphology–controller pair is scored using multiple rollouts, causing non-robust designs to yield lower averaged returns.

To complement these built-in constraints, we further provide more detailed quantitative experiments that isolate and measure their individual effects.

**Power Constraint.** We evaluate performance under various power penalty coefficients (0.001, 0.01, 0.1), extending beyond the mild penalty (0.0001) used in the main experiments. Table 9 reports the detailed performance and control-effort statistics under each penalty coefficient. The generated morphologies are visualized in Figure 11. Increasing the penalty produces three consistent effects:

- Impact on Performance (velocity reward). As the penalty coefficient increases, both methods experience reduced forward-progress reward. However, Stackelberg PPO exhibits a substantially smaller degradation, maintaining stronger performance across all tested settings.

- Impact on Control Effort. Larger penalties encourage more conservative actuation strategies for both approaches, reflected by the lower penalty terms in the table.

- Impact on Morphology–Control Co-Design. With stronger penalties, the optimized morphologies tend to adopt shorter, thicker, and more symmetric limbs, paired with low-torque gaits characteristic of energy-efficient locomotion.

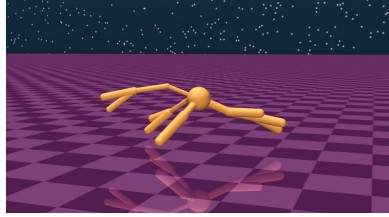
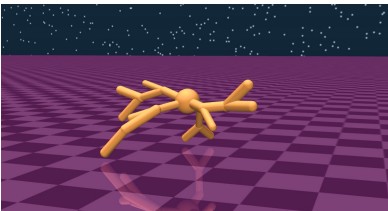

**Penalty Coef. = 0.0001**  **Penalty Coef. = 0.1**

Figure 11: Power constraint under different penalty coefficients. As the penalty increases, the co-designed morphologies transition toward shorter, thicker, and more symmetric limbs.

Table 9: Power-constraint setting: performance and power penalties under different penalty coefficients.

| Penalty Coef. | Performance | | Power Penalty | |
|---|---|---|---|---|
| | Stackelberg PPO (ours) | BodyGen | Stackelberg PPO (ours) | BodyGen |
| 0.0001 | $\mathbf{11047.90_{\pm 126.20}}$ | $9098.72_{\pm 558.26}$ | $5631.42_{\pm 674.03}$ | $2745.84_{\pm 284.55}$ |
| 0.001 | $\mathbf{10191.15_{\pm 371.81}}$ | $7501.61_{\pm 671.33}$ | $5342.16_{\pm 584.25}$ | $5948.16_{\pm 254.84}$ |
| 0.01 | $\mathbf{9853.19_{\pm 229.37}}$ | $8304.00_{\pm 497.56}$ | $1582.48_{\pm 697.61}$ | $468.12_{\pm 516.33}$ |
| 0.1 | $\mathbf{10585.25_{\pm 146.80}}$ | $8974.48_{\pm 574.29}$ | $25.50_{\pm 22.27}$ | $26.34_{\pm 23.64}$ |

**Manufacturability Constraint.** A manufacturability cost penalty is applied by incorporating two components into the leader objective: structural complexity is measured by the number of body elements, and material cost is defined as the total mass. Table 10 summarizes the resulting performance and morphology characteristics under different penalty coefficients. The trends are consistent with those observed in the power constraint experiments in (i): our method consistently achieves better reward–cost tradeoffs across all penalty levels. As shown in Figure 12, the generated morphologies are compact than the original structure, with fewer distal branches, shorter limbs, and mass concentrated near the root. These structures exhibit lower inertia and more efficient force transmission, supporting stable forward locomotion under cost constraints.

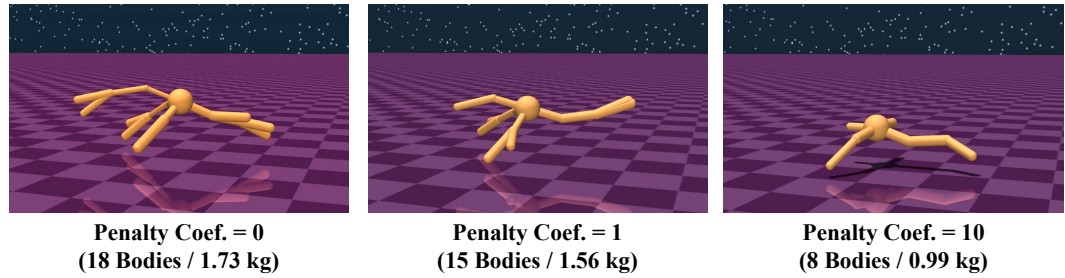

**Penalty Coef. = 0**
**(18 Bodies / 1.73 kg)**
    **Penalty Coef. = 1**
**(15 Bodies / 1.56 kg)**
    **Penalty Coef. = 10**
**(8 Bodies / 0.99 kg)**

Figure 12: Manufacturability constraint under different penalty coefficients. Higher penalties on structure complexity and material mass encourage designs with fewer body elements, reduced branching, and mass concentrated near the root, producing compact morphologies that are easier to fabricate.

Table 10: Manufacturability constraint setting: performance, morphology complexity, and material cost under different penalty levels.

| Penalty Coef. | Performance | | Morphology Complexity | | Material Cost | |
|---|---|---|---|---|---|---|
| | Stackelberg PPO (ours) | BodyGen | Stackelberg PPO (ours) | BodyGen | Stackelberg PPO (ours) | BodyGen |
| 0 | $\mathbf{11047.90_{\pm 126.20}}$ | $9098.72_{\pm 558.26}$ | $16.40_{\pm 2.45}$ | $13.67_{\pm 2.08}$ | $1.71_{\pm 0.23}$ | $1.57_{\pm 0.32}$ |
| 1 | $\mathbf{7892.93_{\pm 349.84}}$ | $6531.37_{\pm 437.26}$ | $13.67_{\pm 2.03}$ | $9.67_{\pm 1.61}$ | $1.59_{\pm 0.18}$ | $1.32_{\pm 0.16}$ |
| 10 | $\mathbf{6825.47_{\pm 303.09}}$ | $5372.10_{\pm 364.79}$ | $8.25_{\pm 0.91}$ | $7.50_{\pm 1.24}$ | $0.94_{\pm 0.08}$ | $0.93_{\pm 0.10}$ |

**Torque Limits Constraint.** A torque-limit penalty is incorporated by enforcing a 50 N·m cap on all joints and adding a proportional violation cost to the leader objective. Table 11 summarizes the quantitative results, and the morphological effects are shown in Figure 13. As in the manufacturability and control-effort settings, our method achieves stronger reward–cost tradeoffs when the controller retains sufficient expressiveness (penalty = 0.01). Under the stronger penalty (0.1), the tightened actuation constraints reduce the feasible morphology space for all methods, narrowing the performance gap.

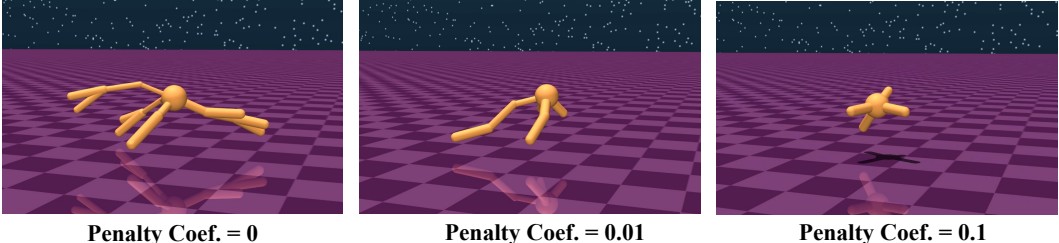

| Penalty Coef. = 0 | Penalty Coef. = 0.01 | Penalty Coef. = 0.1 |

Figure 13: Torque limits constraint under different penalty coefficients. Tighter actuation limits lead to noticeably simpler and more compact structures, with shorter limbs and reduced distal branching.

Table 11: Torque limits constraint: performance and torque-violation penalties under different torque-penalty coefficients.

| Penalty Coef. | Performance | | Limit Violation Penalty | |
|---|---|---|---|---|
| | **Stackelberg PPO (ours)** | **BodyGen** | **Stackelberg PPO (ours)** | **BodyGen** |
| 0.01 | $\mathbf{7893.42_{\pm 84.62}}$ | $6311.75_{\pm 98.31}$ | $20210.50_{\pm 6503.22}$ | $11350.45_{\pm 5338.31}$ |
| 0.1 | $\mathbf{3133.01_{\pm 70.44}}$ | $3121.80_{\pm 54.03}$ | $1106.45_{\pm 64.69}$ | $899.35_{\pm 49.40}$ |

**Payload Constraint.** To evaluate the agent's ability to maintain locomotion under additional load, we attach an extra mass to the root link to serve as a payload. During training, the payload value is randomized within a fixed range (0-0.6 kg) to promote generalization. After training, we evaluate each method under three fixed payload levels (0.2 kg, 0.4 kg, 0.6 kg). As shown in Table 12, Stackelberg PPO consistently maintains higher forward progress across all payload settings. Figure 14 further compares morphologies trained with and without payload. Under load, the evolved structures become more symmetric and better support the additional mass, indicating that Stackelberg PPO adapts the topology itself rather than relying solely on controller compensation.

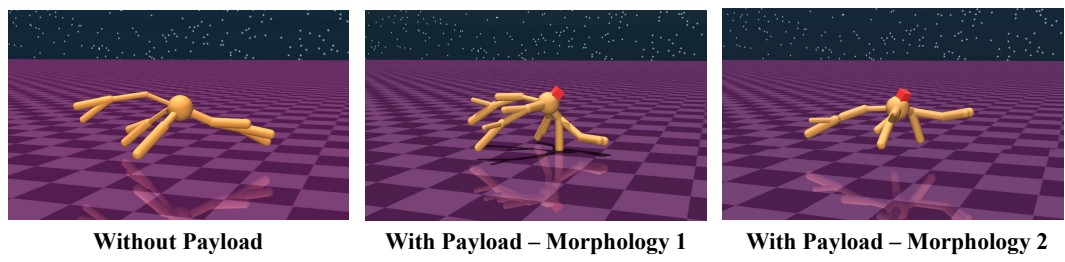

| Without Payload | With Payload – Morphology 1 | With Payload – Morphology 2 |

Figure 14: Morphology comparison trained with and without payload. Payload induces more symmetric and load-supporting structures.

Table 12: Payload constraint: performance comparison under different payload weights.

| Payload Weight | Stackelberg PPO (ours) | BodyGen |
|---|---|---|
| 0.2 kg | $\mathbf{8675.08}_{\pm\mathbf{286.42}}$ | $5186.31_{\pm659.87}$ |
| 0.4 kg | $\mathbf{6966.34}_{\pm\mathbf{473.43}}$ | $5116.54_{\pm546.55}$ |
| 0.6 kg | $\mathbf{7347.46}_{\pm\mathbf{478.01}}$ | $4523.89_{\pm536.99}$ |

**Robustness Evaluation.** We evaluate robustness under two settings: random external forces applied to the root body at every control step, and terrain friction noise created by randomly varying the ground's friction in each episode. For each disturbance level, all policies are tested across multiple stochastic rollouts, and we report the resulting forward-progress reward. Tables 13 and 14 summarize the results. Across all disturbance magnitudes, Stackelberg PPO consistently demonstrates substantially higher robustness. For example, when external forces increase from 2 N to 6 N, performance decreases by only 5.91% for Stackelberg PPO, compared to a much larger 59.57% decline for BodyGen. A similar pattern holds under terrain friction noise. These improvements arise primarily from more symmetric, mechanically balanced morphologies that better tolerate external forces and friction variability.

Table 13: Robustness evaluation: performance under different levels of external disturbance forces.

| Level | Stackelberg PPO (ours) | BodyGen |
|---|---|---|
| 2.0 N | $\mathbf{11557.31}_{\pm\mathbf{124.68}}$ | $6963.05_{\pm450.48}$ |
| 4.0 N | $\mathbf{11290.13}_{\pm\mathbf{164.54}}$ | $4621.82_{\pm597.71}$ |
| 6.0 N | $\mathbf{10875.23}_{\pm\mathbf{250.97}}$ | $2816.16_{\pm857.83}$ |

Table 14: Robustness evaluation: performance under different levels of terrain friction noise.

| Level | Stackelberg PPO (ours) | BodyGen |
|---|---|---|
| 30% | $\mathbf{11424.66}_{\pm\mathbf{112.08}}$ | $7326.04_{\pm421.73}$ |
| 50% | $\mathbf{11333.43}_{\pm\mathbf{141.42}}$ | $6795.55_{\pm493.31}$ |
| 70% | $\mathbf{10892.09}_{\pm\mathbf{149.72}}$ | $5062.85_{\pm579.66}$ |

### E.7 DISCUSSION AND EXTENDED EVALUATION ON REALISTIC CO-DESIGN CHALLENGES

In this section, we present broader analyses of morphology–control co-design and extend our results along four representative challenge dimensions: (i) diverse co-design environments, (ii) multi-objective and role-specific rewards, (iii) robustness and generalization under unseen disturbances, and (iv) the use of morphology priors. These studies highlight both the empirical advantages of Stackelberg PPO and the conceptual benefits of explicitly decoupling structure design from control learning. Together, they demonstrate that our framework scales naturally to more complex co-design settings that better reflect real-world robotic demands, and they point toward promising directions for building more adaptive and physically grounded morphology–control systems.

**Diverse co-design environments.** Standard co-design benchmarks focus almost exclusively on flat-terrain locomotion, which poses limited structural or behavioral challenge. To expose a broader range of morphology–control interactions, we introduce more demanding environments—most notably difficult terrain and manipulation—that require non-periodic motions, contact management, and functional differentiation across limbs. In the Stepper environments, agents must coordinate structure and control to handle large discontinuities without exteroceptive sensing. On low stairs, they develop stable stepping and small hops; on high stairs, the difficulty induces long-range, high-amplitude jumping behaviors. These emergent solutions reflect the stronger morphological and dynamical adaptation required by complex terrain. In the pusher task, co-design must jointly support locomotion and precise force application. Learned morphologies exhibit clear role specialization: some limbs provide acceleration and stability, while others regulate contact orientation and apply controlled pushing forces. Baseline methods typically recover only the locomotion component, relying on collision-based propulsion. These environments reveal aspects of the co-design problem that flat locomotion cannot capture, and they demonstrate that Stackelberg PPO scales to richer settings requiring terrain adaptation, contact reasoning, and multi-role morphology design.

**Multi-objective and role-specific reward design.** As shown earlier in Appendix E.6, our framework naturally accommodates additional objectives such as power consumption or payload capacity. The resulting morphologies and controllers smoothly adapt to the trade-offs introduced by these objectives, validating the method's multi-objective co-design capability. Furthermore, the leader–follower decomposition allows reward terms to be assigned selectively to the structure-design or control-learning stages. For example, complexity or material-cost penalties can be applied only to the leader (structure) updates, enabling constraints on morphology without interfering with controller learning. This role-specific reward routing provides a high degree of flexibility for real-world design requirements.

**Robustness and generalization under unseen disturbances** Although our current setting does not include exteroceptive sensing and is not intended for zero-shot transfer to arbitrary unseen worlds, we evaluate generalization and robustness under an obstacle-navigation task not seen during training. Policies are trained only on flat terrain (Crawler task) and then tested in environments containing either sparse or dense grids of square obstacles. As reported in Table 15, Stackelberg PPO obtains higher forward progress than BodyGen across both difficulty levels. The visualization in Figure 15 further shows that the morphologies produced by our method maintain more consistent forward motion, whereas baseline agents more frequently stall or deviate under unexpected contacts. These results illustrate that the co-designed morphology–policy pair exhibits meaningful robustness to previously unseen disturbances and obstacle interactions.

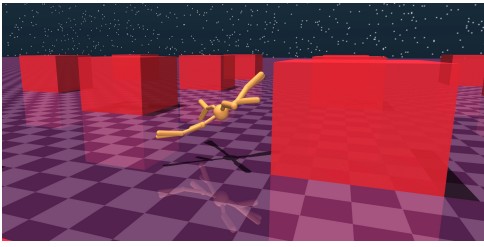

**Dense Obstacle Navigation**     **Sparse Obstacle Navigation**

Figure 15: Visualization the unseen obstacle-navigation task environment.

Table 15: Performance in the unseen obstacle-navigation task under two obstacle densities.

| Obstacle Type | Spacing | Performance | |
|---|---|---|---|
| | | Stackelberg PPO (ours) | BodyGen |
| Sparse Obstacle | 16 m (∼4× robot width) | $\mathbf{1790.45_{\pm 161.77}}$ | $1061.55_{\pm 228.40}$ |
| Dense Obstacle | 8 m (∼2× robot width) | $\mathbf{1698.52_{\pm 733.02}}$ | $1007.21_{\pm 157.42}$ |

**Incorporating and benefiting from morphology priors.** Our framework also supports reusing morphology priors obtained from related tasks. To examine this, we transfer morphologies evolved in the Crawler environment to initialize training in the Pusher task. Table 16 shows that both Stackelberg PPO and BodyGen benefit from priors in terms of final performance and the number of environment steps required to reach a threshold reward. Stackelberg PPO consistently obtains higher final reward and requires fewer steps under both "with prior" and "without prior" conditions. Figure 16 visualizes representative morphologies produced under this setup. While priors accelerate training, it is generally advisable to choose priors that encode broadly useful structural patterns—such as stable support geometries or balanced limb arrangements—rather than narrowly specialized solutions. Such general-purpose priors provide a more flexible foundation for downstream adaptation and reduce the risk of over-constraining the design space.

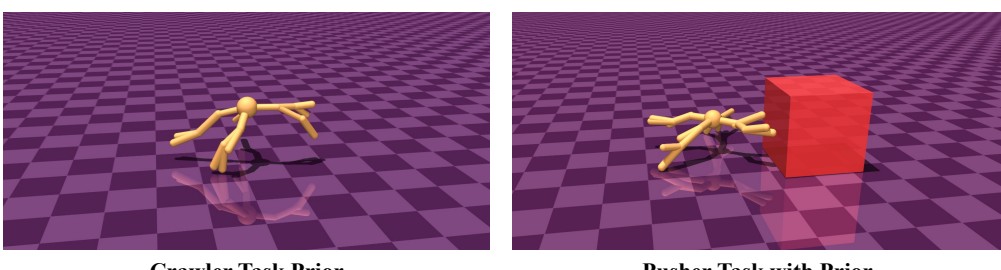

**Crawler Task Prior**         **Pusher Task with Prior**

Figure 16: Cross-task reuse of morphology priors: Crawler prior (left) and the resulting Pusher morphology (right).

Table 16: Performance and sample efficiency in the Pusher task with and without morphology priors.

| Condition | Performance | | Steps to Threshold (2500 Reward) | |
|---|---|---|---|---|
| | Stackelberg PPO (ours) | BodyGen | Stackelberg PPO (ours) | BodyGen |
| With Prior | $\mathbf{4822.59_{\pm 114.32}}$ | $4575.52_{\pm 112.78}$ | $\mathbf{\sim 8M}$ | $\sim 9M$ |
| Without Prior | $\mathbf{3462.77_{\pm 368.09}}$ | $2779.95_{\pm 509.18}$ | $\mathbf{\sim 32M}$ | $\sim 44M$ |

