# OpenReview forum: "Efficient Morphology-Control Co-Design via Stackelberg Proximal Policy Optimization"
_ICLR.cc/2026/Conference — ICLR 2026 Poster_

### Official Review · Reviewer_Md1x · 2025-10-27

**Soundness:** 3
**Presentation:** 2
**Contribution:** 3
**Rating:** 4
**Confidence:** 4

**Summary:**

This paper presents a novel reinforcement learning-based brain-body co-design framework, named Stackelberg Proximal Policy Optimization (Stackelberg PPO). The core innovation lies in formally framing the co-design process as a Phase-Separated Stackelberg Markov Game, where a "leader" agent sequentially constructs the robot morphology, and a "follower" agent learns the control policy for the resulting body. The authors derive a novel Stackelberg policy gradient using a likelihood-ratio trick, which allows the leader to anticipate the follower's adaptation without requiring backpropagation through the non-differentiable simulator. Integrated with PPO's clipping mechanism for stability, the method demonstrates significant improvements in final performance but is kind of computationally expensive.

**Strengths:**

1. The paper contributes a rigorous reformulation, which formulates the brain-body co-design problem as a Stackelberg game.

2. The experiments are comprehensive and convincing.

3. The paper includes thorough ablation studies that validate key design choices, such as the regularization parameter.

**Weaknesses:**

1. The proposed method introduces very significant algorithmic complexity. The requirement to compute or approximate inverse Hessians via conjugate gradient, even with efficient tricks, adds computational overhead and implementation hurdles compared to vanilla PPO. Furthermore, the Stackelberg gradient relies on estimating second-order terms. As the complexity of the morphology and control spaces increases, how does the computational cost of this estimator scale?

2. The paper claims that "Transform2act enables joint training of structure and control without retraining per design. However, the interface
between structure and control remains non-differentiable: the two modules are optimized independently under a shared reward, with no explicit gradient flow between them". I don't quite understand what this sentence means. I think it is differentiable via policy gradient modeling.

3. The authors are encouraged to clarify and analyze differences between their work and other competion-based co-design methods, such as CompetEvo and "The Body is Not a Given".

4. The proposed approach seems to be too complex, relies on a lot of tricks, and can cause a lot of trouble for engineering implementation. It's more like a new PPO algorithm than a specific algorithm for co-design problem. Is it possible to use this algorithm in the neural architecture search problem (which also has a bi-level structure)?

**Questions:**

Please refer to the "Weakness" Section.

---

> ### Author Response · Authors · 2025-11-27
> **Response to Reviewer Md1x (1/2)**
>
> We appreciate the valuable comments and advice from Reviewer Md1x. Below, we provide detailed responses to the reviewer's questions.
>
> > W1: The proposed method introduces very significant algorithmic complexity. The requirement to compute or approximate inverse Hessians via conjugate gradient, even with efficient tricks, adds computational overhead and implementation hurdles compared to vanilla PPO. Furthermore, the Stackelberg gradient relies on estimating second-order terms. As the complexity of the morphology and control spaces increases, how does the computational cost of this estimator scale?
> ### Answer:
>
>
>
> Thank you for this thoughtful question.
> Although Stackelberg PPO appears more complex than PPO, its additional computational cost remains small and scales **linearly** with the sizes of the morphology and control spaces.
>
> This is because we never construct or invert the Hessian explicitly: the conjugate-gradient step uses only inexpensive Hessian–vector products ($\approx$ one backward pass), so the estimator grows **linearly rather than quadratically** with parameter dimensionality.
> Furthermore, rollout generation dominates the overall runtime in co-design, meaning the extra update cost introduced by Stackelberg PPO has minimal impact on wall-clock performance.
>
> As shown in the table below, increasing the morphology and control complexity does not introduce superlinear overhead in our method.
> For reference, the anonymized codebase is available at:
> https://anonymous.4open.science/r/StackelbergPPO-anonymous
>
> Table R5-1: The wall-clock time of our Stackelberg PPO and baseline methods.
> The morphology and control complexity is reflected by the *space size*, which refers to the number of body components. Here, “mean” denotes the average actual size during training, and “max” denotes the maximum allowable size for that task.
>
> | Environment | Space Size (mean) | Space Size (max) |  Stackelberg PPO (ours) | PPO (BodyGen) | Addition Cost |
> | ----------- | ----------------- | -----------------| ----------------------- | ------------- | ------------- |
> | TerrainCrosser  | $4.50_{\pm 0.76}$ | $14$    | $33.88_{\pm 0.42}$          | $27.87_{\pm 1.27}$      | $21.56%$             |
> | Swimmer         | $5.50_{\pm 0.76}$ | $14$    | $32.64_{\pm 0.74}$          | $28.13_{\pm 0.45}$      | $16.03%$             |
> | Cheetah         | $6.57_{\pm 0.90}$ | $14$   | $32.96_{\pm 0.67}$          | $29.52_{\pm 1.03}$      | $11.65%$             |
> | Glider-Hard     | $7.33_{\pm 1.49}$ | $9$    | $32.93_{\pm 0.71}$          | $28.93_{\pm 1.50}$      | $13.83%$             |
> | Walker-Hard     | $8.43_{\pm 1.50}$ | $27$   | $32.54_{\pm 0.62}$          | $30.21_{\pm 1.22}$      | $7.71%$              |
> | Stepper-Hard    | $9.57_{\pm 0.90}$ | $29$    | $32.70_{\pm 1.01}$          | $30.25_{\pm 2.06}$      | $8.10%$              |
> | Pusher          | $14.33_{\pm 4.07}$ | $29$   | $33.41_{\pm 0.82}$          | $29.24_{\pm 1.12}$      | $14.26%$             |
> | Stepper-Regular | $16.40_{\pm 4.69}$ | $29$  | $32.83_{\pm 0.87}$          | $30.17_{\pm 1.41}$      | $8.82%$              |
> | Crawler         | $18.25_{\pm 1.29}$ | $29$  | $33.73_{\pm 0.64}$          | $30.54_{\pm 1.33}$      | $10.45%$             |
>
> ---
> > W2: The paper claims that "Transform2act enables joint training of structure and control without retraining per design. However, the interface between structure and control remains non-differentiable: the two modules are optimized independently under a shared reward, with no explicit gradient flow between them". I don't quite understand what this sentence means. I think it is differentiable via policy gradient modeling.
> ### Answer:
>
> We appreciate this helpful comment.
> Here, “non-differentiable” specifically refers to the fact that the interaction between the morphology and control design procedures is not differentiable.
> This naturally arises in all co-design methods because morphology design involves discrete operations.
> Although the leader module in Transformer2Act [r1] is trained with differentiable updates, gradients cannot flow across the interface from control to morphology, which prevents efficient learning from capturing the coupling between the two procedures.
>
> We have clarified this point in Sec. 2 of the revised paper.

---

> ### Author Response · Authors · 2025-11-27
> **Response to Reviewer Md1x (2/2)**
>
> > W3: The authors are encouraged to clarify and analyze differences between their work and other competition-based co-design methods, such as CompetEvo and "The Body is Not a Given".
> ### Answer:
>
> Thank you for pointing out this line of work.
> CompetEvo [r2] and The Body is Not a Given [r3] address competitive tasks, but their methods do not capture the coupling between the agents during updates, which is a key innovation of our approach.
> Moreover, their methods only optimize attributes under a fixed topology and do not create new topologies.
> We have cited these papers and clarified the differences in Sec. 2 of the revised paper.
>
> ---
> > W4(1/2): The proposed approach seems to be too complex, relies on a lot of tricks, and can cause a lot of trouble for engineering implementation.
>
> ### Answer:
>
>
>
> Although our paper uses the word *“trick”* for several components (which may cause misunderstanding, e.g., the likelihood-ratio clipping trick), **these are actually well-established and mature implementation tools widely used in modern RL frameworks**.
> To clarify:
>
> - *Likelihood-ratio trick*: Also known as the log-derivative trick [r4], a fundamental technique used throughout policy-gradient RL. We use it to derive a newly derived Stackelberg gradient formulation to bypass the non-differentiable problem and obtain a tractable surrogate.
> - *Likelihood-ratio clipping trick*: This is exactly used in the standard PPO objective [r5], grounded in trust-region theory and widely used in modern RL. We use it to stabilize the Stackelberg surrogate gradients, supported by our novel theoretical proof of local equivalence.
> - *Pearlmutter's trick:* This is a built-in feature of modern autodiff frameworks (PyTorch, JAX) and is routinely used in natural-gradient and second-order RL methods.
>
> We use them as tools to derive a new, tractable Stackelberg gradient formulation for a non-differentiable leader–follower setting, grounded in new theoretical foundations.
> To avoid future misunderstandings, we have revised the terminology in the paper to use the standard names for these established techniques.
>
> ---
> > W4(2/2):  It's more like a new PPO algorithm than a specific algorithm for co-design problem. Is it possible to use this algorithm in the neural architecture search problem (which also has a bi-level structure)?
> ### Answer:
>
> Thanks for this insightful comment.
> We hadn't considered this before.
> Yes — the reviewer is correct that our method is not limited to the co-design setting, but can potentially be applied to other bi-level problems such as neural architecture search (NAS).
>
> Our goal here was specifically to progress morphology–control co-design, a problem which clearly manifests the Stackelberg structure, with explicit leader–follower roles and a discrete, non-differentiable interface.
> Extensions to NAS are interesting future work but beyond the scope of this paper.
>
> ### References
>
> [r1] Yuan, Ye et al. “Transform2Act: Learning a Transform-and-Control Policy for Efficient Agent Design.” International Conference on Learning Representations (ICLR 2022).
>
> [r2] Huang, Kangyao et al. “CompetEvo: Towards Morphological Evolution from Competition.” Proceedings of the Thirty-Third International Joint Conference on Artificial Intelligence (IJCAI 2024).
>
> [r3] Banarse, Dylan et al. “The Body Is Not a Given: Joint Agent Policy Learning and Morphology Evolution.” Proceedings of the 18th International Conference on Autonomous Agents and MultiAgent Systems (AAMAS 2019).
>
> [r4] Sutton, Richard S. et al. “Policy Gradient Methods for Reinforcement Learning with Function Approximation.”
> Advances in Neural Information Processing Systems (NeurIPS 2000).
>
> [r5] Schulman, John et al. “Proximal Policy Optimization Algorithms.” arXiv preprint arXiv:1707.06347 (2017).

---

### Official Review · Reviewer_1urx · 2025-10-28

**Soundness:** 2
**Presentation:** 3
**Contribution:** 3
**Rating:** 4
**Confidence:** 3

**Summary:**

This paper makes a step on further improving embodiment co-design based on a recent interesting work BodyGen[1]. How to balance the bi-level optimization of co-design, is indeed a critical point for this area. This paper introduces Stackelberg Game, making this paper a promising solution for this target. The authors provide sufficient experiment results, and achieve some advantages over previous works.

[1] Lu, Haofei, et al. "Bodygen: Advancing towards efficient embodiment co-design." arXiv preprint arXiv:2503.00533 (2025).

**Strengths:**

The experiments cover multiple challenging co-design benchmarks, including both 2D and 3D locomotion as well as complex terrain settings. The comparisons against strong and diverse baselines demonstrate the robustness and generality of the approach. The method is supported by principled mathematical analysis with explicit gradient derivations under the Stackelberg formulation. The theoretical insights connect naturally with the empirical benefits in sample efficiency and stability.

**Weaknesses:**

I hope the author could provide anonymous code, which is strongly encouraged by ICRL. For the remaining weakness, please refer to questions.

**Questions:**

Personally, I have several questions about this paper:
1) As far as I know, the bi-level problem introduced in the previous work, e.g. BodyGen and Transform2Act, their formulation is $max_{morph} (max_{control, morph})$, which already based on that the inner control policy is the best control policy. They just flatten the formulation with a universal controller, in order not to re-train the entire controller from the scratch. However, in the formulation of this paper, aka Stackelberg Game, I don't find any core differences with the old formulation.
2) Following 1), for practical training using modern neural network, I believe that, if you use an end2end training using PPO like BodyGen, the inner universal control policy is always the "best one".
3) You emphasise the adaptiveness of the inner controller according to the outer morph-designer, can you provide any experiments, demonstrating the evolving (especially topology differences) during training, e.g. 1/5, 2/5 of total iterations, what the morphology is like.
4) I am curious about why you do not use all the tasks provided by your baseline, BodyGen & Transform2Act, since you build your entire pipeline on the top of BodyGen. Instead, you add some your personal task for evaluation.

---

> ### Author Response · Authors · 2025-11-27
> **Response to Reviewer 1urx (1/3)**
>
> We sincerely appreciate the reviewer 1urx’s thoughtful and constructive feedback. Below we address the raised questions and concerns in detail.
>
> > W1: I hope the author could provide anonymous code, which is strongly encouraged by ICRL. For the remaining weakness, please refer to questions.
>
> ### Answer:
>
> Thank you for the suggestion. We provide the fully anonymized codebase at:
> [https://anonymous.4open.science/r/StackelbergPPO-anonymous](https://anonymous.4open.science/r/StackelbergPPO-anonymous)
>
> Additional visualizations and videos can be found at: [https://stackelberg-ppo-anonymous.netlify.app/](https://stackelberg-ppo-anonymous.netlify.app/)
>
> ---
> > Q1: As far as I know, the bi-level problem introduced in the previous work, e.g. BodyGen and Transform2Act, their formulation is $max_{morph}(max_{control,morph})$, which already based on that the inner control policy is the best control policy. They just flatten the formulation with a universal controller, in order not to re-train the entire controller from the scratch. However, in the formulation of this paper, aka Stackelberg Game, I don't find any core differences with the old formulation.
>
> ### Answer:
>
> Thanks for this insightful comment.
>
> **I.** In the original problem formulation,
>
>    - Compared to the prior Stackelberg Markov Game (SMG), our paper aims to highlight that the co-design problem is, in fact, **a novel variant** of SMG, characterized by
>  *a new non-differentiable and phase-separated leader–follower interaction*. This is not identified by either prior SMG work nor co-design work.
>
>    - Compared to prior co-design work, we acknowledge that they also adopt a bi-level formulation. The difference lies in our newly introduce **asymmetric objectives**:
>    the leader optimizes both morphology properties and control, while the follower optimizes control only (see the definition of $J^L$ and $J^F$ in eqs. (4–5)).
> In contrast, prior co-design approaches assume that the leader and follower **share a single objective** that considers only control (see Sec. 3 in BodyGen paper [r1]).
>
> **II.** In the algorithmic objective formulation, our method is fundamentally different from that used in prior co-design formulations.
> As the reviewer correctly notes, prior co-design methods such as BodyGen "flatten the formulation" by optimizing a **joint objective**
>
> $$
> \max_{\theta^{L},\theta^{F}} J^L(\theta^{L},\theta^{F}).
> $$
>
> In contrast, our method still adopts the **original bi-level objective**
>
> $$\max_{\theta^L} J^L(\theta^L, \theta^F_*(\theta^L)) \text{ s.t. } \theta^F_*(\theta^L) = \arg\max_{\theta^F} J^F(\theta^L, \theta^F)$$
>
> *Note that the difference between these two objectives is non-trivial and leads to fundamentally different leader-update behavior.*
>
> - Under the joint objective, the follower’s parameters $\theta^{F}$ are treated as independent optimization variables rather than a function of $\theta^{L}$, so the leader gradient only captures the *direct effect of $\theta^{L}$*.
> - Under the bi-level objective, however, the leader gradient additionally includes an *implicit term* that arises from the follower’s optimality condition, capturing that $\theta^{F}_*(\theta^{L})$ is an implicit function of $\theta^{L}$ (see eq. 2 in the paper).
> **Combining the explicit and implicit terms yields the correct Stackelberg update and provides the true steepest-ascent direction for improving overall performance, which is a key innovation of our approach.**

---

> ### Author Response · Authors · 2025-11-27
> **Response to Reviewer 1urx (2/3)**
>
> > Q2: Following 1), for practical training using modern neural network, I believe that, if you use an end2end training using PPO like BodyGen, the inner universal control policy is always the "best one".
> >
> > Q3: You emphasise the adaptiveness of the inner controller according to the outer morph-designer, can you provide any experiments, demonstrating the evolving (especially topology differences) during training, e.g. 1/5, 2/5 of total iterations, what the morphology is like.
>
> ### Answer:
>
> In practice, it is typically intractable for a controller to remain optimal across highly diverse morphologies, due to the limited representational capacity of neural networks.
> We acknowledge that prior work, particularly BodyGen [r1], has made notable progress toward improving universal controllers.
> However, we still observe in practice that the controller continues to adapt as the morphologies evolve during training. We elaborate on this below.
>
> **I. *We first give some background of the update process here:***
>
> In practice, at update $i$, the leader policy $\pi_{\theta^L_{(i)}}^{L}$generates morphologies $s^L_{(i),T}$, and the follower adapts its parameters $\theta^F_{(i),\ast}$ accordingly, yielding the dependency
>
> $$
> \theta^L_{(i)} \Rightarrow s^L_{(i),T} \Rightarrow \theta^F_{(i),\ast},
> $$
>
> Intuitively, as the leader updates and generates new morphologies, the follower tends to adapt to these evolving proposals.
>
> **II. *Below, we highlight the main characteristics observed during training:***
>
>
> 1. **Morphology evolves considerably during training.**
> We provide visualizations at different training stages (e.g., 1/5, 2/5, …) showing that the generated morphologies differ substantially at various milestones. See [https://stackelberg-ppo-anonymous.netlify.app/re4](https://stackelberg-ppo-anonymous.netlify.app/re4) (and Figure 8-9 in the revised paper).
>
> 2. **The controller keeps adapting as training progresses.**
>    As shown in Table R4-1, *the KL divergence of the controller stays at a substantial level throughout training (even in the final stages)*, indicating that the controller continues to adapt throughout.
>
> 3. **Learning a universal controller that is optimal across diverse morphologies is generally intractable in practice.**
>    Table R4-2 summarizes our controlled experiment: a universal controller trained on the *combined* morphology set (A+B) fails to outperform controllers specialized on either A or B. This demonstrates that a single controller can hardly simultaneously optimize performance for diverse morphologies.
>
> Table R4-1: KL divergence of the follower (controller) policy throughout the training process in the Stepper-Regular environment.
>
> |                        | Stage 1/5 | Stage 2/5 | Stage 3/5 | Stage 4/5 | Stage 5/5 |
> | ---------------------- | --------- | --------- | --------- | --------- | --------- |
> | Stackelberg PPO (ours) | $0.0120$  | $0.0131$ |$0.0163$|$0.0212$|$0.0224$|
> | BodyGen                | $0.0103$| $0.0124$ |$0.0151$| $0.0182$|$0.0213$|
>
> Table R4-2: Performance of controllers trained on morphology sets $A$, $B$, and their union $A+B$, evaluated on sets $A$ and $B$, respectively, in the Stepper-Regular environment.  Here, $A$ and $B$ are the morphology sets generated by two different leader policies, respectively.
>
>
> |                                                           | Morphology Set $A$ | Morphology Set $B$ |
> | --------------------------------------------------------- | ------------------ | ------------------ |
> | $\theta_{A,*}^F$ (Trained on morphology set $A$)          |      $\mathbf{5914.04}$     |      $2301.46$     |
> | $\theta_{B,*}^F$  (Trained on morphology set $B$)         |      $2504.59$     |      $\mathbf{6542.22}$     |
> | $\theta_{A+B,*}^F$ (Trained on the morphology set  $A+B$) |      $3632.04$     |      $3835.12$     |

---

> ### Author Response · Authors · 2025-11-27
> **Response to Reviewer 1urx (3/3)**
>
> > Q4: I am curious about why you do not use all the tasks provided by your baseline, BodyGen \& Transform2Act, since you build your entire pipeline on the top of BodyGen. Instead, you add some your personal task for evaluation.
>
> ### Answer:
>
> We clarify that **our original submission includes all the *environment types* used in BodyGen**, and we clarify below to address misunderstandings.
>
> **I.** Due to the page limit, the Swimmer and Walker results were placed in the appendix E.2 of the original submission (both show improvements, and Walker exhibits a particularly strong gain of +16.89%). We have now moved them to the main paper in Figure 2 in the revised paper.
>
> **II.** The only minor difference lies in the Glider and Walker environments: we report results on the *Hard* version, not including the *Regular* and *Medium* ones used in BodyGen.
> *The Hard setting is a more general and challenging version of Regular/Medium*—the only change is a larger allowable topology depth, while all other environment aspects (physics, rewards, controller interface, training protocol) remain identical.
>
> To further address the reviewers’ concerns, we have now added results on the Regular and Medium variants of Glider and Walker below and also in Figure 7 in the revised paper.
> As shown in the table below, Stackelberg PPO outperforms all compared methods across all six environments.
>
> Table R4-3: The scores of the algorithms on additional environments. Note that *Glider-Hard Walker-Hard are already included in the original submission*.
>
>
> |                | Stackelberg PPO (ours) | BodyGen | Transform2Act | NGE | ESS |
> | -------------- | ---------------------- | ------- | --------------- | ---- | ---- |
> | Glider-Regular | $\mathbf{11315.03_{\pm 112.62}}$ | $10682.11_{\pm 193.71}$ | $6640.04_{\pm 417.61}$ | $1369.48_{\pm 426.83}$ | $361.14_{\pm 141.73}$ |
> | Glider-Medium  | $\mathbf{12043.40_{\pm 307.14}}$ | $11458.92_{\pm 289.58}$ | $5298.47_{\pm 523.86}$ | $1498.86_{\pm 628.30}$ | $362.44_{\pm 72.26}$ |
> | Glider-Hard    | $\mathbf{12414.50_{\pm 498.53}}$ | $11049.95_{\pm 468.44}$ | $6120.62_{\pm 1086.62}$ | $2081.25_{\pm 348.17}$ | $541.55_{\pm 107.56}$ |
> | Walker-Regular | $\mathbf{12681.65_{\pm 324.17}}$ | $12050.73_{\pm 206.10}$ | $8237.20_{\pm 761.36}$ | $1339.91_{\pm 544.08}$ | $305.38_{\pm 61.14}$ |
> | Walker-Medium  | $\mathbf{12620.63_{\pm 510.91}}$ | $11888.84_{\pm 376.96}$ | $6021.85_{\pm 494.09}$ | $2400.79_{\pm 393.71}$ | $388.26_{\pm 76.73}$ |
> | Walker-Hard    | $\mathbf{13612.32_{\pm 501.26}}$ | $11645.89_{\pm 797.77}$ | $4420.63_{\pm 267.48}$ | $1504.55_{\pm 553.15}$ | $636.03_{\pm 125.74}$ |
>
> **III.**
> In the original submission, we introduced two additional tasks aiming to better evaluate our method under more challenging and diagnostic conditions.
> For example, in Stepper-Regular and Stepper-Hard, the agent must climb stair-like structures rather than perform locomotion on flat terrain, providing a more challenging test of morphology–control co-design.
> Furthermore, to address concerns of other reviewers,  we further include additional challenging tasks, such as payload-carrying and manipulation tasks, in the revised paper.
>
> References
>
> [r1] Lu, Haofei, et al. “BodyGen: Advancing Towards Efficient Embodiment Co-Design.” The International Conference on Learning Representations (ICLR 2025).

---

> ### Comment · Reviewer_1urx · 2025-11-28
> **Thank you for your response. I will raise my score.**
>
> Thank you for your response. I will raise my score.

---

### Official Review · Reviewer_vMFr · 2025-10-29

**Soundness:** 2
**Presentation:** 3
**Contribution:** 2
**Rating:** 4
**Confidence:** 4

**Summary:**

The paper reframes morphology–control co-design as a phase-separated Stackelberg game and derives a likelihood-ratio–based Stackelberg PPO that captures follower adaptation without back-propagating through a non-differentiable interface. The method claims better sample efficiency and final reward across six MuJoCo tasks versus PPO-based and evolutionary baselines, with theoretical support (local equivalence surrogates) and stabilizing design choices (PPO clipping, Fisher metric).

**Strengths:**

1. **Clear problem framing for non-differentiable interfaces.** The phase-separated SMG formalization precisely matches co-design realities (discrete edits then control), addressing why prior Stackelberg methods fail here. Treating co-design as leader–follower nicely explains why joint training can wobble or fail.

2. **Technically neat gradient path.** The trajectory likelihood-ratio surrogate for the cross-derivative (Theorem 1) avoids differentiating through morphology transitions and yields local equivalence to true Stackelberg gradients. This is practical for implementation, easy for adoption with downstream ideas.

3. **Stability machinery.** PPO-style clipping and Fisher (natural-gradient) Hessian approximation are well-motivated for practical computation. The $\lambda$ sweep and Fisher vs. analytic Hessian give me a bit of insight into stability.

**Weaknesses:**

1. There are concerns about the experiment outlines:
- Prior Stackelberg RL (e.g., Stackelberg actor-critic / policy-gradient) is acknowledged, but there’s no controlled ablation that swaps in those estimators under the same phase-separated setting to isolate what PPO-clipping contributes vs. the SID term itself.
- Only four seeds are used “due to cost”; yet the authors claim “+22.1% average, +31.9% on 3D”, which needs more statistical significance. These environments are not that complicated to run at least 10 seeds.
- The paper argues for efficiency, but training still takes ~30 hours per model on an A100 + 10 CPU cores. There should be implementation optimization, or the absence of wall-clock comparisons to ES baselines (which can parallelize) leaves practical efficiency unclear.
- Most environments reward forward velocity (with optional small effort penalty), so improvements might partially reflect optimization for locomotion speed rather than broader co-design quality (e.g., robustness, energy, material constraints).

2. On the theory side, surrogates are locally equivalent; the practical bias/variance of the coupled gradient estimator (esp. through long leader horizons) is not deeply characterized.

**Questions:**

1. Please address the concerns in the Weakness section.

2. Estimator properties. What are the empirical bias/variance characteristics of the follower-implicit gradient term (eqs. 6–9) as the leader horizon grows? Any variance-reduction beyond standard advantage baselines?

3. Local equivalence radius. How sensitive are results to step size, i.e., when does local surrogate equivalence break, and how does PPO clipping mitigate that in practice? Provide KL traces or constraint violations.

4. Reward shaping sensitivity. How do results change if you (i) increase control-effort penalties, (ii) impose morphology complexity/material costs as leader rewards, or (iii) evaluate robustness (disturbances/terrain noise)?

5. Generality. Does Stackelberg PPO extend beyond locomotion (e.g., manipulation, multi-objective design)? Any results on unseen tasks or morphology priors to show transfer?

---

> ### Author Response · Authors · 2025-11-27
> **Response to Reviewer vMFr (1/7)**
>
> We sincerely appreciate Reviewer vMFr's constructive comments. Below we address the questions and concerns.
>
> > W1: Prior Stackelberg RL (e.g., Stackelberg actor-critic / policy-gradient) is acknowledged, but there’s no controlled ablation that swaps in those estimators under the same phase-separated setting to isolate what PPO-clipping contributes vs. the SID term itself.
>
> ### Answer:
>
> **Prior Stackelberg RL methods (e.g., Stackelberg MADDPG [r1]) are fundamentally inapplicable to our setting with non-differentiable leader–follower interfaces**, as they all rely on differentiable interfaces to enable direct backpropagation.
> In contrast, we introduce a new approach that performs Stackelberg Implicit Differentiation (SID) **without** needing differentiable interfaces, thereby enabling SID in settings *where previous methods simply cannot operate*.
>
> Since prior SID derivations cannot be applied in our non-differentiable setting, we instead perform a controlled ablation that isolates the effects of our new SID estimator and PPO clipping by evaluating the following variants:
>
> 1. PPO clipping + our new SID estimator,
> 2. PPO-*only* (without SID), which is already included in our submission as the baseline BodyGen, and
> 3. SID-*only* (without PPO clipping),
>
> The results in the table below show that **both our new SID estimator and PPO clipping provide clear performance gains**.
> We have included these new results in Appendix E.3 of the revised paper.
>
> Table R3-1: Ablation studies on the components of *our new SID estimator* and *PPO clipping*, under the same phase-separated, non-differentiable setting.
>
> | Environment | SID+PPO (full) | PPO-*only* (without SID) | SID-*only* (without PPO clipping) |
> | ----------- | -------------- | ------------------------ | --------------------------------- |
> |Stepper-Regular|$\mathbf{7215.20_{\pm 449.02}}$|$4685.94_{\pm 845.23}$|$1257.33_{\pm 530.25}$|
> |Crawler| $\mathbf{11047.90_{\pm 126.20}}$|$9098.72_{\pm 558.26}$|$35.77_{\pm 12.25}$|
> |Cheetah| $\mathbf{13514.94_{\pm 653.62}}$|$11575.87_{\pm 640.65}$|$472.89_{\pm 77.40}$|
> |Glider| $\mathbf{12414.50_{\pm 498.53}}$|$11049.95_{\pm 468.44}$|$566.81_{\pm 89.96}$|
>
> ---
> > W2: Only four seeds are used “due to cost”; yet the authors claim “+22.1% average, +31.9% on 3D”, which needs more statistical significance. These environments are not that complicated to run at least 10 seeds.
>
> ### Answer:
>
>
>
> **Our experimental design strictly follows the protocol of the prior co-design work BodyGen [r2], which also uses 4 random seeds (NGE paper [r3] uses 1).**
> We fully agree that using more seeds would further strengthen statistical significance. However, the computational cost of running 10 seeds is far higher than the reviewer may expect.
> **Running all GPU-required experiments with 10 random seeds, including the reviewer-requested studies, would require at least 675 A100 GPU-days** (each baseline and our method requires at least 30 GPU hours per run per environment), which is well beyond our computational budget.
>
> To nevertheless address the reviewer’s concern, **we have been able to increase the number of random seeds from 4 to 7** (BodyGen uses 4, and NGE uses only 1).
> We hope that the reviewer will find this a satisfactory middle ground.
>
> The updated 7-seed results remain consistent with our original findings, again showing “+20.66 % on average, and +32.02 % on the complex 3D tasks.”
> We have updated the paper in Section 6.1 accordingly.

---

> ### Author Response · Authors · 2025-11-27
> **Response to Reviewer vMFr (2/7)**
>
> > W3: The paper argues for efficiency, but training still takes ~30 hours per model on an A100 + 10 CPU cores. There should be implementation optimization, or the absence of wall-clock comparisons to ES baselines (which can parallelize) leaves practical efficiency unclear.
>
> ### Answer:
>
> We clarify that the primary notion of *efficiency* in our work is **sample efficiency**, rather than *wall-clock efficiency*.
>
> To address the reviewer’s concern about running time, we summarize both wall-clock training time and sample-efficiency results in the tables below (for training curves, see Fig. 2 in the original submission).
>
> - *Compared to BodyGen*, our method achieves substantially better sample efficiency by requiring **-39%** fewer samples to reach the performance threshold while also obtaining **+20.66%** higher final scores.
> In terms of wall-clock time, the difference between the two methods is modest (+13%), keeping the overall training cost comparable.
>
> - *Compared to ES baselines*, although ESS attains shorter wall-clock time using *6×* more CPU cores (64 cores), its performance is extremely poor, achieving only a **0.16** fraction of our method’s performance.
>
> Overall, our method delivers strong sample efficiency and performance while remaining competitive in training wall-clock time.
> We have updated these results in Appendix E.4 in the revised paper.
>
> Table R3-2: Sample efficiency comparison: number of samples required to achieve the performance threshold.
>
> | Environment      | Threshold | Stackelberg PPO (ours) | BodyGen | Transform2Act | NGE | ESS |
> |------------------|-----------|--------------------------|---------|----------------|-----|-----|
> | Crawler          | $\mathit{9000}$    | $\mathbf{25.8}$                   | $47.2$  | $\infty$       | $\infty$ | $\infty$ |
> | Cheetah          | $\mathit{11000}$   | $\mathbf{19.2}$                   | $42.1$  | $\infty$       | $\infty$ | $\infty$ |
> | Swimmer          | $\mathit{1200}$    | $\mathbf{14.8}$                    | $17.0$   | $\infty$         | $\infty$ | $\infty$ |
> | Walker-Hard      | $\mathit{10000}$   | $\mathbf{18.1}$                   | $30.3$  | $\infty$       | $\infty$ | $\infty$ |
> | Glider-Hard      | $\mathit{11000}$   | $\mathbf{23.6}$                   | $49.7$  | $\infty$       | $\infty$ | $\infty$ |
> | TerrainCrosser   | $\mathit{3500}$    | $\mathbf{23.9}$                   | $33.8$  | $\infty$       | $\infty$ | $\infty$ |
> | Pusher           | $\mathit{2500}$    | $\mathbf{29.3}$                   | $39.1$  | $\infty$       | $\infty$ | $\infty$ |
> | Stepper-Regular  | $\mathit{4500}$    | $\mathbf{18.5}$                   | $40.4$  | $\infty$       | $\infty$ | $\infty$ |
> | Stepper-Hard     | $\mathit{4500}$    | $\mathbf{27.2}$                   | $43.1$| $\infty$       | $\infty$ | $\infty$ |
>
> Table R3-3: Wall-clock training time, under 10 CPU cores and A100 GPU averaged over 9 environments. The ES baseline (NGE) is also shown with 64 CPU cores to illustrate its parallelization advantage.
>
> |                | Stackelberg PPO(Ours , 10 CPU cores) | BodyGen (PPO, 10 CPU cores) | NGE (ES-based, 10 CPU cores) | NGE (ES-based, 64 CPU cores) |
> | -------------- | ------------------------------------ | --------------------------- | ---------------------------- | ---------------------------- |
> | Wallclock Time (hours) | $33.07_{\pm 0.49} $            | $29.43_{\pm 0.97}$    | $45.16_{\pm 3.72}$     | $13.52_{\pm 1.52}$     |

---

> ### Author Response · Authors · 2025-11-27
> **Response to Reviewer vMFr (3/7)**
>
> > W4: Most environments reward forward velocity (with optional small effort penalty), so improvements might partially reflect optimization for locomotion speed rather than broader co-design quality (e.g., robustness, energy, material constraints).
> >
> > Q4: Reward shaping sensitivity. How do results change if you (i) increase control-effort penalties, (ii) impose morphology complexity/material costs as leader rewards, or (iii) evaluate robustness (disturbances/terrain noise)?
>
> ### Answer:
>
> We thank the reviewer for the insightful suggestion, which greatly expands our analysis and helps better highlight the potential of our method.
> We now clarify these points and provide additional supporting results.
>
> **I. Existing Experiments**
>
> We would like to clarify that **most co-design qualities raised by the reviewer are already included in our original experiments** to some extent. In particular, they are reflected as follows:
>
> - *Robustness* is already inherently captured by the current evaluation protocol: each morphology–controller pair is evaluated over multiple rollouts, so robust designs achieve higher averages while non-robust ones yield lower.
> - *Energy usage* is already accounted for through a small effort penalty, consistent with the reviewer’s understanding (see Appendix C in the original submission).
> - *Material constraints* are also captured through hard-coded limits on morphology-editing actions, such as bounds on limb length and topology depth (see Appendix E.2 in the original submission).
> Regarding the reviewer’s concern about broader measures of morphological complexity (e.g., encouraging minimal node count), we address this point in II (ii) below.
>
>
> **II. Additional Experiments**
>
> Below, we address the reviewer’s additional concerns regarding reward shaping and broader co-design quality.
>
> **(i) Increase control-effort penalties**
>
> We further evaluate performance under stronger control-effort penalty coefficients (0.001, 0.01, 0.1), extending the original 0.0001 used in our main experiments.
> Table R3-4 show the results.
> The effects of larger penalties can be seen from the following three aspects:
>
> - *Effect in Performance (velocity reward):* As the penalty increases, both our method and BodyGen experience reduced performance, but our method consistently shows a smaller degradation across all settings.
>
> - *Effect in Control effort:* Larger penalties improve control efficiency for both methods, which is reflected in higher penalty-related rewards.
>
> - *Effect in Morphology–control:* Under stronger penalties, the co-designed solutions shift toward shorter, thicker, and more symmetric limbs, accompanied by a low-torque, conservative gait characteristic of energy-efficient locomotion. Please refer to [https://stackelberg-ppo-anonymous.netlify.app/re3/4i](https://stackelberg-ppo-anonymous.netlify.app/re3/4i) for examples of the generated morphologies.
>
> Table R3-4: Performance (velocity reward) and control effort penalty under different control-effort penalty coefficients.
>
>
>
> | Penalty Coef. | Performance |          | Control Effort Penalty |          |
> |---------------|-------------|----------|--------------|----------|
> |               | Stackelberg PPO (ours) | BodyGen  | Stackelberg PPO (ours) | BodyGen |
> | $0.0001$ | $\mathbf{11047.90_{\pm 126.20}}$ | $9098.72_{\pm 558.26}$ | $5631.42_{\pm 674.03}$ | $2745.84_{\pm 284.55}$ |
> | $0.001$  | $\mathbf{10191.15_{\pm 371.81}}$ | $7501.61_{\pm 671.33}$ | $5342.16_{\pm 584.25}$ | $5948.16_{\pm 254.84}$ |
> | $0.01$   | $\mathbf{9853.19_{\pm 229.37}}$  | $8304.00_{\pm 497.56}$ | $1582.48_{\pm 697.61}$ | $468.12_{\pm 516.33}$ |
> | $0.1$    | $\mathbf{10585.25_{\pm 146.80}}$ | $8974.48_{\pm 574.29}$ | $25.50_{\pm 22.27}$ | $26.34_{\pm 23.64}$ |
>
> [continued in the next panel]

---

> ### Author Response · Authors · 2025-11-27
> **Response to Reviewer vMFr (4/7)**
>
> **(ii) Morphology complexity/material costs as leader rewards**
>
> We incorporate these properties by modifying the leader reward: morphology complexity is measured by the number of nodes in the topology, and material cost is defined by the total mass.
>
> The table below summarizes the resulting performance and morphology characteristics under different penalty coefficients. The trends are consistent with those observed in the control-effort experiments in (i): our method consistently achieves better reward–cost tradeoffs across all penalty levels.
> With a larger penalty coefficient, the evolved morphologies become more compact and structurally simpler, see [https://stackelberg-ppo-anonymous.netlify.app/re3/4ii](https://stackelberg-ppo-anonymous.netlify.app/re3/4ii) for the visualizations.
>
> Table R3-5: Performance, morphology complexity, and material cost under different penalty coefficients.
>
>
>
> | Penalty Coef. | Performance |          | Morphology Complexity |          | Material Cost |          |
> |---------------|-------------|----------|------------------------|----------|----------------|----------|
> |               | Stackelberg PPO (ours) | BodyGen  | Stackelberg PPO (ours) | BodyGen (baseline) | Stackelberg PPO (ours) | BodyGen  |
> | $0$   | $\mathbf{11047.90_{\pm126.20}}$ | $9098.72_{\pm558.26}$ | $16.40_{\pm2.45}$ | $13.67_{\pm2.08}$ | $1.71_{\pm0.23}$ | $1.57_{\pm0.32}$ |
> | $1$   | $\mathbf{7892.93_{\pm349.84}}$  | $6531.37_{\pm437.26}$ | $13.67_{\pm2.03}$ | $9.67_{\pm1.61}$  | $1.59_{\pm0.18}$ | $1.32_{\pm0.16}$ |
> | $10$  | $\mathbf{6825.47_{\pm303.09}}$  | $5372.10_{\pm364.79}$ | $8.25_{\pm0.91}$  | $7.50_{\pm1.24}$  | $0.94_{\pm0.08}$ | $0.93_{\pm0.10}$ |
>
>
>
> **(iii) Robustness evaluation**
>
> We evaluate robustness under two settings: random *disturbance forces* applied to the root body at every control step, and *terrain friction noise* created by randomly varying the ground’s friction in each episode.
> As shown in the tables below, our method consistently demonstrates substantially higher robustness across all disturbance levels. For example, when increasing the disturbance force from 2 N to 6 N, BodyGen drops by **–59.57%**, whereas our method drops by only **–5.91%**.
>
> Table R3-6: Performance under various levels of *disturbance forces*.
>
> | Level   | Stackelberg PPO (ours)  | BodyGen      |
> | ------- | ----------------------- | ---------------------- |
> | $2.0$ N | $\mathbf{11557.31_{\pm 124.68}}$ | $6963.05_{\pm 450.48}$ |
> | $4.0$ N | $\mathbf{11290.13_{\pm 164.54}}$ | $4621.82_{\pm 597.71}$ |
> | $6.0$ N | $\mathbf{10875.23_{\pm 250.97}}$ | $2816.16_{\pm 857.83}$ |
>
> Table R3-7: Performance under various levels of *terrain friction noise*.
>
> | Level   | Stackelberg PPO (ours)  | BodyGen      |
> | ------- | ----------------------- | ---------------------- |
> | $30%$  | $\mathbf{11424.66_{\pm 112.08}}$ | $7326.04_{\pm 421.73}$ |
> | $50%$  | $\mathbf{11333.43_{\pm 141.42}}$ | $6795.55_{\pm 493.31}$ |
> | $70%$  | $\mathbf{10892.09_{\pm 149.72}}$ | $5062.85_{\pm 579.66}$ |
>
> We have added full experimental details and results in Appendix E.6 of the revised paper.

---

> ### Author Response · Authors · 2025-11-27
> **Response to Reviewer vMFr (5/7)**
>
> > W5: On the theory side, surrogates are locally equivalent; the practical bias/variance of the coupled gradient estimator (esp. through long leader horizons) is not deeply characterized.
> >
> > Q2: Estimator properties. What are the empirical bias/variance characteristics of the follower-implicit gradient term (eqs. 6–9) as the leader horizon grows? Any variance-reduction beyond standard advantage baselines?
>
> ### Answer:
>
>
>
> Thanks for this insightful comment.
>
> We report the bias and variance of our Stackelberg surrogate gradient in eq. (9) (incorporating the terms from eqs. 6 to 8).
> We also compare against BodyGen, whose gradient estimator omits the implicit term (step 2 in eq. 9), making it theoretically biased.
>
> The tables below reveal two key observations:
>
> - As the leader horizon increases, we do not observe a clear upward trend in the empirical bias or variance.
>   This is because the estimator operates as an average, rather than a cumulative sum, over the leader–follower terms, which is consistent with the equivalent state-visitation form of eqs. (6–8).
>     (For additional detailed explanation, please refer to our response to 1st Reviewer B8Qz,W2)
>
> - Our estimator consistently shows lower empirical bias compared to BodyGen, which aligns with the theoretical difference that BodyGen does not include the implicit term.
>
> Table R3-8: Empirical *bias* of different gradient estimators as the leader horizon $T$ increases.
>
> | Leader Horizon $T$ | Stackelberg PPO (ours) | BodyGen |
> | -------------------- | ---------------------- | ------- |
> | $3$                   | $0.72$                  | $1.14$    |
> | $5$                   | $0.67$                  | $1.57$    |
> | $7$                   | $0.97$                  | $1.92$   |
> | $9$                   | $0.74$                  | $1.59$    |
> | $11$                   | $0.88$                   | $2.17$    |
>
> Table R3-9: Empirical *variance* of different gradient estimators as the leader horizon $T$ increases.
>
> | Leader Horizon $T$ | Stackelberg PPO (ours) | BodyGen |
> | -------------------- | ---------------------- | ------- |
> | $3$                   | $0.91$                  | $0.87$   |
> | $5$                   | $0.89$                  |$0.78$   |
> | $7$                   | $1.23$                  | $1.27$   |
> | $9$                    | $0.94$                  | $0.91$   |
> | $11$                   | $1.13$                   | $1.11$   |
>
> *Experimental Details about Bias and Variance Calculation:*
> The empirical *bias* is computed as the mean absolute difference between the estimator and the reference “ground-truth” gradient, and the empirical *variance* is computed as the sample variance of the estimator.
> The reference gradient is obtained using a high-precision Monte Carlo estimate averaged over 1,024 rollouts.
> All experiments for both our method and BodyGen use the same follower-adaptation budget, the same initialization, and identical rollout collections to ensure a fair comparison.

---

> ### Author Response · Authors · 2025-11-27
> **Response to Reviewer vMFr (6/7)**
>
> > Q3: Local equivalence radius. How sensitive are results to step size, i.e., when does local surrogate equivalence break, and how does PPO clipping mitigate that in practice? Provide KL traces or constraint violations.
>
> ### Answer:
>
> Similar to the performance lower-bound theory in trust region policy optimization [r4], *the surrogate becomes less faithful to the true objective as the updated policy $\pi_\theta$ drifts farther from the old policy $\pi_{\theta_o}$*.
> This breakdown occurs when the policy update becomes too large.PPO clipping mitigates this issue by limiting the size of policy updates, bounding the likelihood ratio $\frac{\pi_\theta(a\mid s)}{\pi_{\theta_o}(a\mid s)}$ within $[1-\epsilon,1+\epsilon]$.
>
> The table below reports the results under different values of the clipping parameter $\epsilon$.
> Reward and KL-divergence curves are available at: [https://stackelberg-ppo-anonymous.netlify.app/re3/3](https://stackelberg-ppo-anonymous.netlify.app/re3/3).
> As shown in the results, increasing the clipping parameter $\epsilon$, which loosens the update restriction, leads to larger KL divergence.
> Performance peaks at $\epsilon = 0.4$. Overly small $\epsilon$ values (e.g., 0.1) hinder improvement, while overly large values (e.g., 0.8, or no clipping) cause the surrogate gradient to deviate, leading to degraded performance.
>
> Table R3-10: Performance, likelihood-ratio constraint violations, and KL divergences under different clipping parameters $\epsilon$.
>
> | Clipping Parameter | Score                        | Likelihood Ratio Constraint Violations (**%**) | Average KL Divergences           |
> | ------------------ | ---------------------------- | ---------------------------------------------- | --------------------------------- |
> | $\epsilon=0.1$     | $4934.52_{\pm 646.40}$      | $14.82_{\pm 0.55}$       | $0.0030_{\pm 0.0006}$            |
> | $\epsilon=0.2$     | $7215.20_{\pm 449.02}$      | $13.39_{\pm 0.83}$       | $0.0196_{\pm 0.0025}$             |
> | $\epsilon=0.4$     | $\mathbf{7907.01_{\pm 208.02}}$       | $9.92_{\pm 0.94}$        | $0.0343_{\pm 0.0153}$             |
> | $\epsilon=0.6$     | $4778.18_{\pm 407.84}$      | $9.10_{\pm 1.30}$        | $0.0665_{\pm 0.0188}$             |
> | $\epsilon=0.8$     | $2656.92_{\pm 503.93}$      | $7.02_{\pm 0.81}$        | $0.1340_{\pm 0.0388}$             |
> | NO Clipping        | $1233.26_{\pm 443.98}$       | $0$                      | $1.7726_{\pm 0.1539}$             |

---

> ### Author Response · Authors · 2025-11-27
> **Response to Reviewer vMFr (7/7)**
>
> > Q5: Generality. Does Stackelberg PPO extend beyond locomotion (e.g., manipulation, multi-objective design)? Any results on unseen tasks or morphology priors to show transfer?
>
> ### Answer:
>
> We thank the reviewer for the thoughtful question.
> We conducted additional experiments and found that our method generalizes well to the more challenging settings raised by the reviewer, including manipulation, multi-objective design, generalization to unseen tasks, and training with morphology priors, and it consistently outperforms the strongest benchmark baseline in these settings.
> Details are provided below.
>
> **(i) Manipulation**
>
> We further evaluate the methods on a new task, Pusher, where the agent must move a cube forward while avoiding sideways deviation.
> The table below indicates that our method outperforms BodyGen on this task.
>
> We surprisingly observe an emergent pattern of the generated morphologies: *our method often evolves morphologies with a natural functional separation—the front section specializes in pushing the box, while the rear section provides propulsion.* Representative morphologies are shown in Figure(i) in [https://stackelberg-ppo-anonymous.netlify.app/re3/5](https://stackelberg-ppo-anonymous.netlify.app/re3/5)
> This behavior is not typically observed in the baseline.
>
> Table R3-11: Performance in the Pusher environment, averaged over 7 random seeds.
>
> |             | Stackelberg PPO (ours) | BodyGen  |
> | ----------- | ----------------------- | -------------------- |
> | Pusher | $\mathbf{3462.77_{\pm 368.09}}$    | $2779.95_{\pm 509.18}$ |
>
> **(ii) Multi-objective design**
>
> We consider a multi-objective task in which the agent must move at high speed while carrying a payload.
> We train the models under random payloads and then evaluate them across various payload levels.
> As shown in the table below, our method achieves substantially higher performance than the baseline.
>
> Our response to W4 above also includes additional results on other multi-objective factors, such as control effort, morphological complexity, and material cost.
>
> Table R3-12: Performance in the Crawler-with-payload environment under various payload levels, averaged over 7 random seeds.
>
>    | Payload Weight | Stackelberg PPO (ours) | BodyGen      |
>    | ----------- | ----------------------- | ---------------------- |
>    | $0.2$ kg    | $\mathbf{8675.08_{\pm 286.42}}$  | $5186.31_{\pm 659.87}$ |
>    | $0.4$ kg    | $\mathbf{6966.34_{\pm 473.43}}$  | $5116.54_{\pm 546.55}$ |
>    | $0.6$ kg    | $\mathbf{7347.46_{\pm 478.01}}$  | $4523.89_{\pm 536.99}$ |
>
>
> **(iii) Unseen tasks**
>
> We evaluate generalization by training the policies on flat terrain and testing them in environments where the agent must navigate around multiple square obstacles.
> Our method achieves higher performance than BodyGen, indicating that it generalizes better to the unseen task.
>
> Table R3-13: Performance in the Crawler unseen-obstacle environment under various obstacle densities, averaged over 7 random seeds.
>
> |          Obstacle Type        |             | Stackelberg PPO (ours) | BodyGen    |
> | --------------- | ---------------------------- | ----------------------- | -------------------- |
> | Sparse Obstacle | $16$ m ($\sim4$×robot width) | $\mathbf{1790.45_{\pm 161.77}}$    | $1061.55_{\pm 228.40}$ |
> | Dense Obstacle  | $8$ m ($\sim2$×robot width)  | $\mathbf{1698.52_{\pm 733.02}}$    | $1007.21_{\pm 157.42}$ |
>
> **(iv) Morphology priors**
>
> We train the models on the Pusher task using morphology priors obtained from the Crawler task.
> As shown in the table below, both our method and BodyGen benefit from these priors in terms of performance and sample efficiency, but our method consistently outperforms the baseline.
>
> Table R3-14: Comparison of performance and sample efficiency with and without priors, averaged over 7 random seeds.
>
> | Condition     | Performance                     |                        |  Steps to Threshold |                     |
> | ------------- | ------------------------------- | ---------------------- | --------------------------- | ------------------- |
> |               | Stackelberg PPO (ours)          | BodyGen                | Stackelberg PPO (ours)      | BodyGen             |
> | With Prior    | $\mathbf{4822.59_{\pm 114.32}}$ | $4575.52_{\pm 112.78}$ | $\mathbf{\sim 8\text{ M}}$  | ${\sim 9\text{ M}}$ |
> | Without Prior | $\mathbf{3462.77_{\pm 368.09}}$ | $2779.95_{\pm 509.18}$ | $\mathbf{\sim 32\text{ M}}$ | $\sim 44\text{ M}$  |
>
> We have included the full experimental details and results in Appendix E.6 and E.7 in the revised paper.
>
> ### References
>
> [r1] Yang, et al. “Stackelberg Games for Learning Emergent Behaviors during Competitive Autocurricula.” ICRA 2023
>
> [r2] Lu, et al. “BodyGen: Advancing Towards Efficient Embodiment Co-Design.”ICLR 2025
>
> [r3] Wang, et al. “Neural Graph Evolution: Towards Efficient Automatic Robot Design.”ICLR 2019
>
> [r4] Schulman, et al. “Trust Region Policy Optimization.”ICML 2015

---

### Official Review · Reviewer_ry9P · 2025-11-01

**Soundness:** 3
**Presentation:** 3
**Contribution:** 2
**Rating:** 6
**Confidence:** 3

**Summary:**

This work proposes efficient morphology control co-design via Stackelberg PPO. It formulates morphology–control co-design as a phase-separated Stackelberg Markov Game, where the leader is the morphology designer and the follower the policy controller. The Stackelberg PPO is a policy-gradient method that uses an implicit-differentiation surrogate (likelihood–ratio–based) across a non-differentiable leader–follower interface, plus Fisher-information preconditioning and PPO-style gradient clipping. On MuJoCo co-design benchmarks (Crawler, Cheetah, Glider, TerrainCrosser) and two new stair tasks, the method reports better stability and final reward than PPO-based co-design and evolutionary baselines (e.g., +22% on average, ~+32% on harder 3D tasks).

**Strengths:**

- This paper proposes a novel formulation of the morphology co-design problem, where the gradient from the morphology design phase is not differentiable. The Stackelberg game formulation is interesting and well-motivated. This avoids ad-hoc joint updates and gives a clean bilevel control–morphology structure.
    - The derivation for a Phase-Separated Stackelberg Markov Game’s **likelihood-ratio surrogate** with a **Fisher preconditioner** creates a tractable gradient signal for the leader without assuming differentiability of morphology parameters or unrolling large follower optimization loops. This is a **meaningful algorithmic advance** over evolutionary search or unrolled gradient-based bilevel RL, which are expensive and brittle.
- The multi-step morphology evolution, compared to prior art (e.g. Transform2Act) is a good insight and brings in notable improvement.
- Training efficiency improvement is notably higher than prior art as the morphology editors are now more informed with controller training results.

**Weaknesses:**

- Most evaluations reduce to forward-velocity rewards on stylized locomotion creatures; even the new stair tasks keep the same simple progress reward. This makes it hard to assess whether Stackelberg coupling helps with *real* co-design constraints (payload, torque limits, manufacturability, sensor placement, power, robustness).
- The λ/Fisher ablations are informative, but I’d like visibility into **data efficiency**: how many follower steps are actually saved vs PPO co-design? Since the thesis is “less re-optimization,” show sample-complexity curves for controller updates per viable morphology. (There is a follower-sampling hyperparameter, but not a direct “control-retraining cost” metric.)

**Questions:**

- How sensitive are results to morphology editing horizon T?
- Does the follower avoid full re-training when morphologies change significantly, or does the advantage mostly come from better leader gradients? Any evidence of faster controller adaptation?

---

> ### Author Response · Authors · 2025-11-27
> **Response to Reviewer ry9P (1/3)**
>
> We sincerely thank reviewer ry9P for the valuable comments. Below we address the questions and concerns.
>
> > W1: Most evaluations reduce to forward-velocity rewards on stylized locomotion creatures; even the new stair tasks keep the same simple progress reward. This makes it hard to assess whether Stackelberg coupling helps with real co-design constraints (payload, torque limits, manufacturability, sensor placement, power, robustness).
>
> ### Answer:
>
> Thank you for the insightful suggestion.
>
> **I. Existing Experiments**
>
> We would like to clarify that **many co-design qualities raised by the reviewer are already included in our original experiments** to some extent. In particular, they are reflected as follows:
>
> - *Torque limits* are already modeled by a hard-coded bound on the “allowable torque” attribute in the morphology design action.
> - *Manufacturability* is also captured through hard-coded limits on morphology-editing actions, such as bounds on limb length and topology depth (see Appendix E.2 in the original submission).
> - *Power constraint* is already accounted for through a small effort penalty (see Appendix C in the original submission).
> - *Robustness* is already inherently captured by the current evaluation protocol: each morphology–controller pair is evaluated over multiple rollouts, so robust designs achieve higher averages while non-robust ones yield lower.
>
> **II. Additional Experiments**
>
> Below, we address the reviewer’s additional concerns regarding experiments under broader co-design constraints.
> *Although such analyses are not typically contained in prior co-design work*, we sincerely appreciate the reviewer’s suggestions, as they allow us to further illustrate the strengths of our method and improve the completeness of the paper.
> The corresponding visualizations of the tasks and generated morphologies are available at:
> [https://stackelberg-ppo-anonymous.netlify.app/re2](https://stackelberg-ppo-anonymous.netlify.app/re2)
>
> **(i) Payload-Carrying Task**
>
> We design a new task in which the agent must move at high speed while carrying a payload.
> We train the models under random payloads and then evaluate them across various payload levels.
> As shown in the table below, our method achieves substantially higher performance than the baseline.
>
> Table R2-1: Performance in the Crawler-with-payload environment under various payload levels, averaged over 7 random seeds.
>
>    | Payload Weight | Stackelberg PPO (ours) | BodyGen      |
>    | ----------- | ----------------------- | ---------------------- |
>    | $0.2$ kg    | $\mathbf{8675.08_{\pm 286.42}}$  | $5186.31_{\pm 659.87}$ |
>    | $0.4$ kg    | $\mathbf{6966.34_{\pm 473.43}}$  | $5116.54_{\pm 546.55}$ |
>    | $0.6$ kg    | $\mathbf{7347.46_{\pm 478.01}}$  | $4523.89_{\pm 536.99}$ |
>
> **(ii) Power Constraint with Stronger Penalties**
>
> We further evaluate performance under stronger control-effort penalty coefficients (0.001, 0.01, 0.1), extending the original 0.0001 used in our main experiments.
> The table below show the results.
> The effects of larger penalties can be seen from the following three aspects:
>
> - *Effect in Performance (velocity reward):* As the penalty increases, both our method and BodyGen experience reduced performance, but our method consistently shows a smaller degradation across all settings.
>
> - *Effect in Control effort:* Larger penalties improve control efficiency for both methods, which is reflected in higher penalty-related rewards.
>
> - *Effect in Morphology–control:* Under stronger penalties, the co-designed solutions shift toward shorter, thicker, and more symmetric limbs, accompanied by a low-torque, conservative gait characteristic of energy-efficient locomotion.
>
> Table R2-2: Performance (velocity reward) and control effort penalty under different control-effort penalty coefficients.
>
>
>
> | Penalty Coef. | Performance |          | Control Effort Penalty |          |
> |---------------|-------------|----------|--------------|----------|
> |               | Stackelberg PPO (ours) | BodyGen  | Stackelberg PPO (ours) | BodyGen |
> | $0.0001$ | $\mathbf{11047.90_{\pm 126.20}}$ | $9098.72_{\pm 558.26}$ | $5631.42_{\pm 674.03}$ | $2745.84_{\pm 284.55}$ |
> | $0.001$  | $\mathbf{10191.15_{\pm 371.81}}$ | $7501.61_{\pm 671.33}$ | $5342.16_{\pm 584.25}$ | $5948.16_{\pm 254.84}$ |
> | $0.01$   | $\mathbf{9853.19_{\pm 229.37}}$  | $8304.00_{\pm 497.56}$ | $1582.48_{\pm 697.61}$ | $468.12_{\pm 516.33}$ |
> | $0.1$    | $\mathbf{10585.25_{\pm 146.80}}$ | $8974.48_{\pm 574.29}$ | $25.50_{\pm 22.27}$ | $26.34_{\pm 23.64}$ |
>
>
>
> *[continued in the next panel]*

---

> ### Author Response · Authors · 2025-11-27
> **Response to Reviewer ry9P (2/3)**
>
> **(iii) Torque-Limit Penalty**
>
> In addition to the *hard-coded* bounds on torque limits used in our existing experiments, we incorporate a torque violation penalty into the leader objective once the designed torque exceeds this bound.
>
> The table below summarizes performance and the torque-violation penalty under different penalty coefficients. With a small coefficient (0.01), our method outperforms BodyGen. When the penalty is increased to 0.1, the performance of all methods drops noticeably and the gap narrows.
>
> Table R2-3: Performance, torque violation penalty, and material cost under different penalty coefficients.
>
> | Penalty Coef. | Performance |          | Torque Violation Penalty  |          |
> |---------------|-------------|----------|------------------------|----------|
> |               | Stackelberg PPO (ours) | BodyGen | Stackelberg PPO (ours) | BodyGen |
> | $0.01$ | $\mathbf{7893.42_{\pm 84.62}}$ | $6311.75_{\pm 98.31}$ | $20210.50_{\pm 6503.22}$ | $11350.45_{\pm 5338.31}$ |
> | $0.1$  | $\mathbf{3133.01_{\pm 70.44}}$ | $3121.80_{\pm 54.03}$ | $1106.45_{\pm 64.69}$ | $899.35_{\pm 49.40}$ |
>
> **(iv) Manufacturability Penalty**
>
> In addition to the *hard-coded* physical bounds used in our existing experiments, we incorporate a manufacturability cost penalty into the leader objective, consisting of two components: structural complexity is measured by the number of body elements, and material cost is defined as the total mass.
>
> The table below summarizes the resulting performance and morphology characteristics under different penalty coefficients. The trends are consistent with those observed in the power penalty experiments in (ii): our method consistently achieves better reward–cost tradeoffs across all penalty levels.
> With a larger penalty coefficient, the evolved morphologies become more compact and structurally simpler, see Figure (iv) in [https://stackelberg-ppo-anonymous.netlify.app/re2](https://stackelberg-ppo-anonymous.netlify.app/re2) for the visualizations.
>
> Table R2-4: Performance, morphology complexity, and material cost under different penalty coefficients.
>
> | Penalty Coef. | Performance |          | Morphology Complexity |          | Material Cost |          |
> |---------------|-------------|----------|------------------------|----------|----------------|----------|
> |               | Stackelberg PPO (ours) | BodyGen  | Stackelberg PPO (ours) | BodyGen (baseline) | Stackelberg PPO (ours) | BodyGen  |
> | $0$   | $\mathbf{11047.90_{\pm126.20}}$ | $9098.72_{\pm558.26}$ | $16.40_{\pm2.45}$ | $13.67_{\pm2.08}$ | $1.71_{\pm0.23}$ | $1.57_{\pm0.32}$ |
> | $1$   | $\mathbf{7892.93_{\pm349.84}}$  | $6531.37_{\pm437.26}$ | $13.67_{\pm2.03}$ | $9.67_{\pm1.61}$  | $1.59_{\pm0.18}$ | $1.32_{\pm0.16}$ |
> | $10$  | $\mathbf{6825.47_{\pm303.09}}$  | $5372.10_{\pm364.79}$ | $8.25_{\pm0.91}$  | $7.50_{\pm1.24}$  | $0.94_{\pm0.08}$ | $0.93_{\pm0.10}$ |
>
> **(v) Robustness Evaluation**
>
> As the robustness is already inherently captured by the current evaluation protocol, we evaluate the robustness of the trained models under two settings: random *disturbance forces* applied to the root body at every control step, and *terrain friction noise* created by randomly varying the ground’s friction in each episode.
> As shown in the tables below, our method consistently demonstrates substantially higher robustness across all disturbance levels. For example, when increasing the disturbance force from 2 N to 6 N (Newtons), our method drops by only **–5.91%**, whereas BodyGen drops by **–59.57%**.
>
> Table R2-5: Performance under various levels of *disturbance forces*.
>
> | Level   | Stackelberg PPO (ours)  | BodyGen      |
> | ------- | ----------------------- | ---------------------- |
> | $2.0$ N | $\mathbf{11557.31_{\pm 124.68}}$ | $6963.05_{\pm 450.48}$ |
> | $4.0$ N | $\mathbf{11290.13_{\pm 164.54}}$ | $4621.82_{\pm 597.71}$ |
> | $6.0$ N | $\mathbf{10875.23_{\pm 250.97}}$ | $2816.16_{\pm 857.83}$ |
>
> Table R2-6: Performance under various levels of *terrain friction noise*.
>
> | Level   | Stackelberg PPO (ours)  | BodyGen      |
> | ------- | ----------------------- | ---------------------- |
> | $30%$  | $\mathbf{11424.66_{\pm 112.08}}$ | $7326.04_{\pm 421.73}$ |
> | $50%$  | $\mathbf{11333.43_{\pm 141.42}}$ | $6795.55_{\pm 493.31}$ |
> | $70%$  | $\mathbf{10892.09_{\pm 149.72}}$ | $5062.85_{\pm 579.66}$ |
>
> We have added these results in Appendix E.6 of the revised paper.

---

> ### Author Response · Authors · 2025-11-27
> **Response to Reviewer ry9P (3/3)**
>
> > W2: The $\lambda$/Fisher ablations are informative, but I’d like visibility into data efficiency: how many follower steps are actually saved vs PPO co-design? Since the thesis is “less re-optimization,” show sample-complexity curves for controller updates per viable morphology. (There is a follower-sampling hyperparameter, but not a direct “control-retraining cost” metric.)
>
> ### Answer:
>
> In all our experiments, **all methods use *exactly the same* total of 50 million saved follower steps** for co-design.
> The training curves with respect to the number of follower steps are already presented in Fig. 2 in the original submission.
> We also provide a table below summarizing the number of samples required to reach the performance threshold.
> Our method achieves substantially better sample efficiency than the other baselines.
>
> As a minor clarification, the "follower-sampling hyperparameter" refers to the number of follower samples used when updating the leader gradient in eq. (6).
>
> Table R2-7: Sample efficiency comparison: number of samples (million) required to achieve the performance threshold.
>
> | Environment      | Threshold | Stackelberg PPO (ours) | BodyGen | Transform2Act | NGE | ESS |
> |------------------|-----------|--------------------------|---------|----------------|-----|-----|
> | Crawler          | $\mathit{9000}$    | $\mathbf{25.8}$                   | $47.2$  | $\infty$       | $\infty$ | $\infty$ |
> | Cheetah          | $\mathit{11000}$   | $\mathbf{19.2}$                   | $42.1$  | $\infty$       | $\infty$ | $\infty$ |
> | Swimmer          | $\mathit{1200}$    | $\mathbf{14.8}$                    | $17.0$   | $\infty$         | $\infty$ | $\infty$ |
> | Walker-Hard      | $\mathit{10000}$   | $\mathbf{18.1}$                   | $30.3$  | $\infty$       | $\infty$ | $\infty$ |
> | Glider-Hard      | $\mathit{11000}$   | $\mathbf{23.6}$                   | $49.7$  | $\infty$       | $\infty$ | $\infty$ |
> | TerrainCrosser   | $\mathit{3500}$    | $\mathbf{23.9}$                   | $33.8$  | $\infty$       | $\infty$ | $\infty$ |
> | Pusher           | $\mathit{2500}$    | $\mathbf{29.3}$                   | $39.1$  | $\infty$       | $\infty$ | $\infty$ |
> | Stepper-Regular  | $\mathit{4500}$    | $\mathbf{18.5}$                   | $40.4$  | $\infty$       | $\infty$ | $\infty$ |
> | Stepper-Hard     | $\mathit{4500}$    | $\mathbf{27.2}$                   | $43.1$| $\infty$       | $\infty$ | $\infty$ |
>
> We appreciate the reviewer's comment and have clarified these points in the revised paper.
>
> ---
> > Q1: How sensitive are results to morphology editing horizon T?
>
> ### Answer:
>
> The table below reports the performance under different leader horizons $T$.
> Increasing the horizon generally improves performance by enabling richer and more powerful morphological edits.
> However, an excessively large horizon ($T=11$) introduces learning difficulties and causes some degradation, yet it still performs better than very small horizons ($T=3$).
>
> Table R2-8: Performance under various leader horizons in Stepper-Regular environment.
>
> | Leader Horizon | Stackelberg PPO (ours) | BodyGen    |
> | -------------- | ---------------------- | ---------------------- |
> | $3$            | $\mathbf{6188.99_{\pm 681.06}}$ | $3663.06_{\pm 571.30}$ |
> | $5$            | $\mathbf{7215.20_{\pm 449.02}}$ | $4685.94_{\pm 645.23}$ |
> | $7$            | $\mathbf{8260.74_{\pm 148.58}}$ | $5879.60_{\pm 175.41}$ |
> | $9$            | $\mathbf{6739.51_{\pm 631.35}}$ | $2975.11_{\pm 486.54}$ |
> | $11$           | $\mathbf{6874.34_{\pm 604.42}}$ | $2716.77_{\pm 457.61}$ |
>
> ---
> > Q2: Does the follower avoid full re-training when morphologies change significantly, or does the advantage mostly come from better leader gradients? Any evidence of faster controller adaptation?
>
> ### Answer:
>
> The reviewer’s intuition is correct: **the advantage comes from better leader gradients rather than faster follower adaptation.** This is supported by our comparison with BodyGen, where the only difference is the leader’s gradient; starting from the same initialization, our method consistently outperforms BodyGen across almost all environments (see Fig. 2 in our original submission).
>
> To further support this point, we conduct an additional experiment:
> we take *10 intermediate checkpoints* from BodyGen’s training process (e.g., at 1/10, 2/10, … of training), and train both BodyGen and our method from these checkpoints for *one epoch*, then compare the resulting performance changes.
>
> Table R2-9: Average performance change after one epoch of training from the same checkpoint model, averaged over 10 checkpoints and 7 random seeds, evaluated on environment Stepper-Regular.
>
> |                                  | Stackelberg PPO (Ours) | BodyGen |
> | -------------------------------- | ---------------------- | ------- |
> | Performance Change After 1 Epoch |         $\mathbf{+0.392_{\pm0.075}%}$             |    $+0.224_{\pm0.043}%$    |

---

### Official Review · Reviewer_B8Qz · 2025-11-03

**Soundness:** 3
**Presentation:** 3
**Contribution:** 2
**Rating:** 6
**Confidence:** 3

**Summary:**

The paper formulates morphology–control co-design as a phase-separated Stackelberg Markov Game (SMG) where a leader edits morphology via discrete actions and a follower optimizes control on the resulting body. Because the leader–follower interface is non-differentiable, the authors derive a trajectory likelihood-ratio loss for the Stackelberg cross-derivative and use the natural gradient approximation as a stable inverse-Hessian proxy. Combining everything gives the Stackelberg PPO. The experiments on various mujoco-based morphology control tasks demonstrate the competitiveness of the proposed method.

**Strengths:**

1. The formulation of morphology–control co-design as a phase-separated Stackelberg game is intuitive and matches the causal structure between design and control.

2. The paper gives a clear algorithmic pipeline that can be implemented with existing PPO infrastructure.

3. Experimental results are consistent across several benchmarks and ablations are reasonably detailed.

**Weaknesses:**

1. The technical novelty is modest, since the likelihood-ratio surrogate, natural gradient, and PPO clipping are all existing techniques; the contribution lies mostly in integrating them coherently.

2. the cross-derivative estimator in eq. (6) seems to have very high variance.

3. The baselines do not include prior Stackelberg RL algorithms (e.g., Stackelberg Actor-Critic), so it is unclear whether the improvement arises from the Stackelberg formulation itself.

**Questions:**

The natural policy gradient is more computationally expensive than regular PPO. How does the computation efficiency compare with the other baselines, given the experiments are all in simulation?

---

> ### Author Response · Authors · 2025-11-27
> **Response to Reviewer B8Qz (1/3)**
>
> We appreciate the insightful and helpful comments from Reviewer B8Qz. Please find our responses to your concerns and questions below.
>
> > W1: The technical novelty is modest, since the likelihood-ratio surrogate, natural gradient, and PPO clipping are all existing techniques; the contribution lies mostly in integrating them coherently.
>
> ### Answer:
>
> Thank you for raising this point. We believe there has been a small misunderstanding, and we clarify it here. *The core contribution* of our work is not a naive integration of existing tools. Rather, we derive a new, tractable Stackelberg gradient formulation for a non-differentiable leader–follower setting, grounded in new theoretical foundations.
> While terms such as “likelihood-ratio,” “natural gradient,” and “clipping” coincide in terminology with existing methods, **the optimization target, mathematical form, and theoretical basis in our approach are completely new** for this Stackelberg setting, and do **not** appear in prior RL or co-design methods.
>
> Beyond the core contribution above, *there are also non-trivial differences at the component level*, and we highlight the key ones below.
>
> 1. **Our likelihood-ratio surrogate in eq. (6) is fundamentally different from existing likelihood-ratio surrogates:**
>
>    - **Different setting:** Eq. (6) is derived for a leader–follower Stackelberg game, whereas existing likelihood-ratio surrogates are designed for the single-agent setting. These two settings involve entirely different optimization objectives.
>
>    - **Different mathematical form:**
>    A naïve approach would be to directly reuse the standard single-agent likelihood-ratio surrogate for the leader updates, like PPO in BodyGen [r1].
>    Instead, our derivation under the Stackelberg objective yields a distinct mathematical structure involving leader–follower coupling terms that do not arise in single-agent RL.
>      To the best of our knowledge, *this form in eq. (6) has never appeared in prior RL or co-design literature.*
>
> 2. **Our use of PPO-style clipping is *not* a simple reuse of standard PPO clipping:**
>
>
>
>    - **Different theoretical basis:** Our use of clipping is grounded in a novel local-approximation theory (Theorem 1), which establishes that eq. (6) locally approximates the true Stackelberg cross-derivative. This is fundamentally different from existing local approximation theories in a single-agent setting.
>    To the best of our knowledge, *this local-approximation theory has not appeared in either prior RL or co-design work.*

---

> ### Author Response · Authors · 2025-11-27
> **Response to Reviewer B8Qz (2/3)**
>
> > W2: the cross-derivative estimator in eq. (6) seems to have very high variance.
> ### Answer:
>
> Thanks for this very insightful comment.
> The original form in eq. (6) may *appear* high-variance because written in the trajectory-based representation.
> However, in practice, we implement this operator using an equivalent state-visitation form, where the variance structure becomes much clearer and does not exhibit the apparent high variance.
>
> $$
> L_{{L,F}}^{F}(\theta^L, \theta^F; \theta_o^L, \theta_o^F) = c E \Biggl[ \frac{\pi_{\theta^L}^{L}( a^L |s^L )}{\pi_{\theta_o^L}^{L}( a^L |s^L )} \Biggl[ \gamma^T E \Biggl[ \frac{\pi_{\theta^F}^{F}( a^F |s^F ;s_T^L )}{\pi_{\theta_o^F}^{F}( a^F |s^F ;s_T^L )} A_{\pi_{\theta_o^F}^{F}}^{F}( s^F ,a^F ;s_T^L ) \Biggr] \Biggr] \Biggr]
> $$
>
> The outer expectation is taken over
> $s^{L} \sim d_{{\theta_o^L}}^{L},\ a^{L}\sim \pi_{{\theta_o^L}}^{L},\ s_{T}^{L} \sim d_{\theta_o^L}^{L,T}$,
> where
> $d_{\theta_o^L}^{L,t}(s^{L}) = P(s_t^{L}=s^{L};\pi_{{\theta_o^L}}^{L})$ is the visitation distribution probability of leader policy at step $t$,
> and
> $d_{\theta_o^L}^L (s^L) \triangleq 1 / T \sum_t d_{\theta_o^L}^{L,t} (s^L)$.
> The inner expectation is taken over
> $s^F \sim d_{\theta_o^F}^F(\cdot; s_T^L),\ a^F \sim \pi_{{\theta_o^F}}^{F}(\cdot; s_T^L)$,
> where $d_{\theta_o^F}^F$ denotes the follower’s visitation distribution.
> The constant $c = T/(1-\gamma)$ normalizes the distribution; in practice its effect can be absorbed by tuning the learning rate.
>
> Note that this transformation mirrors the standard equivalence between the trajectory form and visitation form in REINFORCE [r2], confirming that this estimator does not suffer from abnormally high variance.
>
> In practice, we exactly take the average over the samples sampled from the distribution rather than taking the cumulative.
> As shown in the table below, increasing the leader horizon does *not* introduce higher variance in the updates from eq. (6) compared to the BodyGen baseline [r1].
>
> Table R1-1: Mean and standard deviation of performance across different leader horizons.
>
> | Leader Horizon | Stackelberg PPO (ours) | BodyGen     |
> | -------------- | ---------------------- | ---------------------- |
> | $3$            | $\mathbf{6188.99_{\pm 681.06}}$ | $3663.06_{\pm 571.30}$ |
> | $5$            | $\mathbf{7215.20_{\pm 449.02}}$ | $4685.94_{\pm 645.23}$ |
> | $7$            | $\mathbf{8260.74_{\pm 148.58}}$ | $6879.60_{\pm 175.41}$ |
> | $9$            | $\mathbf{6739.51_{\pm 631.35}}$ | $3375.11_{\pm 486.54}$ |
> | $11$           | $\mathbf{6874.34_{\pm 604.42}}$ | $3216.77_{\pm 657.61}$ |
>
> We have updated this in Figure 4(c) in the revised paper.
>
> ---
> > W3: The baselines do not include prior Stackelberg RL algorithms (e.g., Stackelberg Actor-Critic), so it is unclear whether the improvement arises from the Stackelberg formulation itself.
> ### Answer:
>
> **Prior Stackelberg RL methods (e.g., Stackelberg MADDPG [r3]) are fundamentally inapplicable to our setting with non-differentiable leader–follower interfaces**, as they all rely on differentiable interfaces to enable direct backpropagation.
> In contrast, we introduce a new approach that performs Stackelberg Implicit Differentiation (SID) **without** needing differentiable interfaces, thereby enabling SID in settings *where previous methods simply cannot operate*.
>
> Since prior SID derivations cannot be applied in our non-differentiable setting, we instead perform a controlled ablation that isolates the effects of our new SID estimator and PPO clipping by evaluating the following variants:
>
> 1. PPO clipping + our new SID estimator,
> 2. PPO-*only* (without SID), and
> 3. SID-*only* (without PPO clipping),
>
> The results in the table below show that **both our new SID estimator and PPO clipping provide clear performance gains**.
> We have included these new results in Appendix E.3 of the revised paper.
>
> Table R1-2: Ablation studies on the components of *our new SID estimator* and *PPO clipping*, under the same phase-separated, non-differentiable setting.
>
> | Environment | SID+PPO (full) | PPO-*only* (without SID) | SID-*only* (without PPO clipping) |
> | ----------- | -------------- | ------------------------ | --------------------------------- |
> |Stepper-Regular|$\mathbf{7215.20_{\pm 449.02}}$|$4685.94_{\pm 845.23}$|$1257.33_{\pm 530.25}$|
> |Crawler| $\mathbf{11047.90_{\pm 126.20}}$|$9098.72_{\pm 558.26}$|$35.77_{\pm 12.25}$|
> |Cheetah| $\mathbf{13514.94_{\pm 653.62}}$|$11575.87_{\pm 640.65}$|$472.89_{\pm 77.40}$|
> |Glider| $\mathbf{12414.50_{\pm 498.53}}$|$11049.95_{\pm 468.44}$|$566.81_{\pm 89.96}$|

---

> ### Author Response · Authors · 2025-11-27
> **Response to Reviewer B8Qz (3/3)**
>
> > Q1: The natural policy gradient is more computationally expensive than regular PPO. How does the computation efficiency compare with the other baselines, given the experiments are all in simulation?
> ### Answer:
>
>
> To address the reviewer’s concern, we summarize both wall-clock training time and sample-efficiency results in the tables below (for training curves, see Fig. 2 in the original submission).
>
> - *Compared to BodyGen*, our method achieves substantially better sample efficiency by requiring **-39%** fewer samples to reach the performance threshold while also obtaining **+20.66%** higher final scores.
> In terms of wall-clock time, the reviewer is correct that our method requires extra training cost, but the overhead is modest (+13%), keeping the overall cost comparable.
>
> - *Compared to ES baselines*, although ESS attains shorter wall-clock time using *6×* more CPU cores (64 cores), its performance is extremely poor, achieving only a **0.16** fraction of our method’s performance.
>
> Overall, our method delivers strong sample efficiency and performance while remaining competitive in training wall-clock time.
> We have updated these results in Appendix E.4 in the revised paper.
>
> Table R1-3: Sample efficiency comparison: number of samples required to achieve the performance threshold.
>
> | Environment      | Threshold | Stackelberg PPO (ours) | BodyGen | Transform2Act | NGE | ESS |
> |------------------|-----------|--------------------------|---------|----------------|-----|-----|
> | Crawler          | $\mathit{9000}$    | $\mathbf{25.8}$                   | $47.2$  | $\infty$       | $\infty$ | $\infty$ |
> | Cheetah          | $\mathit{11000}$   | $\mathbf{19.2}$                   | $42.1$  | $\infty$       | $\infty$ | $\infty$ |
> | Swimmer          | $\mathit{1200}$    | $\mathbf{14.8}$                    | $17.0$   | $\infty$         | $\infty$ | $\infty$ |
> | Walker-Hard      | $\mathit{10000}$   | $\mathbf{18.1}$                   | $30.3$  | $\infty$       | $\infty$ | $\infty$ |
> | Glider-Hard      | $\mathit{11000}$   | $\mathbf{23.6}$                   | $49.7$  | $\infty$       | $\infty$ | $\infty$ |
> | TerrainCrosser   | $\mathit{3500}$    | $\mathbf{23.9}$                   | $33.8$  | $\infty$       | $\infty$ | $\infty$ |
> | Pusher           | $\mathit{2500}$    | $\mathbf{29.3}$                   | $39.1$  | $\infty$       | $\infty$ | $\infty$ |
> | Stepper-Regular  | $\mathit{4500}$    | $\mathbf{18.5}$                   | $40.4$  | $\infty$       | $\infty$ | $\infty$ |
> | Stepper-Hard     | $\mathit{4500}$    | $\mathbf{27.2}$                   | $43.1$| $\infty$       | $\infty$ | $\infty$ |
>
> Table R1-4: Wall-clock training time, under 10 CPU cores and A100 GPU averaged over 9 environments. The ES baseline (NGE) is also shown with 64 CPU cores to illustrate its parallelization advantage.
>
> |                | Stackelberg PPO(Ours , 10 CPU cores) | BodyGen (PPO, 10 CPU cores) | NGE (ES-based, 10 CPU cores) | NGE (ES-based, 64 CPU cores) |
> | -------------- | ------------------------------------ | --------------------------- | ---------------------------- | ---------------------------- |
> | Wallclock Time (hours) | $33.07_{\pm 0.49}$             | $29.43_{\pm 0.97}$     | $45.16_{\pm 3.72}$      | $13.52_{\pm 1.52}$      |
>
> References
>
> [r1] Lu, Haofei, et al. “BodyGen: Advancing Towards Efficient Embodiment Co-Design.” International Conference on Learning Representations (ICLR 2025).
>
> [r2] Sutton, Richard S. et al. “Policy Gradient Methods for Reinforcement Learning with Function Approximation.” Advances in Neural Information Processing Systems (NeurIPS 2000).
>
> [r3] Yang, Boling, et al. “Stackelberg Games for Learning Emergent Behaviors during Competitive Autocurricula.” IEEE International Conference on Robotics and Automation (ICRA 2023).

---

### Author Response · Authors · 2025-12-04
**Summary for the Area Chair (1/3)**

Dear Area Chairs,

We sincerely thank you for your time and thoughtful oversight throughout the review process. We also sincerely appreciate the reviewers for their constructive feedback, which has helped strengthen the paper.

For the AC’s convenience, we summarize below the *ratings*, the *main contributions*, and the reviews organized both *by reviewer* and *by concern type*.

### I. Summary of the Reviewers' Ratings

We first summarize the current reviewers' ratings and feedback:

- **Three reviewers have confirmed positive evaluations** (B8Qz and ry9P originally *gave a score of 6*, and 1urx acknowledged our response and explicitly stated that *they would raise the score*).
- Due to the known system issue, other reviewers (vMFr and Md1x) were unable to post their responses before the discussion was unexpectedly locked.
Nevertheless, we are confident that our response sufficiently addresses these concerns, supported by solid experimental results as well as clarifications and revisions in the manuscript. For completeness, we summarize the main points below:

    - Reviewer vMFr mainly requested

         - *Additional experiments on broader scenarios or ablations*. We clarified that **many of these experiments *already appear in our original submission***, and **we further expanded them to broader settings using newly created environments that *prior work has not explored*** in our response (see our response to W1-W2, W4, Q3-Q5 for details).
         - *Clarifications on empirical properties* (e.g., efficiency, bias, and variance). Our response clearly reported all of these empirical properties (see our response to W3, W5, Q2).

    - Reviewer Md1x primarily raised concerns about

        - *Implementation complexity*, to which we responded by
        i) demonstrating that the additional computations are **lightweight and scale linearly**,
        ii) providing **open-source code**, and
        iii) reporting wall-clock measurements confirming that the runtime remains comparable to the baselines (see our response to W1, W4).
        - *Clarifications regarding related work*, which have been fully addressed in the revised manuscript with added citations and a clearer discussion of differences from prior work (see our response to W2, W3).

---

### II. Main Contributions and Corresponding Feedback from the Reviewers

We are pleased that reviewers consistently acknowledged the novelty and contributions of our work [r1-r4], particularly highlighting the **new theoretical viewpoint and algorithmic pipeline**, as well as the **great performance gains and efficiency improvements** demonstrated across challenging benchmarks [r5-r8].

Below, we summarize the main contributions together with the corresponding reviewer feedback, with the referenced comments r1–r8 provided in Sec. VI for completeness and transparency.

1. **Novel Variant of the Problem Formulation.**

    We highlight that morphology–control co-design corresponds to a novel variant of a Stackelberg Markov game, characterized by a phase-separated and non-differentiable leader and follower process that presents the primary challenge in leveraging the coupling structure.


    Reviewer feedback:

    - *Clear problem formulation* (B8Qz, ry9P, vMFr, Md1x) [r1]
    - *Good insight in matching the reality structure* (B8Qz, ry9P, vMFr) [r2]

2. **Novel and Tractable Method with Solid Theoretical Foundations.**

    We propose a tractable Stackelberg policy gradient method that leverages intrinsic leader–follower coupling, supported by solid theoretical analysis, to enable more efficient leader updates.

    Reviewer feedback:

    - *Meaningful algorithmic advance* (B8Qz, ry9P, vMFr) [r3]
    - *Principled mathematical analysis* (vMFr, 1urx) [r4]

3. **Strong Empirical Results.**

    Our method, Stackelberg PPO, outperforms state-of-the-art baselines by **+20.66%** across all 9 tasks and by **+32.02%** on 4 complex 3D tasks.

    Reviewer feedback:

    - *Strong performances* (ry9P, vMFr, Md1x) [r5]
    - *Robustness and generality across diverse benchmarks* (B8Qz, ry9P, 1urx) [r6]
    - *Notable improvements in sample efficiency* (ry9P, vMFr) [r7]
    - *Supportive ablation studies* (B8Qz, ry9P, Md1x) [r8]

To the best of our knowledge, this is the first work to explicitly incorporate the leader–follower coupling structure into the update process, enabling more efficient morphology discovery.
As Reviewer 1urx noted:
> *“How to balance the bi-level optimization of co-design is indeed a critical point for this area. This paper introduces Stackelberg Game, making this paper a promising solution for this target.”*

---

> ### Author Response · Authors · 2025-12-04
> **Summary for the Area Chair (2/3)**
>
> ### III. Summary of the Reviews Grouped by Reviewers
>
> We organize the reviews *per reviewer*. Table S1 summarizes each reviewer's main concerns and our responses.
>
>
> Table S1: Summary of all reviewers' concerns and our responses, grouped by the reviewers.
>
> |Reviewer| W/Q  | Key Concerns                             | Our Response                                                 |
> | -------- | ---- | ---------------------------------------- | ------------------------------------------------------------ |
> | **B8Qz** | W1   | Novelty concerns due to overlap in tools | We clarified that *the optimization objective, mathematical structure, and theoretical foundations* of our approach are **entirely new** in the Stackelberg setting and **do not appear** in prior RL or co-design methods. |
> |          | W2   | Concerns about variance in eq. (6)       | We show that the variance **is not high** by providing an *equivalent reformulation* of the objective and reporting *empirical variance comparable* to the baselines. |
> |          | W3   | Missing “prior Stackelberg RL” baselines | We clarified that **prior Stackelberg RL works are fundamentally *inapplicable* to our setting**. Nonetheless, we provided additional empirical ablations based on our components for completeness. |
> |          | Q1   | Computational efficiency                 | We reported computational cost, showing that our method remains *comparable* to the baselines. |
> | **ry9P** | W1     | Experiments on broader scenarios                             | We clarified that **many of these experiments were already included in the original submission**, and we further *expanded them to broader settings using newly created environments that **prior work has not explored***. |
> |          | W2, Q1 | Empirical properties (efficiency and sensitivity with morphology editing horizon $T$) | We provided *additional empirical results* on the efficiency details, and new horizon-$T$ ablations. |
> |          | Q2     | Source of performance gain                                   | We clarified the underlying mechanism and provided *supporting empirical evidence*. |
> | **vMFr** | W4, Q4, Q5 | Experiments on broader scenarios (real-world feasibility)    | We clarified that **many of these experiments were already included in the original submission**, and we further *expanded them to broader settings using newly created environments that **prior work has not explored***. |
> |          | W1–3, Q1–3 | Empirical properties (efficiency, bias/variance), ablations (SID term, PPO clipping), and additional seeds | We provided all requested *empirical analyses*, including efficiency studies, bias/variance reports, ablations, and additional seed evaluations. |
> | **1urx** | W1    | Open-source code                                             | We have released the **open-source code** along with visualization demonstrations. |
> |          | Q1    | Differences from prior co-design formulations                | We clearly clarified the differences in both the problem formulation and the algorithmic objective. |
> |          | Q2–Q3 | Questions about mechanism details (effect of controller adaptation) | We clarified the underlying mechanism and provided supporting experimental evidence. |
> |          | Q4    | Concerns about benchmark environments                        | We explained that all benchmarks used in prior work **were already included** in the original submission and have now been moved to the main paper. |
> | **Md1x** | W1, W4 | Implementation complexity                             | We clarified that *the additional computations are lightweight and scale linearly*, provided *open-source code*, and reported *practical wall-clock measurements* showing comparable cost to the baselines. |
> |          | W2, W3 | Clarification on related work (wording and citations) | We refined the manuscript with *additional citations* and a *clearer discussion of the differences* from prior work. |
> |          | W4     | Applicability to other domains such as NAS            | We agreed that the formulation is broadly applicable and noted its potential extension to other bi-level settings. |
>
> ---

---

> ### Author Response · Authors · 2025-12-04
> **Summary for the Area Chair (3/3)**
>
> ### IV. Summary of the Reviews Grouped by Concern Types
>
> To facilitate a clearer overview for ACs, we group the reviewers’ comments by concern type, including further clarifications, additional details or results, and broader-scenario evaluations.
>
> Table S2: **Clarification of the points that required clearer explanation**. The manuscript has also been revised to prevent potential misunderstandings.
>
> | W/Q | Key Concerns | Our Response |
> | - | - | - |
> | B8Qz (W1) | Novelty concerns due to overlap in tools | We clarified that *the optimization objective, mathematical structure, and theoretical foundations* of our approach are **entirely new** in the Stackelberg setting and **do not appear** in prior RL or co-design methods. |
> | 1urx (Q1) | Differences from prior co-design formulations| We clearly clarified the differences in both the problem formulation and the algorithmic objective. |
> | B8Qz (W2) | Concerns about variance in eq. (6)| We show that the variance **is not high** by providing an *equivalent reformulation* of the objective and reporting *empirical variance comparable* to the baselines. |
> | B8Qz (W3) | Missing “prior Stackelberg RL” baselines | We clarified that **prior Stackelberg RL works are fundamentally *inapplicable* to our setting**. Nonetheless, we provided additional empirical ablations based on our components for completeness. |
> | Md1x  (W1, W4) | Implementation complexity | We clarified that *the additional computations are lightweight and scale linearly*, provided *open-source code*, and reported *practical wall-clock measurements* showing comparable cost to the baselines. |
> | 1urx (Q4) | Concerns about benchmark environments | We explained that all benchmarks used in prior work **were already included** in the original submission and have now been moved to the main paper. |
>
> ---
>
> Table S3: **Additional details and quantification** requested by reviewers, including further empirical analyses, ablations. We have added these additional details in the revised paper.
>
> |W/Q|Key Concerns|Our Response|
> | - | - | - |
> | B8Qz (Q1), ry9P (W2), vMFr (W3) | Computational efficiency                                     | We reported computational cost, showing that our method remains *comparable* to the baselines. |
> | ry9P (Q1), vMFr (W1–2, Q1–3)    | Empirical properties (bias/variance, sensitivity with morphology editing horizon $T$), ablations (SID term, PPO clipping), and additional seeds | We provided all requested *empirical analyses*, ablations, and additional seed evaluations. |
> | 1urx (W1)| Open-source code| We have released the **open-source code** along with visualization demonstrations. |
> | Md1x (W2, W3)| Clarification on related work (wording and citations)| We refined the manuscript with *additional citations* and a *clearer discussion of the differences* from prior work. |
> | ry9P (Q2)| Source of performance gain| We clarified the underlying mechanism and provided *supporting empirical evidence*. |
> | 1urx (Q2–Q3)| Questions about mechanism details (effect of controller adaptation) | We clarified the underlying mechanism and provided supporting experimental evidence. |
>
> ---
>
> Table S4: **Concerns about extension to broader scenarios.**
> We clarified that our original submission already included several broad scenarios (e.g., energy usage, morphology constraints). Nevertheless, we further extended the experiments to new settings that prior work has not explored. *These additional results further demonstrate the strong potential of our proposed method.*
> We have added these additional details in the revised paper.
>
> |W/Q|Key Concerns|Our Response|
> | - | - | - |
> |ry9P (W1), vMFr (W4, Q4, Q5)| Experiments on broader scenarios| We clarified that **many of these experiments were already included in the original submission**, and we further *expanded them to broader settings on newly created environments that **prior work has not explored***. |
> |Md1x (W4)| Applicability to other domains such as NAS | We agreed that the formulation is broadly applicable and noted its potential extension to other bi-level settings. |
>
> ---
>
> ### V. Concluding Remarks
>
> We hope that the clarifications and additional analyses provided during the rebuttal process help convey the value and contributions of this work. We again thank the Area Chairs and reviewers for their valuable time and careful consideration.
>
> Best regards,
>
> The Authors

---

> > ### Author Response · Authors · 2025-12-04
> > **Reference to the Reviewers' Comments**
> >
> > ### VI.  Reference to the Reviewers' Comments
> >
> > [r1]
> > > B8Qz: The formulation ... is **intuitive**...
> > >
> > > ry9P: **a clean bilevel** control–morphology structure.
> > >
> > > vMFr: **Clear problem framing** for non-differentiable interfaces.
> > >
> > > Md1x: **a rigorous reformulation**, which formulates the brain-body co-design problem as a Stackelberg game.
> >
> > [r2]
> > >
> > > B8Qz: **The formulation matches the causal structure** between design and control
> > >
> > > ry9P: **a good insight** and brings in notable improvement; **interesting and well-motivated**;
> > >
> > > vMFr: **precisely matches co-design realities**, addressing why prior Stackelberg methods fail here
> >
> > [r3]
> > >
> > > B8Qz: **a clear algorithmic pipeline** that can be implemented with existing PPO infrastructure.
> > >
> > > ry9P: The derivation ... **creates a tractable gradient** signal for the leader without assuming differentiability of morphology parameters...
> > **a meaningful algorithmic advance** over evolutionary search or unrolled gradient-based bilevel RL
> > >
> > > vMFr: PPO-style clipping and Fisher (natural-gradient) Hessian approximation are **well-motivated for practical computation**.
> >
> > [r4]
> > >
> > > vMFr: ...(Theorem 1) avoids differentiating through morphology transitions and yields local equivalence to true Stackelberg gradients. **This is practical for implementation, easy for adoption with downstream ideas.**
> > >
> > > 1urx: **principled mathematical analysis** with explicit gradient derivations under the Stackelberg formulation; **theoretical insights connect naturally** with the empirical benefits
> >
> > [r5]
> > >
> > > ry9P: brings in **notable improvement**; **better** stability and **final reward** than PPO-based co-design and evolutionary baselines
> > >
> > > vMFr: **better** sample efficiency and **final reward** across six MuJoCo tasks
> > >
> > > Md1x: **significant improvements** in final performance; The experiments are **comprehensive and convincing**
> >
> > [r6]
> > >
> > > B8Qz: Experimental results are **consistent across several benchmarks**
> > >
> > > ry9P: **better stability** and final reward than PPO-based co-design and evolutionary baselines
> > >
> > > 1urx: The comparisons against **strong and diverse baselines** demonstrate the **robustness and generality** of the approach; The experiments cover multiple challenging co-design benchmarks, including both 2D and 3D locomotion as well as complex terrain settings
> >
> > [r7]
> > >
> > > ry9P: **Training efficiency improvement is notably higher** than prior art
> > >
> > > vMFr: **better sample efficiency** and final reward across six MuJoCo tasks
> >
> > [r8]
> > >
> > > B8Qz: **ablations are reasonably detailed**
> > >
> > > ry9P: The $\lambda$/Fisher **ablations are informative**
> > >
> > > Md1x: The paper includes **thorough ablation studies** that validate key design choices

---

### Meta-Review · Area_Chair_9muJ · 2025-12-30

**Summary:**

The paper presents an interesting Stackelberg formulation for morphology–control co-design and demonstrates strong empirical performance across challenging benchmarks. However, reviewers consistently expressed concern that the algorithmic novelty is limited, as the approach appears incremental relative to prior bilevel and joint co-design methods. Several reviewers questioned whether the Stackelberg leader–follower formulation is essential to the observed gains, as opposed to improvements stemming primarily from PPO-style stabilization and engineering choices. There were also concerns about the method’s increased algorithmic complexity and practical scalability, which reviewers felt may outweigh the demonstrated benefits.

**Reviewer Concerns:**

From an AC perspective, it remains somewhat unintuitive whether the Stackelberg leader–follower formulation is the most essential and decisive factor for the observed performance gains. However, at the same time, the additional experiments provided by the authors offer meaningful empirical evidence that helps clarify its role and partially addresses this concern. While the algorithm introduces a certain level of implementation complexity, it is sufficiently novel and technically interesting for the community, and the authors have indicated that open-source code will be made available. Overall, I believe most of the reviewers’ concerns have been reasonably addressed by the rebuttal.

**Reviewer Scores:**

One reviewer explicitly indicated that they would raise their score. The remaining reviewers did not suggest any score decreases during the discussion, and their overall assessments appear largely stable.

---

### Decision · Program_Chairs · 2026-01-26

Accept (Poster)